# Comparative single-cell analyses reveal evolutionary repurposing of a conserved gene programme in bat wing development

Bats are the only mammals capable of self-powered flight, an evolutionary innovation based on the transformation of forelimbs into wings. The bat wing is characterized by an extreme elongation of the second to fifth digits with a wing membrane called the chiropatagium connecting them. Here we investigated the developmental and cellular origin of this structure by comparing bat and mouse limbs using omics tools and single-cell analyses. Despite the substantial morphological differences between the species, we observed an overall conservation of cell populations and gene expression patterns including interdigital apoptosis. Single-cell analyses of micro-dissected embryonic chiropatagium identified a specific fibroblast population, independent of apoptosis-associated interdigital cells, as the origin of this tissue. These distal cells express a conserved gene programme including the transcription factors *MEIS2* and *TBX3*, which are commonly known to specify and pattern the early proximal limb. Transgenic ectopic expression of *MEIS2* and *TBX3* in mouse distal limb cells resulted in the activation of genes expressed during wing development and phenotypic changes related to wing morphology, such as the fusion of digits. Our results elucidate fundamental molecular mechanisms of bat wing development and illustrate how drastic morphological changes can be achieved through repurposing of existing developmental programmes during evolution.

Evolution has fuelled the emergence of a remarkable variety of phenotypes throughout the animal kingdom. In particular, the vertebrate limb displays many fascinating adaptations[1,2] and has long served as a prime example to study the genetic basis of phenotypic evolution[3,4]. An extreme example is the evolution of forelimbs (FLs) into wings in bats (order Chiroptera), the only mammals capable of self-powered flight. Interestingly, the oldest known bat fossil already presents wing-structured FLs, suggesting that flight originated in the most recent common ancestor of all bats[5]. Bat wings are thus a unique and ancient structure, representing an exceptional model for studying limb diversification. Likewise, examining the development of wings can shed light on the mechanisms underlying morphological transformations in evolution[6,7].

During development, limb buds arise from the lateral plate mesoderm (LPM) under the control of three distinct signalling centres: the zone of polarizing activity, the dorsal and ventral ectoderm, and the apical ectodermal ridge[8,9]. These centres confer cellular identity along the anterior–posterior, dorsal–ventral and proximo–distal axes, respectively. Outgrowth along the proximo–distal axis results in the formation of three distinct elements: most proximally the stylopod, followed by the zeugopod and distally the autopod, corresponding to humerus/femur, radius–ulna/tibia–fibula and hand/foot, respectively[10] (Fig. 1a). The bat FL is characterized by elongation of all skeletal elements as well as the presence of membranes, which form the wing. Changes are most pronounced in the autopod, with extremely elongated digits II–V and an interdigital wing membrane connecting

✉e-mail: dario.lupianez@csic.es; mundlos@molgen.mpg.de; fmarrea@upo.es

them, known as the chiropatagium. In contrast, in bat hindlimbs (HLs) and most other pentadactyl species including humans and mice, the tissue between the digits recedes during development resulting in separate digits (Fig. 1a).

Experiments across different species have shown that retinoic acid (RA)-induced apoptosis of interdigital cells plays a central role in digit separation[11,12]. Consequently, one hypothesis for the persistence of interdigital tissue in bats is the suppression of this apoptotic process. Several studies have addressed this hypothesis; however, the results have been inconclusive. Both pro- and anti-apoptotic markers were found to be expressed in the developing chiropatagium[13,14]. In addition, several comparative molecular studies have identified genes with altered patterns of expression in developing wings[15–17]. However, the molecular and evolutionary bases of wing morphology development remain largely unknown, partially due to the limitations of the available methodologies at the time. Recently, single-cell approaches have provided new tools to investigate cell identity and function at unprecedented resolution in many organisms, holding great potential to unravel the basis of evolutionary innovation[18]. Yet, how cell fates are molecularly determined and sustain the emergence of new morphologies remains one of the big unsolved questions in biology.

To investigate the molecular origins of wing formation, we performed single-cell RNA sequencing (scRNA-seq) at multiple time points during bat and equivalent mouse embryonic limb development. Our data reveal conserved cell clusters and gene expression patterns across species, including within the apoptosis-related cell population. Additionally, we characterized the origin of the chiropatagium, which is composed of fibroblastic cells that follow a differentiation trajectory independent of RA-active interdigital cells and repurpose a gene programme typically restricted to the proximal limb. By ectopically expressing two upstream transcription factors (TFs) of this programme, *MEIS2* and *TBX3*, in the distal limb of transgenic mice, we recapitulated key molecular and morphological features observed in developing bat wings. Altogether, our findings demonstrate that an existing proximal cell state and its gene regulatory programme are repurposed in the distal bat FL to generate a novel tissue in a different spatial location.

## Results

### Conservation of cellular composition and interdigital cell death
We collected FLs and HLs for scRNA-seq from mice and bats (*Carollia perspicillata*) covering critical developmental stages of digit separation and wing formation. Samples included an early, morphologically undifferentiated stage (embryonic day (E)11.5 in mice and equivalent to CS15 stage in bats[19]) and a later stage in which the digits form and separate (E13.5 in mice and CS17 in bats); we also included an intermediate time point (E12.5) from mice (Fig. 1b). Using the Seurat v3 single-cell integration tool, we generated an interspecies single-cell transcriptomics limb atlas (Fig. 1c). Cells from both species contributed similarly to all cell clusters (Extended Data Fig. 1). We identified all major cell populations known to be present in developing limbs, including muscle, ectoderm-derived and LPM-derived cells[20–22] (Fig. 1c and Extended Data Fig. 1). Overall, both the composition and identity of limb cells are largely conserved between the species despite notable morphological differences.

As the LPM contributes to the formation of interdigital mesenchyme, cartilage, tendons and other connective tissues within the limb, we specifically focused on this lineage. The LPM-derived cells were further subdivided into 18 clusters and annotated by performing differential gene expression analysis. Based on the calculated markers and previous studies[23,24], we identified three main cell lineages: chondrogenic, fibroblast and mesenchymal (Fig. 1d). The expression of the marker genes used for cluster annotation (Fig. 1e), and marker genes differentially expressed in each cluster (Extended Data Fig. 1), was conserved across species.

Using this interspecies single-cell atlas, we first sought to address the prevailing hypothesis that chiropatagium development is driven by inhibition or reduction of apoptotic cell death in the interdigital tissue[13]. We identified a cluster of interdigital cells characterized by high expression levels of *Aldh1a2* and *Rdh10*, components of RA signalling. RA is regarded as a pivotal regulator of interdigital apoptosis and its expression pattern has been extensively employed to discern the interdigital tissue[25]. Cells from this cluster (3 RA-Id) also expressed main pro-apoptotic factors, including *Bmp2* and *Bmp7*, highlighting it as a central population of apoptotic signalling in both species (Fig. 1f)[26]. Within this cluster, we then analysed the expression of a larger number of genes associated with different cell death processes such as *Bcl2*-, *Bmp*- and *Fgf*-associated signalling and senescence[27]. Our data revealed no significant relative transcriptional differences in pro- or anti-apoptotic factors for the cluster 3 RA-Id between species (Fig. 1g and Extended Data Fig. 2). Interestingly, genes known to be distinctively expressed in the interdigital tissue of bat wings, including the anti-apoptotic *Grem1*[13], also did not show a difference in relative expression, suggesting expression in a different cluster.

To further investigate the presence, intensity and distribution of apoptosis, we stained bat limbs with LysoTracker, a marker of lysosomal activity that correlates with cell death[28]. The differential digit separation in bat limbs was used as an internal control: in bat HLs all digits separate completely, whereas in the FLs only the first digit separates from the second. Digits II–V, in contrast, do not separate in the wing, forming the chiropatagium. We found positive staining in all interdigital zones of bat FLs, with minor differences to interdigit I–II. Likewise, staining in the HL interdigit tissue was similar in intensity and distribution (Fig. 1h and Extended Data Fig. 2). In addition, we confirmed that cell death in bat wings occurs via an apoptotic process activated by the caspase cascade, as indicated by the positive staining for cleaved caspase-3 protein in a similar distribution as that described for LysoTracker staining (Fig. 1h and Extended Data Fig. 2).

In summary, our analysis revealed that the cell composition between mouse and bat limbs is highly conserved. Furthermore, cell death, as shown by the qualitative assays used here, is present in all interdigital tissues in the bats regardless of whether the digits get separated or not. However, it appears more intense between digits I–II of the FLs and HLs than in the other digits. Although it is difficult to compare between species, our results show that interdigital apoptosis is a feature of both bats and mice.

### The developmental origin of the bat chiropatagium
As cell death occurs similarly in both bat and mouse cluster 3 RA-Id, and spatially in both bat FLs and HLs, its inhibition is unlikely to account for the persistence of interdigital tissue. To identify the cells that persist and form the chiropatagium, we independently clustered the mouse and bat datasets and compared them with the integrated results. The clusters showed a good correspondence, with a high correlation of gene expression between species (Fig. 2a,b and Extended Data Fig. 3), suggesting that the chiropatagium is not associated to the emergence of a novel cell cluster in the bat wing.

To trace the molecular and cellular nature of the chiropatagium, we performed scRNA-seq from micro-dissected bat interdigital tissues at a later stage (CS18, equivalent to E14.5 in mice; Fig. 2c). We annotated the chiropatagium-LPM-derived populations by label transfer using the bat FL LPM data as a reference[29,30]. This revealed that the chiropatagium is primarily composed of three different populations of fibroblast cells, with transcriptional correspondence to clusters 7 FbIr, 8 FbA and 10 FbI1 (Fig. 2d). Differential expression analyses against the whole FL LPM dataset showed that the chiropatagium features high expression of *MEIS2*, *COL3A1*, *AKAP12* and *GREM1*, among others (Fig. 2e). Notably, the cluster 3 RA-Id was minimally represented in the chiropatagium (~1%; Fig. 2d), which is consistent with the results

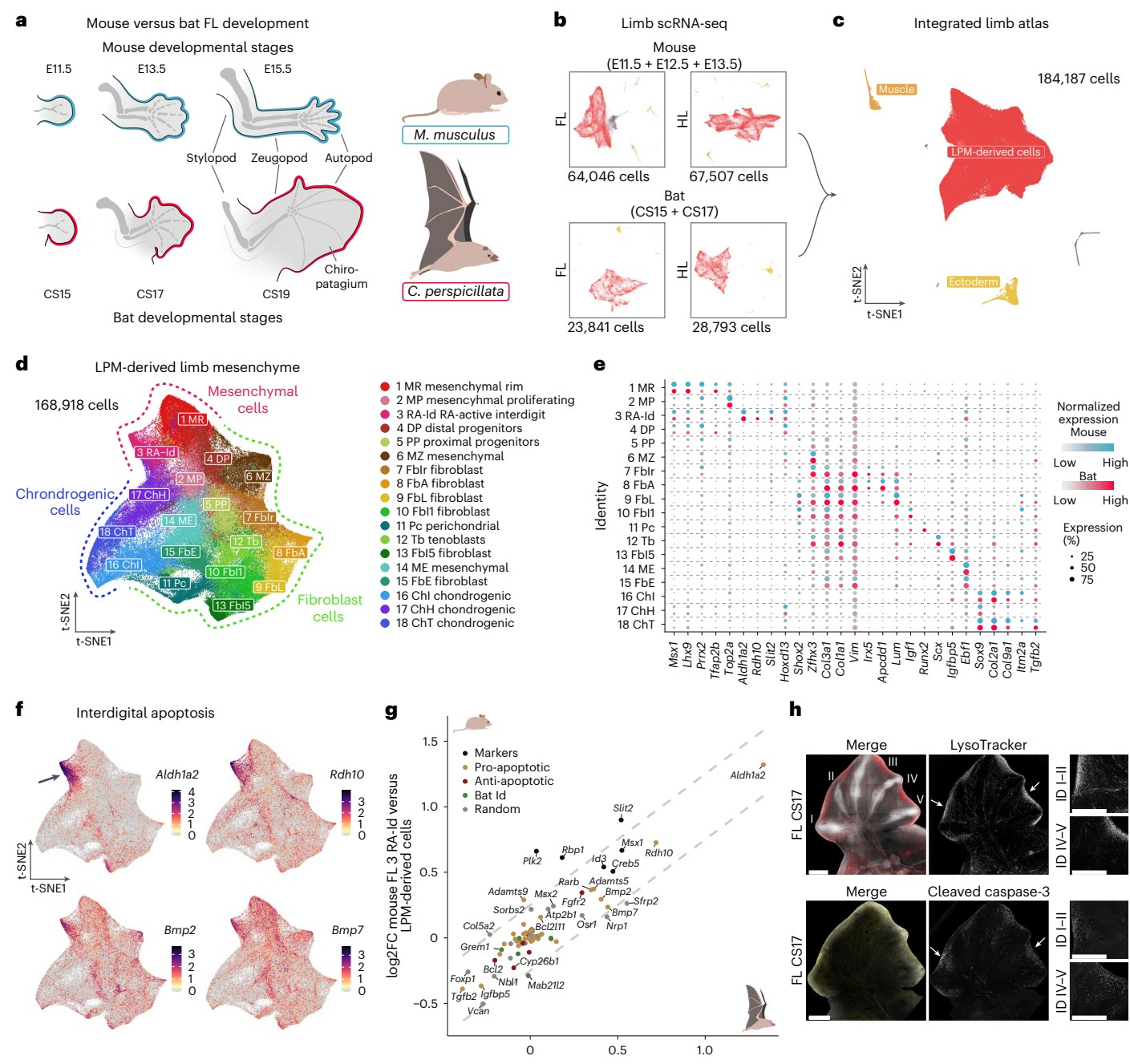

**Fig. 1 | Developmental cell states and interdigital apoptosis are conserved in mouse and bat limbs. a**, Scheme of key embryonic stages of mouse (blue) and bat (red) limb development. **b**, t-SNE plots of mouse and bat FL and HL single-cell datasets. The main cell populations are highlighted (red: LPM-derived cells; orange: muscle cells; yellow: ectodermal cells). **c**, t-SNE plot of integrated interspecies limb atlas. Main cell populations are highlighted as in **b. d**, t-SNE plot of LPM-derived cells with cluster annotations. Main developmental lineages are highlighted (red: mesenchymal; green: fibroblasts; blue: chondrogenic). **e**, Dot plot showing marker gene expression used for cluster annotation (Supplementary Data 1). Colour intensity indicates expression level (blue: mouse; red: bat); dot size indicates the percentage of cells expressing each gene. **f**, t-SNE plots of the integrated data showing the expression of central components of RA metabolism and BMP signalling involved in interdigital cell death. The arrow indicates the

interdigital cell population 3 RA-Id. **g**, Correlation of pro- (yellow) and anti-apoptotic (red) genes in the 3 RA-Id cell population of mouse and bat. Includes marker genes of this population (black) and genes previously reported to be expressed in bat interdigits (green). Shown is the log2FC of gene expression between the cluster 3 RA-Id versus the rest of the FL cells per species. A set of random genes was included as control. Dashed lines represent a difference of 0.25 and −0.25 of the log2FCs. **h**, LysoTracker staining (upper) and immunostaining against cleaved caspase-3 (lower) of bat FL stage CS17 with magnification of interdigital regions (ID; arrows) between digits I and II (which later lack interdigital membrane) and IV and V (later connected by chiropatagium) shown on the right. The arrows indicate the magnified regions. Merged images show DAPI (white) and LysoTracker (red) or cleaved caspase-3 (yellow) signal. $n = 2$. Scale bars, 500 μm.

of the apoptosis staining (Fig. 1h). Thus, the cluster 3 RA-Id can be ruled out as the cellular source of the chiropatagium.

To further elucidate the origin of chiropatagium cells, we inferred developmental trajectories in mouse and bat distal LPM clusters,

focusing on non-skeletal cells expressing *Hoxd13*, a bona fide marker of the autopodial lineage[31–33]. Using the RNA velocity tool scVelo, as well as the pseudotime tool Slingshot, we identified independent trajectories that share the same origin and are defined by differential

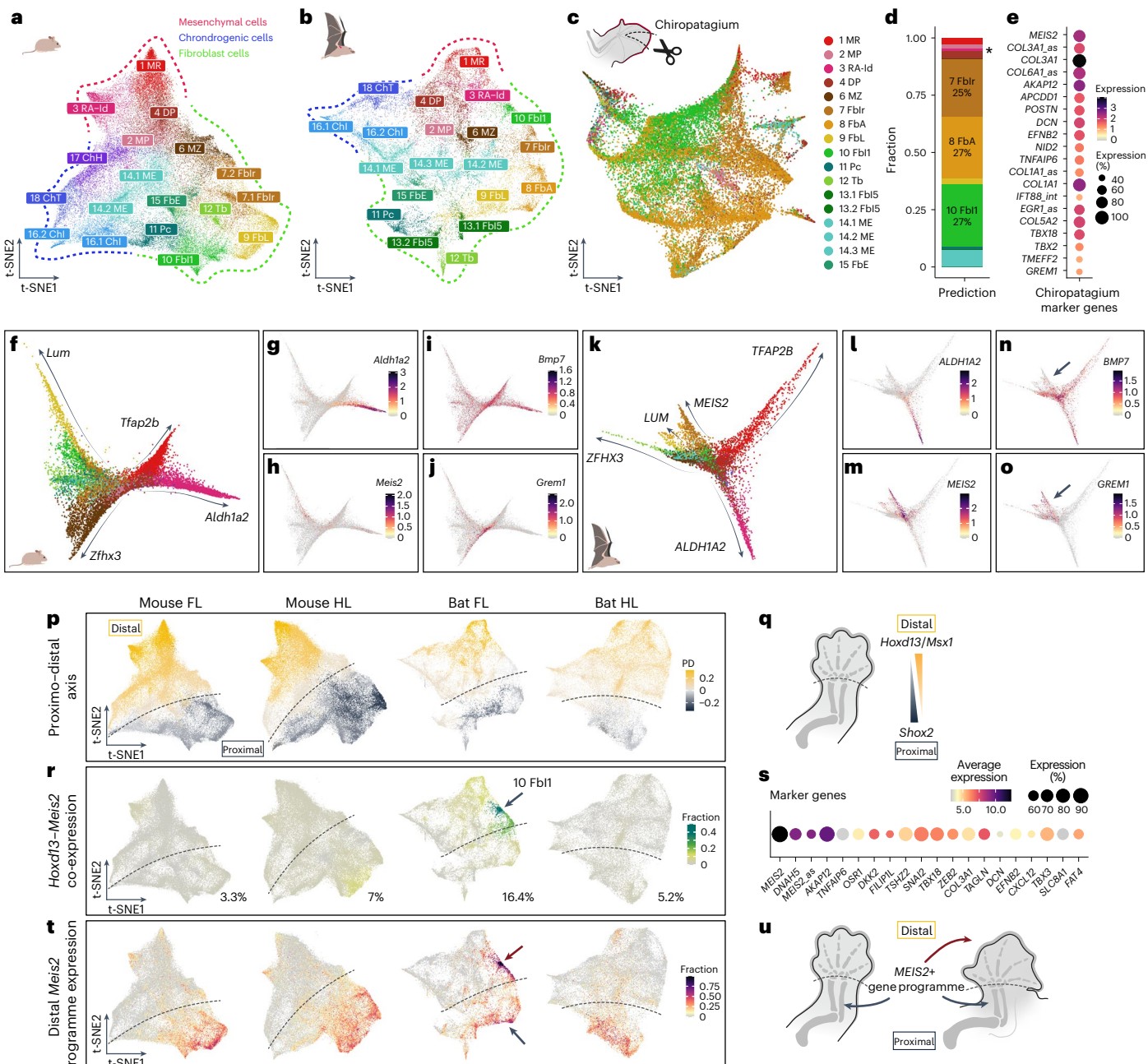

**Fig. 2 | The chiropatagium is composed of fibroblasts expressing MEIS2 following a unique developmental trajectory in the bat FL autopod.**
**a,b**, Mouse (**a**) and bat (**b**) t-SNE plots of individual FL clustering. Main developmental lineages are highlighted (red: mesenchymal; green: fibroblasts; blue: chondrogenic). Cluster labels and colours are derived from Fig. 1d.
**c**, t-SNE plot of chiropatagium cells at stage CS18. Colours and labels by transcriptional correspondence to bat FL LPM-derived cells. **d**, Quantification of correspondence between chiropatagium and LPM-derived cells. Asterisk indicates the 3 RA-Id cells. **e**, Marker genes of chiropatagium cells, compared against bat FL LPM-derived cells. **f**, Differentiation trajectories of *Hoxd13*-positive, non-chondrogenic cells of the mouse FL, derived from RNA velocity and pseudotime analyses indicated by arrows. Trajectories are annotated based on increasing expression of marker genes. **g–j**, Expression of *Aldh1a2* (**g**), *Meis2* (**h**), *Bmp7* (**i**) and *Grem1* (**j**) in mouse FL trajectories. **k**, Same as **f**, for bat cells.

**l–o**, Expression of *ALDH1A2* (**l**), *MEIS2* (**m**), *BMP7* (**n**) and *GREM1* (**o**) in bat FL trajectories. The arrow in **n** and **o** indicates a unique *MEIS2*-positive trajectory identified in the bat FL. **p**, Assignment of a proximal (dark blue) or distal (yellow) identity to each cell of mouse and bat FL and HL. PD is the difference of the expression score of the distal genes (*Hoxd13*, *Msx1*), minus the expression score of the proximal gene (*Shox2*). Dashed lines outline the proximal–distal regions. **q**, Scheme of assignment in **p**. **r**, Fraction of cells co-expressing *Hoxd13* and *Meis2* per cluster. Cluster 10 Fbl1 is highlighted with an arrow. **s**, Marker genes of bat cluster 10 Fbl1 against the rest of the LPM-derived cells. Ordered by adjusted *P* value (not shown, <0.01). Differential expression tested using a Wilcoxon rank sum test implemented in Seurat. **t**, Co-expression score of the genes in **s**. The distal and proximal cells highly expressing this programme are indicated with arrows. **u**, Schematic of *MEIS2*-positive cell expression in proximal fibroblasts of mice and bats, with distal expression specific to bat wings.

increased gene expression (Fig. 2f–o and Extended Data Fig. 4). For example, the cluster 3 RA-Id forms a trajectory with increasing *Aldh1a2* expression (Fig. 2g,l). Moreover, in bat FL we identified an independent trajectory of fibroblasts marked by the expression of the TF *MEIS*.

This trajectory was neither detected in mice nor in bat HLs, suggesting a unique developmental specification for chiropatagium cells (Fig. 2h,m and Extended Data Fig. 4). Moreover, this *MEIS2*+ trajectory also showed high expression of *GREM1*. Both of these have been shown

to be specifically expressed in the interdigital tissue of bat wings, as well as other interdigital markers like *Aldh1a2*[13,14]. Thus, confirming that this cell population shares this space with the cluster 3 RA-Id in bats. Overall, these analyses further show that the chiropatagium develops independently from the interdigital cluster 3 RA-Id. In contrast, this tissue is primarily composed of fibroblast cells expressing *MEIS2*.

MEIS2 is a TF that defines proximal identity at early limb stages[34,35]. To explore its distal role during bat autopod morphogenesis, we first defined the proximo–distal identity for each cell and cluster across all non-integrated datasets. Specifically, we calculated the gene expression ratio between distal (autopod) and proximal markers (*Hoxd13* + *Msx1* versus *Shox2*). Most clusters could be clearly identified as either proximal or distal (Fig. 2p,q). *Meis2* was among the marker genes characterizing the proximal non-skeletal cells in all our samples (Extended Data Figs. 3 and 4). We then quantified the fraction of *Meis2*-positive cells in the distal region by calculating the co-expression of *Hoxd13* and *Meis2*. This analysis revealed that the highest number of cells expressing both factors and highest co-expression levels are found in the bat FLs (Fig. 2r, green colour, 16.4%). This co-expression pattern specifically highlighted fibroblast cluster 10 (arrow in Fig. 2r), followed by cluster 7, each one constituting ~1/3 of chiropatagium cells at later stages (Fig. 2d). We therefore focused on cluster 10 and, by comparing it against the remaining LPM cells, identified 20 marker genes including the TFs *OSR1*, *TBX18* and *TBX3* (Fig. 2s). Given the unusual nature of this cluster, with many cells highly co-expressing distal and proximal markers, we explored the expression of these 20 genes across all samples. Intriguingly, this gene set was found co-expressed at high levels in the proximal fibroblasts (mostly clusters 8 and 9) of FLs and HLs of both species, while its distal co-expression was unique to the bat FLs (Fig. 2t). Similar results were found for the marker genes of cluster 7 (Extended Data Fig. 5). Thus, the chiropatagium consists of fibroblasts that do not derive from the cells of cluster 3 RA-Id. Rather, chiropatagium cells display their own differentiation trajectory characterized by a specific set of genes that includes *MEIS2*, a TF expressed prominently in the proximal limb (Fig. 2u).

### Repurposing of a proximal gene programme in the distal bat wing

Our analyses identified a fibroblast cluster that is unique to the distal bat FLs, yet expresses a gene set that is also present in proximal fibroblast cells of mouse and bat limbs. To determine the degree of transcriptional similarity among these clusters, we performed differential gene expression analyses in bat FLs comparing the proximal (8) and distal (10) fibroblasts against the rest of the LPM cells. We found 223 overexpressed genes, 65% of them (144) displaying high relative expression in both proximal and distal clusters (Fig. 3a). Nevertheless, a subset of genes was specific to distal or proximal clusters (25 and 64, respectively; Fig. 3b). Interestingly, 34 of the shared genes were also highly expressed in mouse proximal fibroblasts, suggesting an evolutionary conserved function for this gene set in limb fibroblasts (triangular points in Fig. 3a,c). Thus, the distal *MEIS2*-positive cluster 10 is characterized by a gene programme that shows substantial transcriptional overlap with a proximal cluster. Gene ontology (GO) enrichment analysis for the shared genes revealed distinct functions, including mesenchymal proliferation, extracellular matrix (ECM) organization and ameboidal-type cell migration (Fig. 3d). These processes are not only indicative of fibroblast identity[36], but also represent essential components of interdigital remodelling[37] and may be highly relevant in the context of wing development. Similar results were found for the gene programme related to cluster 7 (Extended Data Fig. 6). To better understand the relationship and hierarchies of the genes in the programme, we performed a gene regulatory network analysis using SCENIC for each cell cluster. This analysis placed MEIS2 in the regulon with the highest regulon specificity score (RSS) within the bat cluster 10 (RSS > 0.23; Extended Data Fig. 7). Furthermore, MEIS2 also appeared as a direct regulator for numerous genes, including several that belong to the shared proximo–distal programme (Fig. 3e and Extended Data Fig. 7).

To further elucidate how this gene programme is regulated, we generated bulk transcriptomic and epigenomic datasets from distal limbs by physically dissecting them at the level of the wrist (mouse E15.5 and bat CS19 stages; Extended Data Fig. 8). Differential expression analyses between distal FLs and distal HLs showed only small differences for mice, while bat distal FLs showed a higher number of differentially expressed genes (DEGs) compared to HLs. Among the most upregulated genes in bat FLs we found the TFs *MEIS2*, *HOXD9*, *HOXD10*, *HOXA2* and *TBX3*, genes known to be early proximal markers and patterning factors[38,39] (Extended Data Fig. 8). Differential enrichment analyses for active epigenomic regions (marked by accessible chromatin regions detected using an assay for transposase-accessible chromatin (ATAC-seq) and chromatin immunoprecipitation (ChIP-seq) with an antibody against the H3K27ac) revealed a high number of regions specific to the distal bat FL, enriched in TF binding sites for RFX, ATF, GATA, ATG and, notably, MEIS (Fig. 3f,g and Extended Data Fig. 8). As several analyses suggested that MEIS2 plays a critical role in chiropatagium development, we profiled its chromatin binding in distal bat limbs using a dual antibody ChIP-seq assay[34]. We found 4,212 MEIS-binding peaks in active accessible bat genomic regions (ATAC + H3K27ac peaks), of which only 244 correspond to gene promoters. Only 27% (1,142) of the MEIS-binding peaks found in conserved mouse/bat genomic regions (4,259) also display signatures of enhancer activity (H3K27Ac enrichment) in the mouse distal FL. Based on these data we conclude that bat distal MEIS2 activity appears to associate with, and thus regulate, a set of genes/enhancers that is different from those in the mouse. As with other TFs, MEIS seems to bind to several enhancer regions across large genomic distances[40]; therefore we summed up all MEIS-bound regions per regulatory domain, defined by genome-wide chromatin interaction maps (Hi-C) from developing bat limbs. We identified a subset of regulatory domains distinctly enriched with MEIS2 binding signal (Extended Data Fig. 8). By intersecting accessible H3K27ac- and MEIS2-binding enriched domains with genes from the distal/proximal fibroblast gene programme, we narrowed down the list of candidate genes potentially regulated by MEIS2 to 71 (Fig. 3h). The top 20 genes displaying the highest overall MEIS binding signal in their regulatory domains included genes from the fibroblast gene programme, like ECM components and TFs such as *TBX3* and *TBX18* (Fig. 3i, ranked from left to right according to the acetylation coverage). The striking pattern of chromatin activity profiles (H3K27ac and MEIS2 binding) being constrained within regulatory domains is exemplified for the *TBX3* domain (Fig. 3j and *TBX2* in Extended Data Fig. 8). In addition, we compared MEIS2 binding in the distal bat limb with ChIP-seq data from early (E10.5) mouse embryonic limbs[34], where MEIS1/2 is known to have a crucial role in limb patterning. The limited overlap in bound gene promoters (21 regions) suggests that MEIS2 has a distinct regulatory role and differential genome accessibility at both stages (Extended Data Fig. 8). In summary, we identify MEIS2 as a critical TF regulating chiropatagium development, through the pervasive binding at the chromatin landscape of its associated gene programme.

### Distalization of *MEIS2* and *TBX3* induces wing-related phenotypes

Our previous analyses positioned *MEIS2* and *TBX3* as key regulators of the gene programme associated with chiropatagium development. To investigate their effects on limb developmental cell states, we induced the distal limb expression of these two TFs in transgenic mice. The sequences of these TFs in both species result in highly similar proteins (Extended Data Fig. 9). Constructs were generated in which the bat coding sequences of *MEIS2* and *TBX3* were expressed under the control of a previously characterized *Bmp2* enhancer[41]. This enhancer has specific activity in the distal non-skeletal mesenchymal and interdigital part of the limb bud (Fig. 4a). This precise spatio-temporal activity allowed

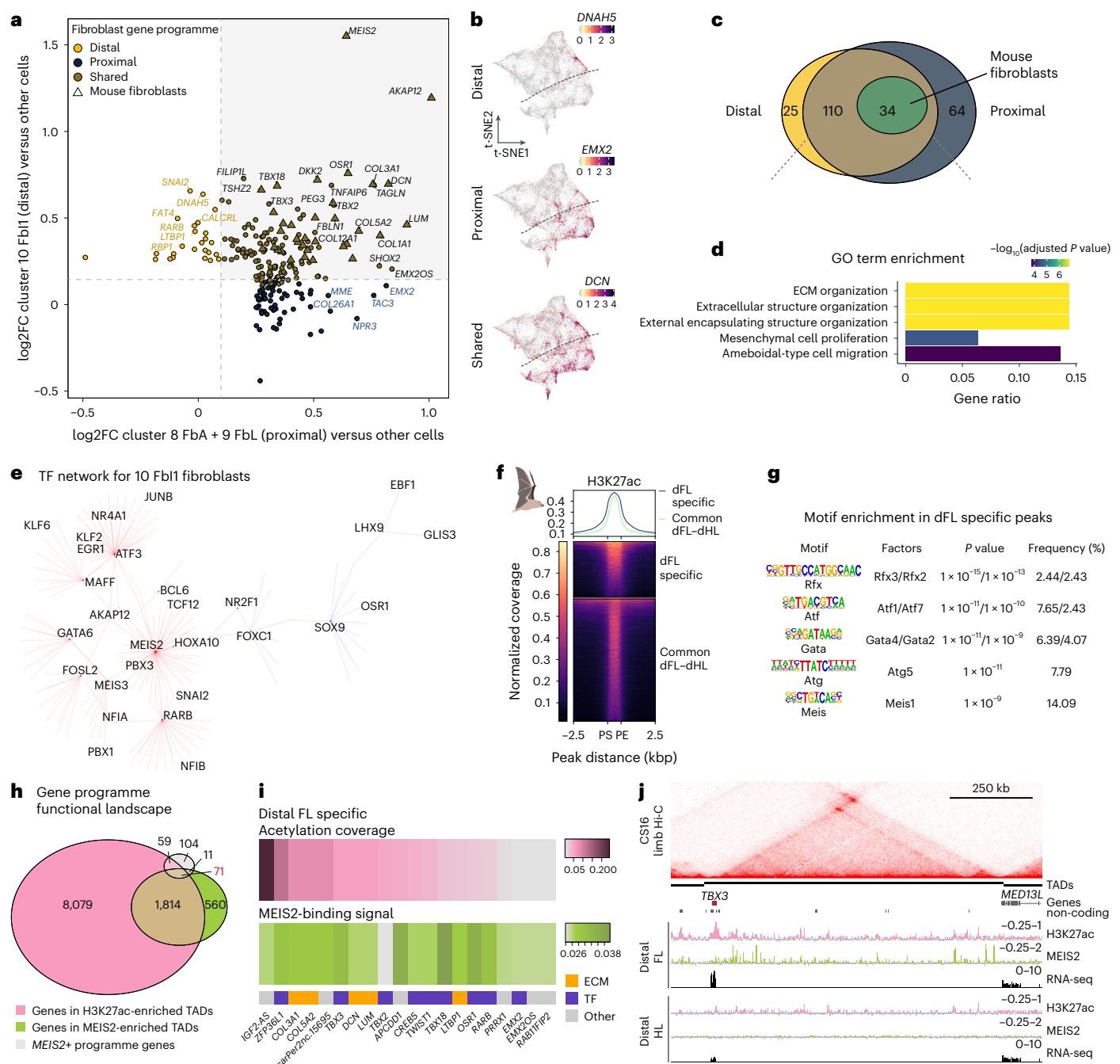

**Fig. 3 | Proximo–distal dissection of mouse and bat limbs reveals repurposing of a proximal gene programme in the distal bat FL. a**, Correlation between DEGs from distal (cluster 10) and proximal (clusters 8 and 9) *MEIS2*-positive clusters in the bat FL identified in Fig. 2. Shown is the log2FC of expression of the respective cluster versus non-fibroblast LPM-derived cells (Supplementary Data 2). Genes shared with mouse fibroblasts are depicted as triangles. The grey-shaded region is the area where the genes have a log2FC > 0.1 in both comparisons. **b**, Representative t-SNE plots of genes expressed in the distal, proximal or both clusters of the bat FL. **c**, Venn diagram showing the overlap (brown) between genes enriched in the proximal (dark blue) and distal (yellow) cells as well as the genes shared with mouse fibroblasts (green). **d**, GO term enrichment analysis of the shared genes from **c**. Shown are the top five enriched terms (Supplementary Data 3). Over-representation analysis implemented in ClusterProfiler (Methods). **e**, SCENIC TF network analysis for genes enriched in cluster 10 FbI1. Red and blue

lines represent positive and negative regulatory connections, respectively. **f**, Tornado plot showing H3K27ac peaks specific to the distal FL (dFL) as well as common peaks of dFL and distal HL (dHL). Shown are regions from peak start (PS) to peak end (PE). **g**, Motif enrichment in distal FL-specific H3K27ac peaks. Shown are the top five binding motifs per TF family. De novo motif enrichment is estimated using the cumulative hypergeometric distribution. **h**, Venn diagram showing the overlap between genes in H3K27ac-enriched and MEIS2-binding enriched topologically associated domains (TADs), as well as genes from the fibroblast gene programme from **a**. **i**, Heatmaps showing the portion of each TAD covered by H3K27ac peaks, and the mean signal per TAD of MEIS binding. Shown are the top 20 genes by MEIS binding signal (Supplementary Data 4). **j**, Bat TBX3 locus with Hi-C from CS16 FLs on top, TAD calling below. The input-subtracted H3K27ac and MEIS2 ChIP-seq tracks are depicted in pink and green, respectively. RNA-seq tracks are shown in black.

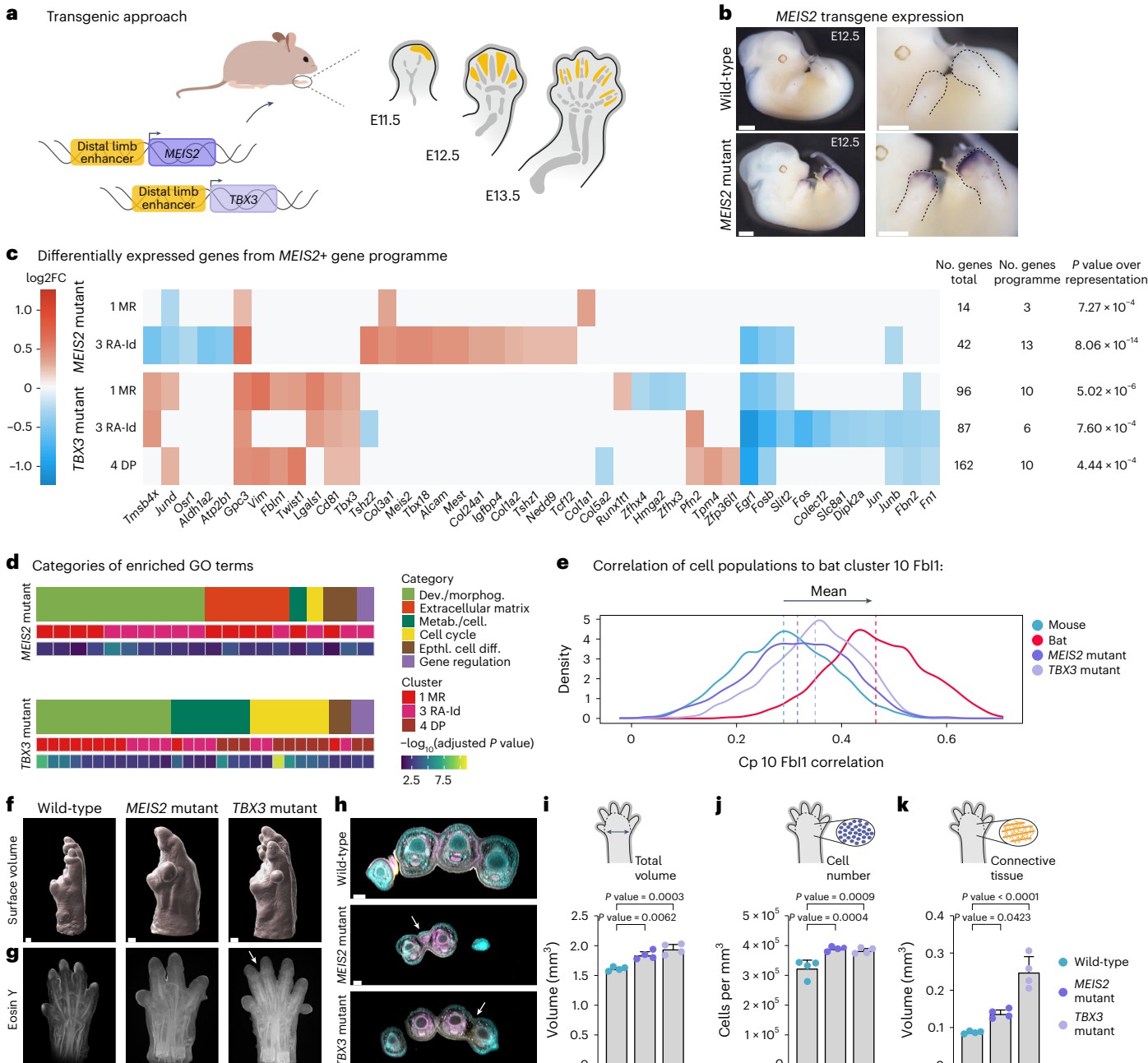

**Fig. 4 | Distal expression of *MEIS2* and *TBX3* in transgenic mouse limbs induces cellular and morphological features related to the chiropatagium.** **a**, Transgenics experiments. Bat *MEIS2* and *TBX3* coding sequences were expressed in mouse limbs using a *Bmp2* enhancer[41] with activity in distal and interdigital mesenchyme (yellow). **b**, WISH from E12.5 wild-type and mutant embryos showing distal activity of transgene constructs (*n* = 2). Scale bar, 1 cm. **c**, Heatmaps showing DEGs from the chiropatagium gene programme in affected single-cell clusters of mouse mutant limbs at E12.5 (Supplementary Data 5 and 6). Non-significant and differences below 0.25 log2FC set to 0. The number of upregulated genes from the fibroblast gene programme and of over-representation *P* values are on the right. Differential expression tested using a likelihood-ratio test on a zero-inflated regression implemented in MAST (Methods). **d**, Proportion of GO term categories (biological functions) upregulated in mutant mice. From the top ten GO terms of the affected cell clusters. Individual GO terms are in Extended Data Fig. 10 and Supplementary Data 7 and 8. Dev./morphog., development and morphogenesis; Metab./cell.,

metabolism and cellular processes; Epthl. cell diff., epithelial cell differentiation. **e**, Correlation of affected mutant cells to the bat cluster 10 Fbl1, based on the expression of genes from the gene programme. Depicted is the density of the correlation of all cells in the affected clusters, and corresponding clusters in mouse wild-type and bat FLs. Dashed line is the mean. **f,g**, Three-dimensional imaging of mouse wild-type and mutant limbs at E15.5 (*n* = 4). Surface representation (**f**) and an Eosin Y staining cross-section (**g**) with an arrow highlighting syndactyly. Scale bar, 200 μm. Magenta, Eosin Y; cyan, nuclei; yellow, autofluorescence. **h**, Cross-sections with arrows indicating tissue between the digits. Scale bar, 100 μm. **i–k**, Quantification of autopod surface volume (**i**), cell number (**j**) and connective tissue volume (**k**) in wild-type and mutant limbs. *n* = 4. Error bars show the standard deviation. Numbers are *P* values of the differences of the mean calculated using a Dunnett test following a one-way analysis of variance. When comparing the wild-type with the *MEIS2* and *TBX3* mutants, the exact *P* values are 0.0062 and 0.003 for total volume, 0.0004 and 0.0009 for cell number, and 0.0423 and 0.0001 for connective tissue volume.

the expression of these factors without inducing detrimental effects in other tissues. In situ hybridization as well as bulk transcriptomic analysis of mutant limbs at E12.5 validated specific expression of the transgenes in distal mouse limbs (Fig. 4b and Extended Data Fig. 9).

We evaluated the impact on gene expression at cellular resolution by performing scRNA-seq on the mutant limbs at E12.5. Focusing on distal clusters, we isolated *Hoxd13*-positive cells and integrated them with corresponding data from our reference mouse atlas. Using this approach, we performed a differential expression analysis on the clusters where *MEIS2* or *TBX3* were differentially expressed in the mutant samples (Fig. 4c). We found that the genes of the chiropatagium gene programme were significantly over-represented within the DEGs (Fig. 4c). Interestingly, we see a downregulation of *Aldh1a2* in the cells of cluster 3 RA-Id, where *Meis2* is ectopically expressed. GO enrichment analysis showed that the upregulated genes are involved in ECM production and proliferation processes, functions also characteristic of the identified gene programme (Fig. 4d and Extended Data Fig. 10). Moreover, we compared the transcriptomic correlation of mouse wild-type and mutant cells to the mean gene expression of fibroblast cluster 10 from bat FLs. This revealed that mouse mutant cells exhibit higher similarity to these bat cells (Fig. 4e). These results highlight the ability of these two TFs, *MEIS2* and *TBX3*, to partially induce the specific gene programme of the chiropatagium.

To evaluate the phenotypic consequences of ectopic distal *MEIS2* and *TBX3* expression, we performed three-dimensional (3D) imaging of mutant and wild-type control limbs at a later developmental stage (E15.5). We marked the nuclei with DRAQ5 and used Eosin Y as a proxy to quantify ECM content (Fig. 4f,g). Both mutants showed a visible increase in the surface volume of the limbs. In addition, all *TBX3* mutants displayed fusion of at least two digits (Fig. 4h and Extended Data Fig. 10; *n* = 4). Transversal sections of these limbs confirmed the retention of the tissue between digits II and III, resembling cutaneous syndactyly in both mutants (Fig. 4h). Quantification analyses of these images revealed a significant increase in the overall autopod volume, cell number and connective tissue content in both mutants (Fig. 4i–k). These results indicate that the expression of *MEIS2* and *TBX3* in the distal and interdigital mesenchyme can recapitulate essential aspects of bat wing development. This includes increased proliferation and matrix production, resulting in retention of interdigital tissue with consecutive fusion of digits. Overall, these analyses support that the distal activation of a gene programme mediated by *MEIS2* and *TBX3* plays a role in bat chiropatagium formation.

## Discussion

This study aims to elucidate the molecular basis and cellular origin of the interdigital wing membrane of bats, the chiropatagium. Previous studies have attempted to identify the genes and mechanisms behind this fascinating evolutionary adaptation. Candidate gene approaches, for instance, suggested an involvement of pro-apoptotic factors such as *BMPs*[42] and their antagonist *GREM1*[13] or a second wave of *SHH* expression in the interdigital space[43]. More systematic approaches using transcriptional profiling and the integration of regulatory data identified genes of the *HoxD* cluster as well as components of canonical Wnt signalling[17]. Yet, these genome-wide studies lacked cellular resolution and therefore much remains elusive. Here, by using scRNA-seq, we were able to assign expression patterns to specific cell populations thereby disentangling previous contradictory observations. Cells expressing RA/BMP pro-apoptotic factors in bats are equivalent to the cluster 3 RA-Id observed in mice, where interdigital regression takes place. In contrast, distal bat fibroblasts express the BMP antagonist *GREM1* (Fig. 2o) previously shown to be expressed in the interdigits of the wing, but not the HLs[13]. Even though these cells are in the same interdigital space as the cluster 3 RA-Id cells, they originate from a distinct developmental trajectory eventually constituting the major component of the chiropatagium. While we do not explore the developmental origin

of this cell population, their presence and persistence in an otherwise disappearing tissue might be explained by their already differentiated state. Experimental manipulations of developing chicken HLs show that before an apoptotic fate, the interdigital mesenchyme is naive with full differentiation potential[44–47], suggesting that apoptosis arises due to the lack of differentiation or further survival signalling[48]. It is, however, possible that suppression of RA/BMP signalling by factors such as *GREM1* serves as an additional factor protecting *MEIS2*+ fibroblasts from apoptosis. Furthermore, as shown by our transgenic experiments, ectopic expression of *MEIS2* results in a downregulation of *Aldh1a2*, indicating that *Meis2* itself may have an anti-apoptotic effect.

Nonetheless, besides apoptosis, other mechanisms including epidermal cell migration[37] and the remodelling of ECM components are involved in interdigital tissue regression[49]. This, together with our results, indicates that apoptosis is not sufficient for sculpting the digits in mammals. Indeed, further analysis of the fibroblast gene programme identified an enrichment of genes associated with these processes, that is, ECM organization, cell migration and proliferation. Alterations in the balance between cell death and proliferation and migration are likely to change the interdigital cell composition and can result in the retention of interdigital tissue (syndactyly)[27,37,50].

A major challenge in comparative single-cell analyses lies in data integration, which risks overcorrection and the consequent masking of biological variation[51]. This was also of concern during our integration of bat and mouse data, where the interdigital distal fibroblasts forming the bat chiropatagium clustered together with other fibroblasts from both species. However, our independent analyses of the bat limb cells revealed conserved composition. Moreover, various analyses, including micro-dissected chiropatagium scRNA-seq, trajectory analyses and epigenomic profiling, revealed that such clustering was not artefactual. Instead, it reflected the activation of similar transcriptional programmes through a distinct regulatory repertoire, ultimately driving a unique bat forewing-specific cell differentiation trajectory. It is well documented that during evolution, the same set of genes is often re-used[52]. For example, the formation of lateral patagia enabling gliding has independently appeared multiple times in marsupials through convergent evolution, where the upstream factor *Emx2* is activated by distinct regulatory elements in different glider species[53].

Here we identify a gene programme that has been repurposed through evolution, where two TFs, MEIS2 and TBX3, appear among the primary regulators. Specifically, we show that MEIS2 is a potential direct activator of many other TFs in bat wings, regulating other downstream genes. Both factors were previously described in different bat species (*Miniopterus natalensis*, *Miniopterus schreibersii*) as expressed in the distal bat FL[15,16], indicating a conserved function in wing development. *Meis2* has also been previously reported to be expressed in distal E14.5 mouse limbs, based on in situ hybridization signal[14]. However, our quantification based on scRNA-seq demonstrates that the expression levels are low and are present in markedly fewer cells compared with the bat FL (Fig. 2r). In contrast, *Meis1* and *Meis2* (Meis) homeobox TFs are well known to be robustly expressed early (<E11.5) in the proximal part of the limb, where they determine the identity of stylopod and zeugopod versus autopod[54]. Accordingly, mutating *Meis* TFs result in limb shortening due to altering the proximo–distal segmental borders[34]. The specification of proximal identity by *Meis* genes is an ancient function conserved across the vertebrate lineage, including mammals[34], birds[54,55] and amphibians[56]. Interestingly, in *Drosophila*, the *Meis* homologue *hth* is also required for proximal leg development[54]. Likewise, *Tbx3* is a gene expressed in the proximal limb mesenchyme and plays a crucial role during limb patterning in establishing anterior–posterior boundaries[57]. The importance of *MEIS2* and *TBX3* in chiropatagium development is supported by our studies in transgenic mice. The gene expression changes observed in mutant limbs, together with the alterations in morphology, cell number and matrix production, reflect key features associated with

the gene programme of chiropatagium cells (Fig. 3). Thus, the ectopic expression of *MEIS2* or *TBX3* in interdigital distal cells induces a gene programme that partially resembles that observed in bats and leads to tissue retention. The recapitulation of only certain aspects of the wing phenotypes is expected, as we are manipulating only one gene at a time from an entire gene programme. Moreover, interspecies approaches have inherent limitations, as the ectopic expression of these genes occurs in a different molecular and cellular context. It is likely that the expression pattern of *MEIS2* observed in bats requires regulatory changes rendering *MEIS2* susceptible to specific FL autopod signals, such as an FL-specific *Hox* code[58,59]. This may encompass the observed activation of 3′ anterior *Hox* paralogues like *HOXA1/2*.

Phenotypic evolutionary innovations can, in principle, arise from gene duplications or losses. Yet, a genomic comparison of six bat reference genomes failed to reveal expansion or loss of any candidate gene that might play a role in limb development[60]. Alternatively, already existing genes can be newly recruited into regulatory gene networks (co-option)[61,62], or regulatory changes can modify gene expression within existing ones[53,63,64]. Instead, our data point towards a high degree of similarity in gene expression between species, suggesting the re-use of a transcriptional programme already existent in the limb but at a different anatomical position. It is probable that the re-use of this gene programme occurs within a markedly disparate epigenetic landscape, thereby activating slightly disparate and novel gene sets. A similar scenario has recently been reported for skeletogenic cells found in different parts of the body[65]. Chondrocytes that originate from different germ layers use distinct sets of regulatory elements for activation of similar gene programmes. Like the chiropatagium cells, which are equivalent at the transcriptional level to the proximal fibroblasts but diverge at the gene regulatory level. Thus, even a change as drastic as the development of a wing from individual digits can apparently be achieved by relatively small changes and the repurposing of already existing and active pathways. Following the principle of parsimony, evolution constructed novelty by making minimal modifications to already existing elements.

## Methods

### Animal samples

**Mice.** Wild-type mouse embryonic tissues were derived from crossings of CD1 × CD1 or C57BL/6J × 129. Transgenic embryos were generated by tetraploid aggregation[66]. Female mice of CD1 genetic background were used as foster mothers. Mice were kept in a controlled environment (12 h light and 12 h dark cycle, temperature of 20–22.2 °C, humidity of 30–50%) and water, food and bedding were changed regularly.

All animal experiments and procedures were conducted as approved by LAGeSo Berlin under the following licence numbers: ZHV120, G0176/19-MaS1_Z, G0243/18-SAld1_G and G0098/23-SAld1_G.

**Bats.** Bat samples (*Carollia perspicillata*) were obtained from a captive population maintained at the Papiliorama zoo in Kerzers, Switzerland. To control population growth, some individuals were occasionally culled by cervical dislocation performed by trained personnel, following general guidelines for animal handling and in vivo research[67]. If pregnant females were present among the culled individuals, embryos were dissected and preserved for different molecular procedures. Females in late pregnancy were not culled for ethical reasons. In addition, bat samples from *C. perspicillata* were obtained from a breeding colony housed at the Institute for Cell Biology and Neuroscience, at the Goethe University in Frankfurt am Main (keeping permit authorized by the RP Darmstadt). Samples collected from the Frankfurt colony originated from bats that were euthanized for collecting brain tissue without any further experimental manipulation (following § 4 Abs. 3 of the German TierSchG). In female bats, after euthanizing, we additionally checked for possible pregnancies (undetectable from the outside) and embryos were dissected whenever present.

## Genome annotation

To generate an annotation of *C. perspicillata*, we collected transcriptomic data from long- (IsoSeq) and short-RNA reads, mapped those to a chromosome-scale assembly, and integrated gene predictions using human (hg38), mouse (mm10) and another phyllostomid bat (*Phyllostomus discolor*) as reference annotations. Briefly, IsoSeq data were first analysed as in ref. 60 to produce high-quality open reading frame predictions. Then, we implemented a strategy to classify and filter transcripts-based features such as known canonical splice sites, known non-canonical splice sites, novel canonical splice sites and novel non-canonical splices. A small set of transcripts with suboptimal features were not used as input for the gene annotation. For example, fusion transcripts (chimeras that include more than one gene), intra-priming (transcripts with more than 85% or at least 10 contiguous adenines within 20 bp upstream of the 3′ end), low coverage (transcripts supporting coding regions by less than 3 reads), reverse-transcriptase-switching predictions (an exon-skipping pattern due to a retrotranscription gap caused by secondary structures in expressed transcripts), nonsense-mediated decay (premature stop codons) and intron retention were features all identified as suboptimal. However, when possible, some of these transcripts were used to annotate untranslated regions (UTRs). Transcript features used for classification were identified using TAMA-GO[68]. Then, new TOGA predictions were generated using an updated version[69] (v. 8f09391; https://github.com/hillerlab/TOGA). We used as reference genomes human (hg38), mouse (mm10) and the pale spear-nosed bat (*Phyllostomus discolor*). Finally, additional RNA-seq data from tissues were generated and analysed, as described in ref. 60. Evidences (RNA-seq transcripts, reclassified IsoSeq transcripts, TOGA predictions and proteins) were integrated using EVM[70], and downstream steps to annotate non-overlapping UTRs, enrich the annotation with non-coding RNAs and assign gene names were performed as described in ref. 60.

The sensitive prediction of genes including UTRs led to typical artefacts where a gene name was assigned more than once. This sometimes caused a mislabelling of an orthologous gene compared to a reference genome annotation (here hg19). Additionally, in some instances (for example, *HOX* gene clusters) transcripts annotated with a unique coding sequence (CDS) were grouped artificially into a single gene based on shared UTR exons. To correct these issues conservatively we used sequence conservation to human (hg19) determined via a one-to-one comparison of both genomes via the alignment software LAST[71] with the following parameters: lastdb -uMAM8 -R11 -c; last-train–revsym–matsym --gapsym -E0.05 -C2; lastal -m10 -E0.05 -C2.

As a prerequisite we lifted human CDS regions to the *Carollia* genome. In cases where a CDS overlaps a conserved region, but a boundary was not conserved, the boundary was interpolated via its distance to the closest overlapping conserved region (approximate lift-over). Known fusion genes and lifted genes with unusually large intron size >30 kb were excluded from subsequent renaming or boundary adjustments.

A *Carollia* transcript was reassigned to a reference gene name if at least 50% of the original CDS boundaries matched the lifted coordinates of the reference annotation, and if other transcripts of the original gene share less CDS boundaries to another reference gene. Once transcripts and genes were renamed, transcripts extending beyond the boundaries of the orthologous reference gene were clipped at the 5′/3′ UTRs.

Finally, for genomic regions without any gene annotation we transferred exon annotations from hg19 to *Carollia* via approximate lift-over. While this procedure may detect mainly approximate or partial gene annotations, it allowed us to recover an additional ~500 genes (for example, *XIST*) otherwise excluded from analyses. The genome annotation resulted in 23,315 transcripts from 18,697 genes. To add long non-coding transcripts, StringTie[72] was used on the short-read RNA-seq data to obtain a transcriptome annotation, which was processed further with PLAR[73] to identify long non-coding RNAs. The ones

not overlapping the initial transcriptome were added to result in 20,421 additional transcripts from 16,141 additional genes.

For the comparative analysis of mouse genes, we used genome version mm39 (GCF_000001635.27) with annotation release 109. Only gene entries of type gene, exon, CDS, pseudogene, transcript, primary_transcript and RNA types (excluding guide_RNA) were processed further. Finally, gene models overlapping exons of known genes, or predicted transcripts where an alternative curated RefSeq-entry (ID starting with NM_ or NR_) existed were removed. Additionally, three fusion transcripts were removed.

Orthologue relationship was determined by a one-to-one comparison of the *Carollia* and mouse genomes via LAST (same parameter settings as for hg19). A mouse gene was defined as an orthologue of the *Carollia* gene with maximum of shared exon boundaries. In case of ambiguity, the gene with highest overlap was assigned. As a consequence, only one-to-one orthologue assignments were generated (Supplementary Data 11).

### scRNA-seq
**Single-cell isolation, methanol preservation and rehydration.** For single-cell gene expression analysis, mouse and bat embryonic limb tissues were dissected and dissociated with trypsin. Cells were filtered through a 40-μm Flowmi Cell Strainer (Merck, no. BAH136800040) and pelleted at 300 × $g$ at 4 °C for 5 min. Cells were resuspended in 1 volume 0.04% BSA/PBS and dehydrated by slowly adding 9 volumes of 100% methanol. Samples were stored at −80 °C until library preparation.

For rehydration, dehydrated cells were centrifuged at 1,000 × $g$ at 4 °C for 10 min and washed twice in 1 ml of rehydration buffer (1% BSA, 0.4 U μl$^{-1}$ Ambion RNse Inhibitor (Invitrogen, no. AM1682) and 0.2 U μl$^{-1}$ SUPERaseIn RNase Inhibitor (Invitrogen, no. AM2696) in 1× DPBS). After the second wash, cells were resuspended in rehydration buffer, counted and diluted to a concentration of 1,000 cells μl$^{-1}$.

**10X Genomics scRNA-seq library preparation.** Single-cell gene expression libraries were prepared using the 10X Genomics Chromium Next GEM Single Cell 3′ GEM, Library & Gel Bead Kit v3.1 (10X Genomics, no. PN-1000121) according to the manufacturer's instructions. The aimed target cell recovery in each experiment was 10,000 cells.

To generate Gel beads in EMulsion (GEMs), reactions were assembled in a Chromium Next GEM Chip G (10X Genomics, no. PN-1000121). Chips were run on a Chromium Controller X/iX. Sample indices were added to the cDNAs via polymerase chain reaction (PCR) using the Single Index Kit T Set A (10X Genomics, no. PN-1000213) or Dual Index Kit TT Seat A (10X Genomics, no. PN-1000215). Library concentration was measured by Qubit dsDNA HS Assay Kit (Invitrogen, no. Q33231) and quality was assessed using Bioanalyzer High Sensitivity DNA Analysis Kit (Agilent, no. 5067-4626). Finally, scRNA-seq libraries were sequenced on an Illumina NovaSeq 6000 with asynchronous 28 bp/90 bp paired-end reads. Single-cell experiments were performed in biological duplicates, with each replicate pair derived from a single different individual.

### Single-cell RNA analysis
**Filtering, normalization and integration.** The different scRNA-seq libraries were processed using 10X Genomics CellRanger v6.0.2[74] and our custom genome annotations for *C. perspicillata* and *M. musculus*. Individual count matrices were filtered for quality based on relative unique molecular identifier (UMI) counts (removed >4 × mean and <0.2 × median of the sample), percentage of ribosomal UMIs using the median absolute deviation (MAD) (removed >median + (3 × MAD) and <median + (3 × MAD)), and relation of UMI count/genes detected (removed <0.15 and UMI count <2/3). The filtered datasets were integrated in a species/limb manner (for example, all mouse FL datasets together). For this we used Seurat v4.3.0[75], first log normalizing each dataset with a factor of 10,000 and scoring the cell cycle state of each

cell. Then, using the SCTransform tool we regressed the percentage of ribosomal UMIs, the UMI count and the S and G2M cycle score of each cell. Using the top 25% integration features, we found integration anchors using the SCT normalization, 20 first dimensions and a K filter of 100. These anchors were used with the Seurat v4.3.0 function IntegrateData and the normalization method 'SCT'.

**Dimensionality reduction and clustering.** Using the integrated data from the top variable features (standard variance > median + MAD of the data) we calculated a principal component analysis (PCA) and used the first 20 (18 for the chiropatagium samples) PCs downstream. We calculated t-distributed Stochastic Neighbor Embedding (t-SNE) plots using fast Fourier transform-accelerated interpolation-based t-SNE[76] in order to retain and represent the global structure of the data[77]. Clusters were computed using FindNeighbors and FindCLusters with a resolution of 0.7 and 42 as a random seed. Marker genes for each cluster were then calculated from the un-integrated expression data using a minimum percentage of expression of 0.25, and a log2 fold change (log2FC) threshold of 0.5.

The cells identified as LPM-derived were subsetted, and the same process was followed to generate the presented individual datasets. From the MmHl dataset, a cluster with a high proportion of haemoglobin UMIs was removed.

**Interspecies atlas.** The integration of all Cp and Mm limb LPM-derived cells followed the same logic. We subsetted the LPM cells from each of the datasets, separated the individual libraries and used the same workflow. For this integration we used the top 12.5% of the integration features. To find clusters we used a resolution of 0.6. The identity labels of this integration were used for the individually clustered datasets (for example, Mm Fl). Each cluster was first classified as one of the three main groups of LPM cells by simple majority, and then was given the identity of the most represented label from that group.

**Apoptosis-related expression comparison.** The cells from the integrated cluster 3 RA-Id were compared against the rest of the cells in a species/limb fashion (for example, cells within the Mm Fl dataset) using the function FindMarkers with a minimum expression percentage and a log2FC threshold of 0.0001, using all genes related to apoptosis, the marker genes from the cluster and 20 random marker genes from other clusters.

**Label transfer to chiropatagium cells.** To transfer annotation labels from the Cp Fl-LPM dataset to the chiropatagium-LPM cells, we found transfer anchors using the first 20 PCs and the SCT normalized data and used the TranferData function.

**Pseudotime analyses.** We calculated RNA velocities using velocyto[78] on our single-cell libraries, using the stricter mode for the Cp samples. From each individual LPM sample, we subsampled the Hoxd13+ cells (>0 UMIs). We re-clustered and annotated the dataset, and then exported it to an AnnData format and integrated the RNA velocity to be further analysed. Using scvelo v0.3.2[79] we filtered the data and found the first- and second-order moments with the 20 first PCs. We then ran the dynamical model and calculated RNA velocity allowing for differential kinetics. Guided by the apparent RNA velocity trajectories, and based on the identities of the clusters, we subsetted the data further to remove the chondrogenic lineage as much as possible. We then generated diffusion maps using the first 15 PCs and chose the diffusion eigenvectors 1 and 2. Using the Slingshot package we inferred trajectories of differentiation and pseudotime values for each cell using the seurat-calculated clusters, and the apparent end and start clusters according to the RNA velocity. With this data, we again computed RNA velocity without a dynamical model. Using CellRank v2.0.4[80], we computed velocity, connectivity

and pseudotime kernels, which were combined in proportions of 0.2, 0.4 and 0.4, respectively.

**Distal–proximal computational dissection.** Each species/limb dataset was given a proximal or distal score using Seurat. With the function AddModuleScore we scored each cell for the sets of genes distal (*MSX1*, *HOXD13*), proximal (*SHOX2*), chondrogenic (*SOX9*, *COL2A1*) and fibroblastic (*DCN*, *ZFHX3*). The same approach was used for other sets of genes. Per cluster, we calculated the mean of the difference between the proximal–distal scores. We then assigned clusters with the 1/3 most extreme score differences as very distal/proximal. We then categorized genes as typical proximal or distal if in both species they are expressed in at least 20% of the cells of 20% of the corresponding clusters, and they show a difference of >0.15 log2FC against the opposite cells. Genes highly expressed in the chondrogenic lineage were excluded. In Extended Data Fig. 4a,b, we show the top (by log2FC) 15 genes expressed in <15% of the opposite cells. Co-expression of genes is measured as $UMIs_x > 0$ and $UMIs_y > 0$.

**Proximo–distal fibroblast expression programme.** In order to understand what defined the expression profile of proximal and distal fibroblasts, without detecting the differences between them, the distal fibroblasts (10 FbI1) were compared against the rest of the cells, except proximal fibroblasts (cells from 8 FbA and 9 FbL labelled as 9 FbL in the main integration), and vice versa. The fibroblast programme 2 was done in the same way with cells from cluster 7 FbIr and cells from 8 FbA labelled as 8 FbA in the main integration. This was done in two rounds, first using the highly variable genes, and then using all the genes that had been detected as differentially expressed in both comparisons. GO terms enrichment analyses were made using clusterProfiler[81] and all the genes expressed in at least nine cells in the sample as the background universe. The function simplify was then used with a cut-off of 0.6 on the adjusted *P* value.

**Mutants' analysis and differential expression.** The single-cell datasets from mutant mice were subsetted for LPM cells expressing *Hoxd13* and integrated with corresponding cells from the mouse wild-type FL. Each of the new clusters found was annotated on the wild-type Mm Fl dataset following the logic before. Using MAST[82], cluster-wise we tested for differential gene expression between the wild-type and mutant cells. For this, we considered the highly variable genes, all the genes part of our proximal/distal fibroblast programme, and excluded all genes on the X chromosome, mitochondrial and ribosomal genes. Only those genes expressed in at least 15% of the cells from either genotype were tested using MAST using a zlm with the formula 'genotype + orig. ident + percent.rp'. We then calculated an lrTest on the genotype coefficient and a subsequent *P* value adjustment using p.adjust. A hypergeometric test was used to assess the over representation of the DE genes (*P* value < 0.01 log2FC > 0.25) in our programme. GO terms were calculated on the totality of genes overexpressed in the mutant cells per cluster using the approach described above. We filtered duplicated terms based on the set of genes present. We then manually grouped the top ten terms by adjusted *P* value of all clusters in the categories presented in Fig. 4d. We calculated the mean expression of the genes in our proximal/distal fibroblast programme in the cluster 10 FbI1 of the bat FL data, and then calculated the Pearson correlation of each cell to this mean using the same genes. For this, we focused on the clusters where we found overexpression of *Meis2* and *Tbx3* (*P* value < 0.01 log2FC > 0.15) in the mutants and the corresponding clusters in the Mm FL and Cp FL datasets.

**Fluorescent microscopy apoptosis assays using LysoTracker and Immunofluorescence against cleaved caspase-3**
Bat embryonic limbs were dissected in cold DPBS and separately processed for the two different cell death assays.

For the lysosomal staining, samples were transferred immediately into 5-μm LysoTracker Red DND-99 (Invitrogen, no. 12090146) in DPBS and incubated at 37 °C for 45 min, then washed four times in DPBS and fixed overnight in 4% PFA/DPBS. After that, samples were washed for 10 min in DPBS, dehydrated through a methanol series (25, 50, 75 and 100%) and stored at −20 °C until imaging.

For the caspase assay, samples were fixed in 4% PFA/DPBS for 1 to 2 h at 4 °C, then washed three times and stored in DPBS at 4 °C until immunofluorescent staining was performed. Samples were washed twice in DPBS for 5 min, then permeabilized in 0.5% Triton-X-PBS (PBST) (3 × 1 h incubation) and blocked in 5% FCS/PBST overnight at 4 °C. Anti-Cleaved-Caspase-3 (D175) Antibody (Rabbit Polyclonal, Cell Signalling Technology, no. 9661, lot 47) was diluted in blocking solution (1:400) and incubated for 72 h at 4 °C. Samples were washed three times with blocking solution and three times with PBST, and then incubated in blocking solution overnight at 4 °C. Donkey Anti-Rabbit IgG (H + L) Highly Cross-Adsorbed Secondary Antibody, Alexa Fluor568 (Invitrogen, no. A10042, lot 2306809) and DAPI were diluted in blocking solution (1:1,000) and incubated for 48 h at 4 °C. Samples were washed three times with blocking solution, three times with PBST, three times with DPBS and post-fixed in 4% PFA/DPBS for 20 min.

**Confocal fluorescence imaging.** At this point, samples from both experiments were similarly treated. They were washed three times with 0.02 M phosphate buffer (pH 7.4) and cleared in refractive index matching solution (13% Histodenz (Sigma-Aldrich D2158) in 0.02 M PB) at 4 °C for at least one day. Whole-mount limbs were then imaged with a Zeiss LSM880 confocal laser-scanning microscope in fast-Airyscan mode. At least 20 *z*-stacks were imaged, covering the entire limbs. *Z*-stacks were then merged as maximum intensity projection with the ZEN software and Airyscan processing was performed. Scale bars were added with Fiji.

### Gene regulatory network analysis
Gene regulatory networks were generated using the Python implementation of SCENIC (pySCENIC)[83].

Raw counts, without SCTransform, and cell-type identities were extracted from the generated Seurat object. These counts were then filtered as described by pySCENIC authors. This included filtering out cells with less than 200 or more than 6,000 genes with counts, and filtering genes appearing in less than three cells.

Vertebrate motifs were downloaded from JASPAR at the following link: https://jaspar.elixir.no/download/data/2024/CORE/JASPAR2024_CORE_vertebrates_non-redundant_pfms_jaspar.zip. These motifs were converted to clusterbuster motifs using Biopython's motif submodule, for input into pySCENIC. TF and motif names were also extracted from the downloaded motifs.

Adjacencies between these TFs were calculated with the filtered counts, using pySCENIC's GRN function. Regulons, highlighting enriched motifs, were then calculated from these adjacencies using pySCENIC's CTX function. Cells where the TF or target gene expression were 0 were masked when calculating correlation between a TF–target pair, both positive and negative regulons were calculated, and no pruning was performed. pySCENIC's default behaviour is to prune regulons based on *cis*-regulatory information; however, owing to lack of compatibility with the novel bat annotation, this step was skipped. Hence, all enriched motifs are included, and regulons were filtered downstream.

The area under the curve (AUC) was then calculated on these regulons using the filtered counts, to determine regulon enrichment using pySCENIC's AUCELL function. This AUC information was combined with cell-type labels to generate RSSs for both positive and negative regulons across all cell types.

Finally, these RSSs were used to generate gene regulatory networks, determining the edges between regulon nodes. Only TF–target

gene connections with a weight (adjacency) greater than 10 were included, filtering out weak TF–target pairings. Moreover, target genes with a mean expression across all cells less than 0.05 were also excluded.

## RNA-seq and analysis

Total RNA was extracted using RNeasy Mini Kit (Quiagen, no. 74106) or RNeasy Micro Kit (Quiagen, no. 74004) according to the manufacturer's instructions. Limb samples derived from E11.5 and E13.5 as well as CS15 and CS17 embryos were directly homogenized in RTL buffer supplemented with 1% β-mercaptoethanol and applied to spin columns. Limb tissues from older embryonic stages (E15.5 and CS19) were crushed in liquid nitrogen using Bel-Art SP Scienceware Liquid Nitrogen-Cooled Mini Mortar prior to homogenization. Genomic DNA was removed using the RNase-Free DNase Set (Quiagen, no. 79254).

For gene expression analysis, samples were poly-A enriched and libraries were prepared using the Kapa HyperPrep Kit (Roche, no. 07962347001). RNA libraries were sequenced on a Novaseq2 with 100 bp paired-end reads. RNA-seq experiments were performed in biological duplicates for the bat samples. For the mouse samples, three and five biological replicates were used for the *TBX3* transgenic and for the *MEIS2* transgenic and wild-type mice, respectively. Read mapping to mm39 and *C. perspicillata* reference genomes was performed using the STAR_2.6.1d software[84] with the following options: --chimSegmentMin 10 --alignIntronMin 20 --outFilterMismatchNoverReadLmax 0.05 --outSAMmode NoQS --outFilterMismatchNmax 10. For samples obtained from transgenic animals, the transgene sequence was temporarily merged with the genome annotation using the option -add. For each sample, read counts per gene were obtained via the R function 'featureCounts', with the parameter --countReadPairs -s 2. For visualization, counts per million (CPM)-normalized bigwig files were created using the 'bamCoverage' tool from deepTools[85] or read counts were normalized to reads per kilobase million based on the number of uniquely mapped reads.

**DEGs.** DEGs were identified from featureCounts[86] count matrices using DESeq2 v1.38.3 (R v4.2.2)[87]. For each comparison, replicate quality was assessed using PCA and Euclidean distance between samples. When necessary, outlier replicates were removed from further analysis. Lowly expressed genes were then filtered using edgeR v3.40.2 filterbyexpr[88] and non-annotated transcripts were removed to ensure one-to-one comparisons between the species. Counts were normalized and differential expression calculated as log2FC from the mean of normalized counts. Genes were assigned as differentially expressed for log2FC larger than ±0.5 and adjusted Wald test *P* value below 0.05.

## ChIP-seq and analysis

Mouse and bat embryonic limbs were dissected and fixed in 1% formaldehyde in 10% FCS/PBS and subsequently snap-frozen and stored at −80 °C until further processing. Chromatin immunoprecipitations were performed using the iDeal ChIP-seq Kit for Histones (Diagenode, no. C01010051) and iDeal ChIP-seq Kit for Transcription Factors (Diagenode, no. C01010055) according to the manufacturer's instructions. Briefly, fixed limbs were lysed and sonicated using a Bioruptor Plus Sonication device (45 cycles, 30 s on, 30 s off, at high power setting) in provided buffers. A total of 5 μg sheared chromatin was used for histone immunoprecipitation with 1 μg of the following antibody: anti-H3K27ac (Diagenode, no. C15410174, RRID:AB_2716835). For MEIS2 immunoprecipitation, 20 μg of sheared chromatin was used with two anti-MEIS antibodies simultaneously, one recognizing the conserved C-terminal domain of MEIS1a and MEIS2a, and the other recognizing all MEIS2 isoforms as previously described[34]. A total of 2 μg of each antibody was used per immunoprecipitation. Antibodies were generously provided by M. Torres. Libraries were prepared using the Kapa HyperPrep Kit (Roche, no. 07962347001) and libraries were sequenced

on a NovaSeq2 with 100 bp paired-end reads. ChIP-seq experiments were performed in biological duplicates.

E10.5 MEIS1/2 ChIP data from mouse FLs were obtained from previously published data[34].

Read mapping of the sequenced samples to mouse and bat reference genomes (mm39/carPer2) was performed using the STAR_2.6.1d software[84]. Reads were then filtered and sorted, and duplicates were removed using SAMtools[89]. For visualization, CPM-normalized bigwig files were created using deepTools 'bamCoverage' tool. Input samples were subtracted using deepTools bamCompare[85].

**H3K27ac differential peak and motif enrichment analysis.** Differential acetylation regions between distal FL (dFL) and distal HL (dHL), as well as acetylation regions common in both conditions, were predicted from ChIP-seq alignments using macs2[90] bdgdiff command with the following parameter: -l 800 -g 500 and a likelihood-ratio cut-off of 1,000.

The coverage of dFL-specific acetylation regions relative to the acetylation regions shared between dFL/dHL was calculated for each topologically associated domain (TAD).

The dFL-specific acetylation regions and the acetylation regions shared between dFL/dHL were first intersected with accessible regions in limbs (CS17). The dFL-specific accessible acetylation regions were given as input and the commonly accessible acetylation regions as background for motif enrichment analysis done by homer2[91]. A *q* value cut-off of 0.01 was used for the enriched motifs. The list of significantly enriched motifs was manually curated so that only the most significant TF motifs in each gene family were retained.

Tornado plots were generated for peak distribution visualization using deeptools v3.5.4[85] The scores per region were calculated using computeMatrix in scale-regions mode, where the scores were based on the ChIP bigwig pileup files and the regions based on BED files defining the ChIP peaks.

**MEIS binding analysis.** MEIS2 binding signal of one of our dFL replicates was aggregated by TADs. The signal (AUC) was calculated using deepTools pyBigwig[85]. By analysing the second derivative of the density distribution of the signal by TAD, we found the dividing point to the subpopulation of enriched TADs. The same procedure was carried out for the coverage length of the acetylation peaks. Genes were categorized as 'Cytoskeleton', 'ECM', or 'Transcription Factor' using the Uniprot database, if the keywords included the terms 'Cytoskeleton', 'Extracellular matrix', or ('DNA-binding' OR "Transcription regulation"), respectively.

## ATAC-seq

ATAC-seq protocol and library preparation was performed as previously described[92]. In short, bat limbs were dissected and dissociated with trypsin. A total of 50,000 cells per reaction were lysed and isolated nuclei were incubated with Tn5 Transposase (Illumina, no. 20034197) for transposition. DNA fragments were purified and barcoded adaptors were added via PCR. Fragments were purified and sequenced as 100 bp paired-end reads on a NovaSeq 6000 system. ATAC-seq experiments were performed in biological duplicates.

For processing, adaptors were trimmed with the cutadapt tool[93] and reads were mapped to indexed reference genomes (mm39/carPer2) using Bowtie2[94]. Reads were then filtered and sorted, and duplicates removed according to ChIP-seq processing. Reproducible peaks were called using Genrich with default parameters.

## *C. perspicillata* embryonic fibroblast culture

Head- and organ-free tissue from a CS16/CS17 female bat embryo (*C. perspicillata*) was minced in DMEM supplemented with 15% FCS and cryo-frozen in DMEM containing 10% DMSO and 15% FCS until further processing. To establish bat embryonic fibroblast culture, tissue pieces were thawed and digested with trypsin for 20 min at 37 °C. Cells were centrifuged at 1,000 × *g* for 5 min, resuspended in

fibroblast culture media (DMEM high glucose, 15% FCS, 1% Pen/Strep, 1% L-glutamine) and transferred to a six-well plate for cell attachment and expansion. Cells were split into a new culture flask when they reached a density of approximately 80% or cryo-frozen in freezing media (DMEM high glucose supplemented with 15% FCS and 10% DMSO) in $1-3 \times 10^6$ aliquots. Fibroblast cells were cultured at 37 °C and 5% $CO_2$.

## Hi-C

Hi-C libraries from *C. perspicillata* embryonic fibroblast cells or bat embryonic FLs were prepared as previously described[95]. In short, approximately $1 \times 10^6$ fibroblast cells and 500,000 limb cells were fixed in 2% formaldehyde in 10% FCS/PBS. After cell lysis, chromatin was digested with DpnII enzyme (NEB, no. R0543), digested ends were marked with biotin-14-dATP (Invitrogen, no. 19524016) and subsequently ligated. Crosslinking was reversed, DNA precipitated and sheared to a fragment size of 300–600 bp using an S-Series 220 Covaris sonicator. The biotin-containing fragments were pulled down with Dynabeads MyOne Streptavidin T1 beads (Invitrogen, no. 65602) and ends were repaired using Klenow Fragment DNA polymerase I (NEB, no. M0210) and T4 DNA polymerase (NEB, no. M0203). Adaptors were added to DNA fragments using NEBNext Multiplex Oligos for Illumina kit (NEB, no. E7335S) and sequencing indices were added by PCR using NEBNext Ultra II Q5Master Mix (NEB, no. M0544). Hi-C libraries were generated as three technical replicates and sequenced on a NovaSeq2 as 100 bp paired-end reads.

For read processing, the *C. perspicillata* reference genome (carPer2) was indexed with the short-read aligner BWA 0.7.17[96]. Raw reads from sequenced Hi-C libraries were then processed using the Juicer pipeline v1.5.6[97]. The three replicates were processed independently and subsequently merged after filtering and deduplication. Hi-C maps with various bin sizes were generated using Juicer tools 1.11.09[97] using the parameter pre -q 30. For displaying Hi-C maps as heatmaps, KR-normalized maps with 5 kb bin size were used.

**TAD calling.** TADs were called using Hi-C data of *C. perspicillata* embryonic fibroblasts with the software TopDom[98] (KR-normalized, resolution: 50 kb, window size: 10).

## Cloning ectopic expression constructs

CRISPR-Cas9 single-guide RNA (sgRNA) construct targeting the safe harbour locus H11 was generated using the same sequence as previously described[99]. sgRNA oligos were cloned into *BbsI* digested and dephosphorylated pSpCas9(BB)-2A-Puro (PX459) V2.0 vector (Addgene; no. 62988). sgRNA sequences can be found in Supplementary Table 1.

For cloning of expression constructs, a pUC-Amp plasmid containing homology arms (0.7 kb) designed on the *H11* knock-in site was ordered from Twist Bioscience, and the Hsp68 promoter was used as minimal promoter. A previously described interdigital enhancer from the *Bmp2* locus[41] was amplified from wild-type G4 cell DNA; *C. perspicillata MEIS2* cDNA was ordered from Twist Bioscience as a fragment; *C. perspicillata TBX3* cDNA was ordered from GeneWiz (Azenta Life Sciences) as a pUC-Amp vector. Kozak sequence (GAGTGG), SV40 polyA signal were included in the design of both overexpression constructs. Backbones and fragments were amplified by PCR (PrimeSTAR GXL Polymerase (Takara, no. R050A)), introducing also overlapping sequences necessary for Gibson assembly. Fragments were assembled using Gibson Assembly Master Mix (NEB, no. M5510) and cloned into 5-alpha Competent *E. coli* (NEB, no. C2987). Products were validated via restriction digestion and subsequent sequencing. Plasmids were purified using Nucleobond Xtra Midi EF kit (Macherey-Nagel, no. 740420) before transfection.

For alignment of MEIS2 and TBX3 protein sequences, bat and mouse coding sequences were translated into amino acid sequences using ExPASy[100]. Sequences were aligned using the Multiple Sequence Alignment tool MultAlin[101].

## Mouse embryonic stem cell culture

Mouse G4 embryonic stem cell (ESC) culture (XY, 129S6/SvEvTac × C57BL/6Ncr F1 hybrid) was performed as previously described[102,103]. Briefly, mouse G4 ESCs were grown on a monolayer of mitomycin-inactivated CD1 mouse embryonic fibroblast feeders on gelatin coated dishes at 37 °C and 7.5% $CO_2$. ESC culture medium containing knockout DMEM with 4.5 mg ml$^{-1}$ glucose and sodium pyruvate (Gibco, no. 10829-018) supplemented with 15% FCS (PANSera ES, no. P30-2600), 10 mM glutamine (Lonza, no. BE17-605E), 1× penicillin/streptomycin (Lonza, no. DE17-603), 1× non-essential amino acids (Gibco, no. 11140-35), 1× nucleosides (Chemicon, no. ES-0008D), 0.1 mM beta-mercaptoethanol (Gibco, no. 3150-010) and 1,000 U ml$^{-1}$ leukaemia inhibitory factor (Chemicon, no. ESG1107) was changed daily. Mouse ESCs were split every 2–3 days or were frozen at a density of $1 \times 10^6$ cells per cryovial in ESC medium containing 20% FCS and 10% DMSO and stored in liquid nitrogen. Cell lines used in this study are summarized in Supplementary Table 1.

## Knock-in genome editing using CRISPR technology

CRISPR-mediated genome editing was subsequently performed as described previously[104]. In short, 300,000 G4 ESCs were seeded on CD1 feeders 16 h prior to transfection. For site-specific knock-ins at H11 locus, ESCs were co-transfected with 8 μg of the sgRNA and 4 μg of the knock-in homology construct. After 24 h, transfected cells were split onto puromycin-resistant DR4 feeders in a ratio of 1:3. For antibiotic selection, cells were treated with puromycin for 48 h. For recovery, mouse ESCs were grown for 4–6 days, after which single colonies were picked into 96-well plates containing CD1 feeders. Cells were grown and split into triplicates after reaching sufficient size. One plate was used for DNA harvesting and genotyping of clones. The other two plates were frozen and stored at −80 °C for expansion of positive clones.

Clones were screened for expression construct knock-ins by PCR detecting site-specific insertion breakpoints. Copy numbers of the insertions were then assessed by quantitative PCR. Positive clones were selected for tetraploid complementation. All primers used for these experiments can be found in Supplementary Table 2 and the recombinant DNAs purchased to companies can be found in Supplementary Table 3.

## Generation of mutant embryos by tetraploid aggregation

For generation of transgenic embryos, selected mutant ESCs were seeded on CD1 feeders, grown for 2 days and then subjected to aggregation by tetraploid morula complementation, as previously described[66]. Female mice of CD1 strain were used as foster mothers.

## Whole-mount in situ hybridization

*MEIS2* transgene mRNAs were detected in embryos by WISH using digoxigenin-labelled antisense RNA probes prepared with DIG RNA labelling mix (Roche, no. 11277073910). Embryos were dissected, fixed in 4% PFA/PBS overnight and dehydrated in a methanol series (25%, 50%, 75% methanol in 0.1% Tween in DPBS) on ice. They were stored in 100% methanol at −20 °C. For staining, embryos were rehydrated in a reversed methanol/PBST series, bleached in 6% $H_2O_2$/PBST for 1 h, treated with 10 μg ml$^{-1}$ proteinase K in PBST for 5 min, and re-fixed in 4% PFA/PBS with 0.2% glutaraldehyde and 0.1% Tween 20. After washing in PBST, embryos were incubated in L1 buffer (50% deionized formamide, 5× SSC, 1% SDS, 0.1% Tween 20) at 68 °C for 10 min, followed by hybridization buffer 1 (L1 with 0.1% tRNA and 0.05% heparin) for 2 h, and then in hybridization buffer 2 (hybridization buffer 1 plus 1.5 μg digoxigenin-labelled RNA probe per embryo) overnight at 68 °C. The next day, unbound probe was washed away using L1, L2 (50% deionized formamide, 2× SSC, 0.1% Tween 20) and L3 buffer (2× SSC, 0.1% Tween 20) for three 30-minute intervals at 68 °C. Embryos were treated with RNase solution (0.1 M NaCl, 0.01 M Tris pH 7.5, 0.2% Tween 20, 100 μg ml$^{-1}$

RNaseA) for 1 h, washed in PBST and blocked in TBST 1 (2% FBS, 0.2% BSA) for 2 h at room temperature. They were then incubated with 1:5,000 anti-digoxigenin-conjugated to alkaline phosphatase antibody (Roche, no. 11093274910) in blocking solution overnight at 4 °C. Unbound antibody was washed off with TBST 2 (TBST with 0.1% Tween 20, 0.05% levamisole). For staining, embryos were washed in alkaline phosphatase buffer (0.02 M NaCl, 0.05 M MgCl2, 0.1% Tween 20, 0.1 M Tris-HCl, 0.05% levamisole), and then stained with BM Purple AP Substrate (Roche, no. 11442074). Finally, stained embryos were imaged using a ZEISS SteREO Discovery.V12 microscope with a Leica DFC420 camera. The primers used to generate the transgene *MEIS2* probe can be found in Supplementary Table 1.

### Limb 3D imaging

PFA-fixed mouse embryo limb specimens were incubated in a solution of 25 µM DraQ5, dissolved in PermBlock solution (1% BSA, 0.5% Tween 20 in PBS), for 12 h. Following three washes with PBS-T, the stained specimens were dehydrated in increasing methanol concentrations (50%, 70%, 95% and 99% (v/v) methanol in ddH$_2$O). Subsequently, the specimens were stained in a solution of 1.5 µM Eosin Y, dissolved in a 1:1 methanol:BABB (benzyl alcohol:benzoate, ratio 1:2) solution for 4 h, followed by an optical clearing procedure with BABB solution twice for 4 h each.

After fluorescence whole-mount staining, optically cleared embryo limb biopsies were imaged using the Lightsheet 7 (Zeiss). The stacks were captured with a step size of 2.5 µm and at 5× magnification. The ZEN 3.1 (black edition) software was utilized for the operation of the light sheet microscope and the acquisition of the images. The digital 3D reconstruction of light sheet image stacks was conducted using the IMARIS Microscopy Image Analysis Software (Oxford Instruments).

All quantifications were performed using IMARIS microscopy image analysis software. For the measurement of the volume of the mouse limb, the autopod region devoid of fingers was considered and analysed using the volume function in IMARIS for each condition and $n = 4$ independent specimens. The total number of cells was quantified within DraQ5-positive nuclei of the autopod, excluding the fingers of the limb, using the spots function in IMARIS for each condition and from $n = 4$ independent samples. For the quantification of connective tissue in the limbs, Eosin Y-positive structures of the limbs were analysed using the volume function in IMARIS for each condition and $n = 4$ independent specimens. The differences of the mean were calculated using a Dunnett test following a one-way analysis of variance.

### Reporting summary

Further information on research design is available in the Nature Portfolio Reporting Summary linked to this article.

## Data availability

Raw and processed functional data produced in this work have been deposited in the Gene Expression Omnibus under GSE275848, GSE275851, GSE275853, GSE275854 and GSE275855. The genome assembly has been deposited at the NCBI repository under BioProject ID PRJNA1265070 and BioSample ID SAMN48582796.

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

## Acknowledgements

We thank A. C. Stiege and U. Fischer for their technical support. We thank the Papiliorama Zoo directorate and staff members for their support during sampling. We thank the people from the transgenic and sequencing units from the Max Planck Institute for Molecular Genetics for their assistance. We thank M. Torres for kindly providing us with the MEIS antibodies. We thank all the members from Mundlos, Lupiáñez and Real groups for their fruitful discussions. C.F. was supported by the EMBO Postdoctoral Fellowship ALT 260-2021. S.M. was supported by a grant (GenRevo) from the European Research Council (ERC) and by the grant MU 880/27-1 from the Deutsche Forschungsgemeinschaft. F.M.R. was supported by grants from the Spanish Research Council (RYC2022-035182-I and PID2023-151163NA-I00/MCIN/AEI/10.13039/501100011033). Research in the Lupiáñez lab was funded by the ERC (grant no. 101045439, 3D-REVOLUTION) and by the Spanish 'Agencia Estatal de Investigación' (grant no. PID2022-143253NB-I00/AEI/10.13039/501100011033/FEDER, UE). Funded by the European Union. Views and opinions expressed are however those of the author(s) only and do not necessarily reflect those of the European Union or the ERC Executive Agency. Neither the European Union nor the granting authority can be held responsible for them. R.H. was supported by the ERC (PREVENT, 101078827). This work was supported by the German Research Foundation (grant HI1423/5-1) and the LOEWE-Centre for Translational Biodiversity Genomics (TBG) funded by the Hessen State Ministry of Higher Education, Research and the Arts (HMWK; LOEWE/1/10/519/03/03.001(0014)/52). M.A.M.-R. acknowledges support from the Spanish Ministry of Science and Innovation (PID2020-115696RB-I00 and PID2023-151484NB-I00) as well as the Catalan government through the AGAUR agency (SGR 01127). A.B. acknowledges support from the Spanish Ministry of Science and Innovation, the State Investigation Agency, and the European Social Fund Plus (PRE2022-101632).

## Author contributions

S.M. and F.M.R. conceived the study. S.M., D.G.L. and F.M.R. designed and supervised the experiments. M.S., S.A., A.R.R., G.A., N.F. and F.M.R. collected samples in Kerzers. M.S., S.A., J.H., B.K.K., S.M. and F.M.R. collected samples in Frankfurt. M.S. generated all the omics data, performed read mapping, designed experiments and composed all the figures. C.F. designed experiments and analyses, performed all the single-cell analyses, and integrated epigenomic analyses. A.E.M. produced the genome annotation and S.A.H. and I.U. further improved it. T.Z. established the genome browser and performed

preliminary bulk analyses. S.A. and J.G. performed the cell death assays. A.B. and M.A.M.-R. performed the gene regulatory network analyses. B.-W.L. analysed the regulatory domains and bulk epigenetic data. A.A.M. analysed the bulk RNA data. M.S. and S.A. generated the mutant mice. R.Y.B. and R.H. performed the 3D imaging of the mutant limbs. R.H., S.A.H., M.V., M.A.M.-R, M.H., D.G.L., S.M. and F.M.R. provided supervision. M.S., C.F., S.M. and F.M.R. wrote the paper with the help of all other co-authors.

## Funding

## Competing interests

The authors declare no competing interests.

## Additional information

**Extended data** is available for this paper at https://doi.org/10.1038/s41559-025-02780-x.

**Correspondence and requests for materials** should be addressed to Darío G. Lupiáñez, Stefan Mundlos or Francisca M. Real.

Magdalena Schindler[1,2,21], Christian Feregrino ⓘ [1,3,21], Silvia Aldrovandi ⓘ [1], Bai-Wei Lo[1], Anna A. Monaco ⓘ [1,2], Alessa R. Ringel ⓘ [1,2], Ariadna E. Morales ⓘ [4,5,6,7], Tobias Zehnder ⓘ [1,2], Rose Yinghan Behncke ⓘ [2,8], Juliane Glaser ⓘ [1], Alexander Barclay ⓘ [9,10,11], Guillaume Andrey ⓘ [12,13], Bjørt K. Kragesteen ⓘ [14], René Hägerling ⓘ [2,8], Stefan A. Haas ⓘ [15], Martin Vingron ⓘ [15], Igor Ulitsky ⓘ [16], Marc A. Marti-Renom[9,10,17], Julio Hechavarria ⓘ [18], Nicolas Fasel ⓘ [19], Michael Hiller ⓘ [4,5,6], Darío G. Lupiáñez ⓘ [3,20] ✉, Stefan Mundlos ⓘ [1,2] ✉ & Francisca M. Real ⓘ [1,2,20] ✉

[1]RG Development and Disease, Max-Planck Institute for Molecular Genetics, Berlin, Germany. [2]Institute for Medical and Human Genetics, Charité-Universitätsmedizin Berlin, Berlin, Germany. [3]Max Delbrück Center for Molecular Medicine in the Helmholtz Association (MDC), Berlin Institute for Medical Systems Biology (BIMSB), Berlin, Germany. [4]Senckenberg Research Institute, Frankfurt, Germany. [5]Faculty of Biosciences, Goethe University Frankfurt, Frankfurt, Germany. [6]LOEWE Centre for Translational Biodiversity Genomics, Frankfurt, Germany. [7]ISEM, University of Montpellier, CNRS, IRD, Montpellier, France. [8]BIH Center for Regenerative Therapies, Berlin Institute of Health at Charité - Universitätsmedizin Berlin, Berlin, Germany. [9]Centre Nacional d'Anàlisi Genòmica (CNAG), Barcelona, Spain. [10]Centre for Genomic Regulation (CRG), Barcelona Institute of Science and Technology (BIST), Barcelona, Spain. [11]Universitat Pompeu Fabra (UPF), Barcelona, Spain. [12]Department of Genetic Medicine and Development, Faculty of Medicine, University of Geneva, Geneva, Switzerland. [13]Institute of Genetics and Genomics in Geneva (iGE3), University of Geneva, Geneva, Switzerland. [14]Department of Medical Biochemistry and Biophysics, Karolinska Institute, Stockholm, Sweden. [15]Department of Computational Molecular Biology, Max Planck Institute for Molecular Genetics, Berlin, Germany. [16]Department of Immunology and Regenerative Biology, Weizmann Institute of Science, Rehovot, Israel. [17]ICREA, Barcelona, Spain. [18]Institut für Zellbiologie und Neurowissenschaft, Goethe-Universität, Frankfurt am Main, Germany. [19]Department of Ecology and Evolution, University of Lausanne, Lausanne, Switzerland. [20]Centro Andaluz de Biología del Desarrollo (CABD), Universidad Pablo de Olavide, Junta de Andalucía, Consejo Superior de Investigaciones Científicas, Seville, Spain. [21]These authors contributed equally: Magdalena Schindler, Christian Feregrino. ✉e-mail: dario.lupianez@csic.es; mundlos@molgen.mpg.de; fmarrea@upo.es

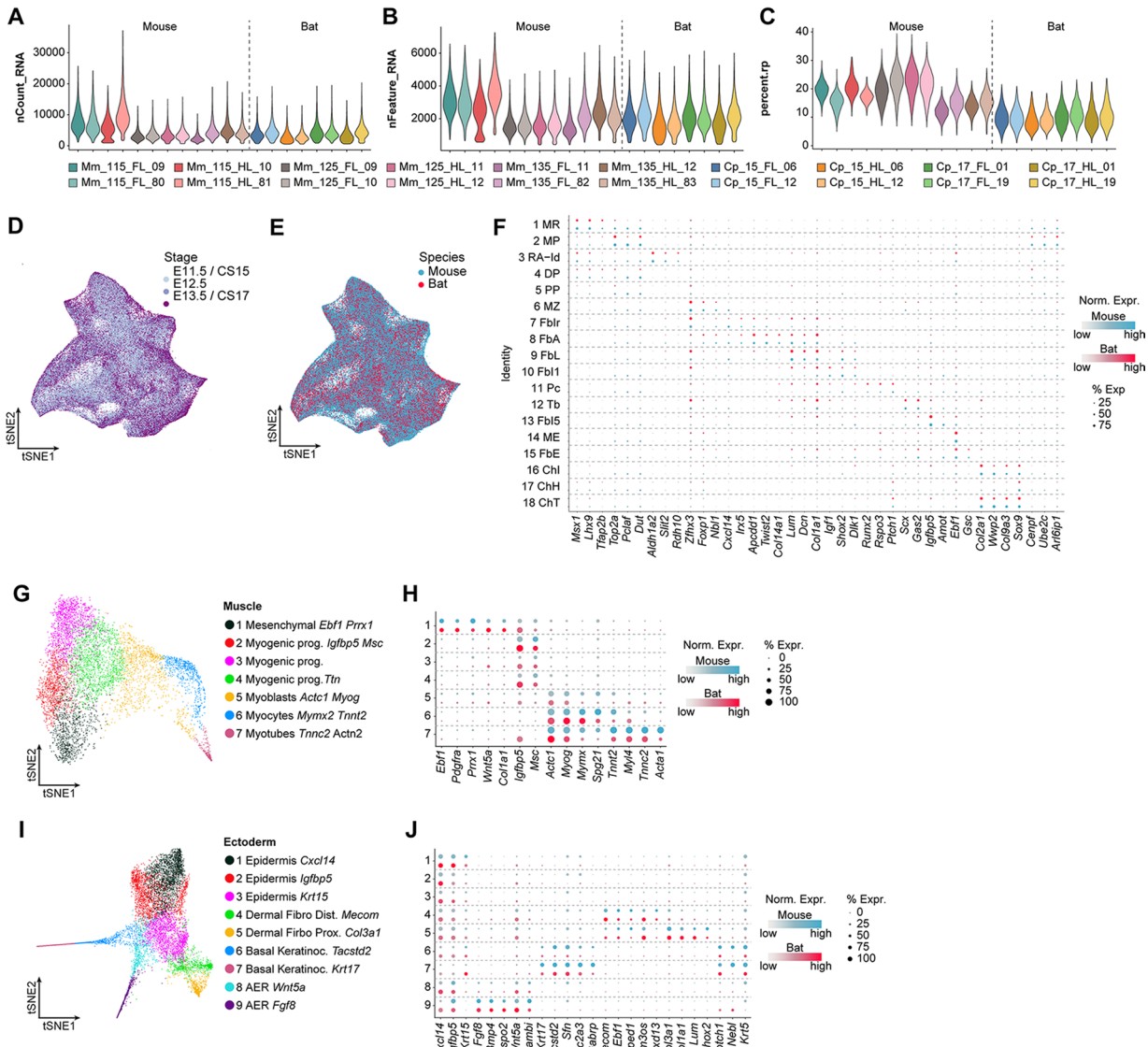

**Extended Data Fig. 1 | Integrated multi-species single-cell atlas.** Related to Fig. 1. **a–c** Quality control measurements of all single-cell libraries. nCount = Number of UMI counts per cell, nFeature = Number of detected expressed genes, percent. rp = percentage of UMIs originating from ribosomal genes. **d** Cell contribution of each stage to the integrated atlas. **e** Species contribution to the integrated atlas. **f** Dot-plot showing the top 3 differentially expressed marker genes per cluster. The color intensity indicates the expression level (blue: mouse; red: bat); the dot size represents the percentage of cells expressing respective genes. **g** Sub-clustering of the muscle cells. **h** Dot-plot showing marker gene expression used for integrated cluster annotation. The color intensity indicates the expression level (blue: mouse; red: bat); the dot size represents the percentage of cells expressing respective marker genes. **i** Sub-clustering of the ectodermal cells. **j** Dot-plot showing marker gene expression used for integrated cluster annotation.

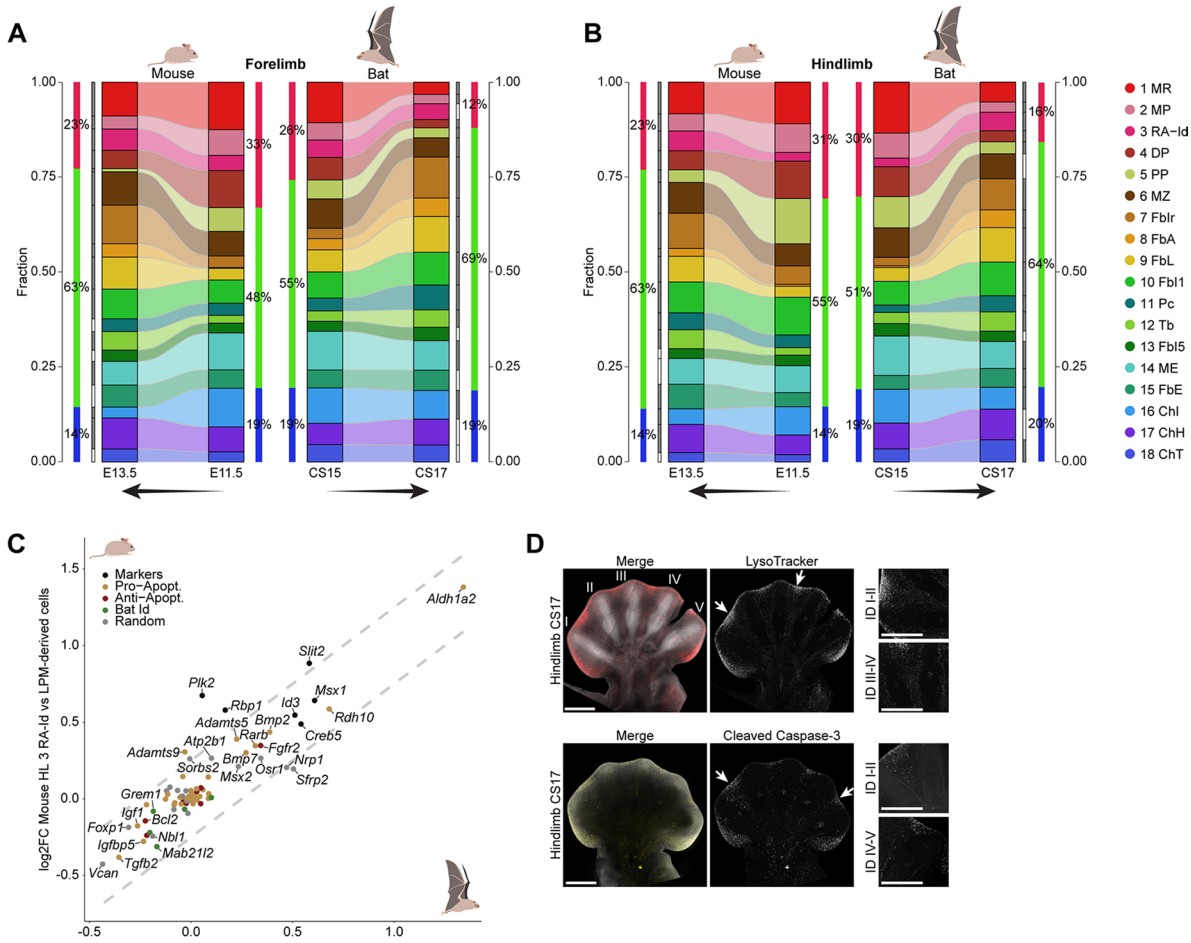

**Extended Data Fig. 2 | Interdigital cell death in hindlimbs.** Related to Fig. 1. **a** and **b** Relative cell proportions over time in mouse and bat forelimbs and hindlimbs. The proportions for Cluster 3 RA-Id are very similar between species: 5% in mouse FL, 4.3% in bat FL, and 3.2% and 3.8% in mouse and bat hindlimbs, respectively. Colored side bars represent the main developmental lineages of the LPM-derived cells. Dark gray bars represent significant changes in proportion. **c** Correlation of pro- (yellow) and anti-apoptotic (red) genes in the interspecies-integrated cell population 3 RA-Id of mouse and bat. Marker genes of this cell population are highlighted in blue; genes previously reported to be expressed in bat interdigital regions are highlighted in green. Shown is the log2FC of differential gene expression analysis between the cluster 3 RA-Id versus the rest of the LPM-derived mesenchyme per species in the HL. A set of random genes was included as control. Dashed lines represent a difference of 0.25 and −0.25 of the log2FCs (Supplementary Data 9). **d** LysoTracker staining (upper panel) and immunostaining against Cleaved Caspase 3 protein (lower panel) of bat HL at stage CS17 with magnification of interdigital regions between digits I and II and IV and V (indicated by arrows) shown on right. Merged images show DAPI (white) and LysoTracker (red) or Cleaved Caspase-3 (yellow) signal. Scale bars represent 500 μm. ID = Interdigital region.

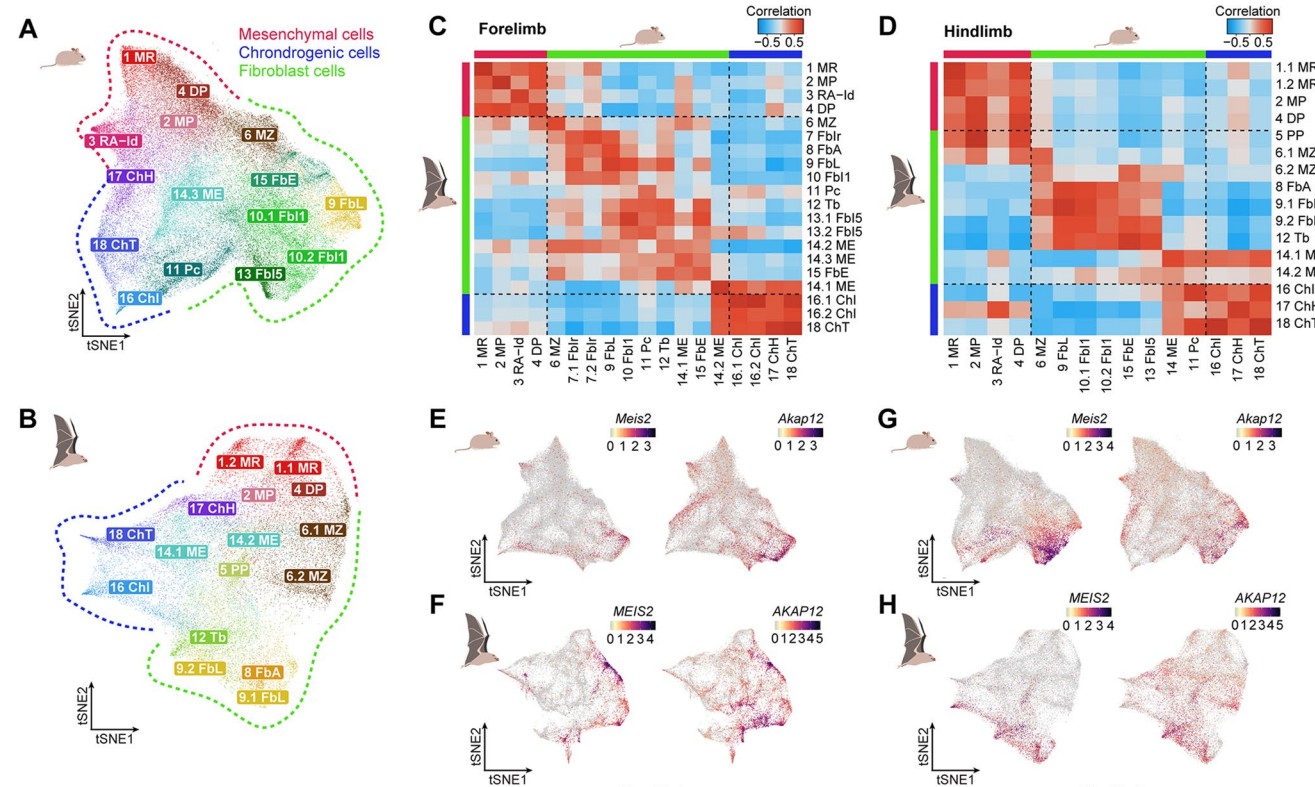

**Extended Data Fig. 3 | Species-specific clustering and cluster correspondence between species.** Related to Fig. 2. **a** Individual mouse HL LPM clustering. **b** Individual bat HL LPM clustering. **c** Clustering correspondence between mouse and bat FLs. Shown is the Pearson correlation of the log2FC of the top 10 marker genes of all clusters between the focus cluster and all other cells. **d** Clustering correspondence between mouse and bat HLs. **e** Expression pattern of *Meis2* and *Akap12* in the LPM of mouse FLs. **f** Expression pattern of *MEIS2* and *AKAP12* in the LPM of bat FLs. **g** Expression pattern of *Meis2* and *Akap12* in the LPM of mouse HLs. **h** Expression pattern of *MEIS2* and *AKAP12* in the LPM of bat HLs.

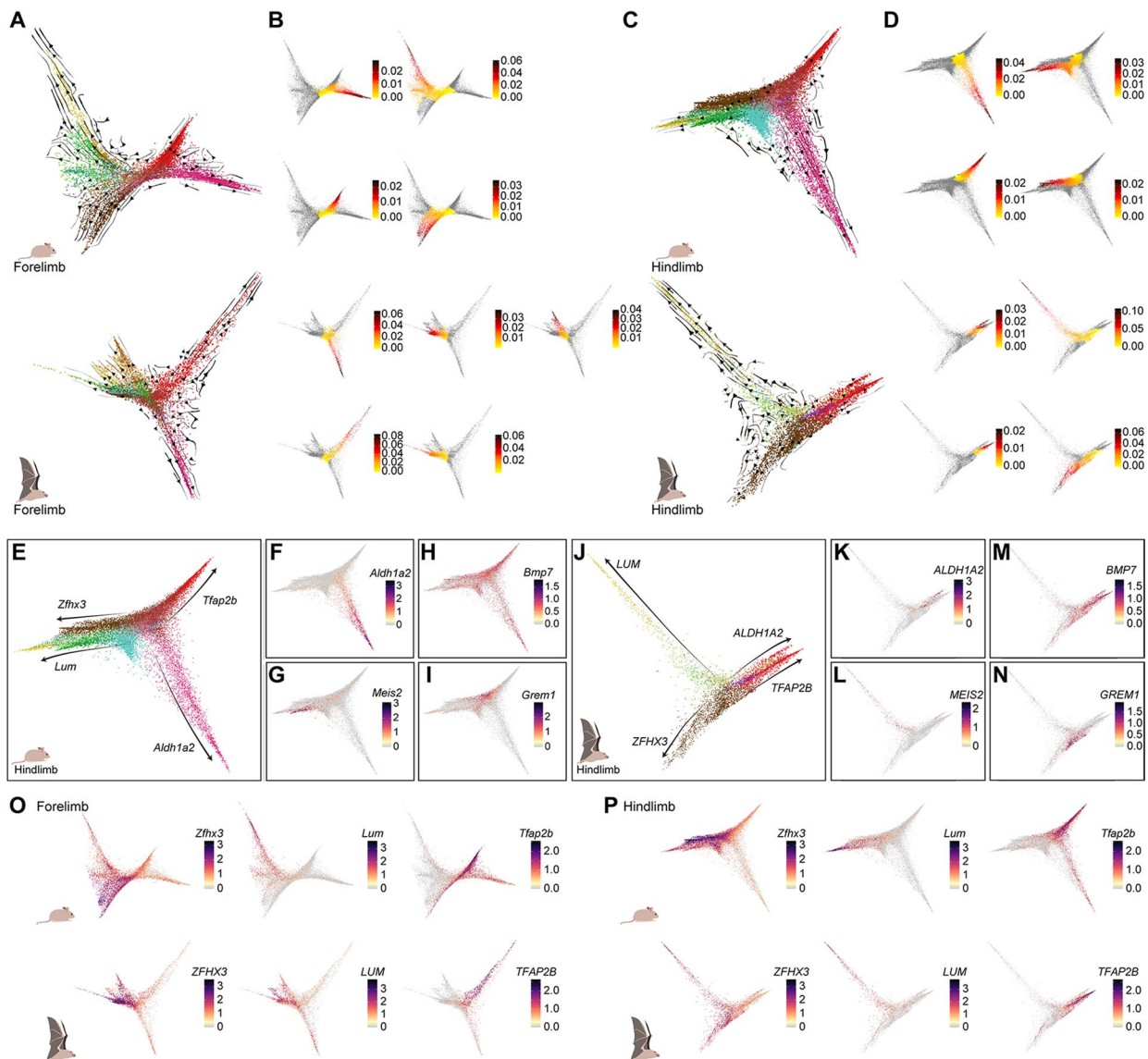

**Extended Data Fig. 4 | Differentiation trajectories of distal limb cells.** Related to Fig. 2. **a** Combined CellRank kernels (0.2*Pseudotime, 0.4*Connectivity, 0.4*Pseudotime) showing transition probabilities between non-chondrogenic *Hoxd13*+ cells of FLs of mouse (top) and bat (bottom). **b** Pseudotime values per trajectory as calculated using slingshot for FLs of mouse (top) and bat (bottom). **c** Combined CellRank kernels showing transition probabilities between non-chondrogenic *Hoxd13*+ cells of HLs of mouse (top) and bat (bottom). **d** Pseudotime values per trajectory as calculated using slingshot for HLs of mouse (top) and bat (bottom). **e** Differentiation trajectories of *Hoxd13*+,

non-chondrogenic cells of the mouse hindlimb, derived from RNA velocity and pseudotime data indicated by arrows. Trajectories were annotated based on increasing expression of marker genes. **f–i** Shown is the expression of *Aldh1a2, Meis2, Bmp7* and *Grem1* in mouse HL trajectories. **j** Differentiation trajectories of *HOXD13*+, non-chondrogenic cells of the bat HL, derived from RNA velocity and pseudotime data indicated by arrows. Trajectories were annotated based on increasing expression of marker genes. **k–n** Shown is the expression of *ALDH1A2, MEIS2, BMP7* and *GREM1* in bat FL trajectories. **o** and **p** Expression pattern of the genes used to annotate the differentiation trajectories in FLs (O) and HLs (P).

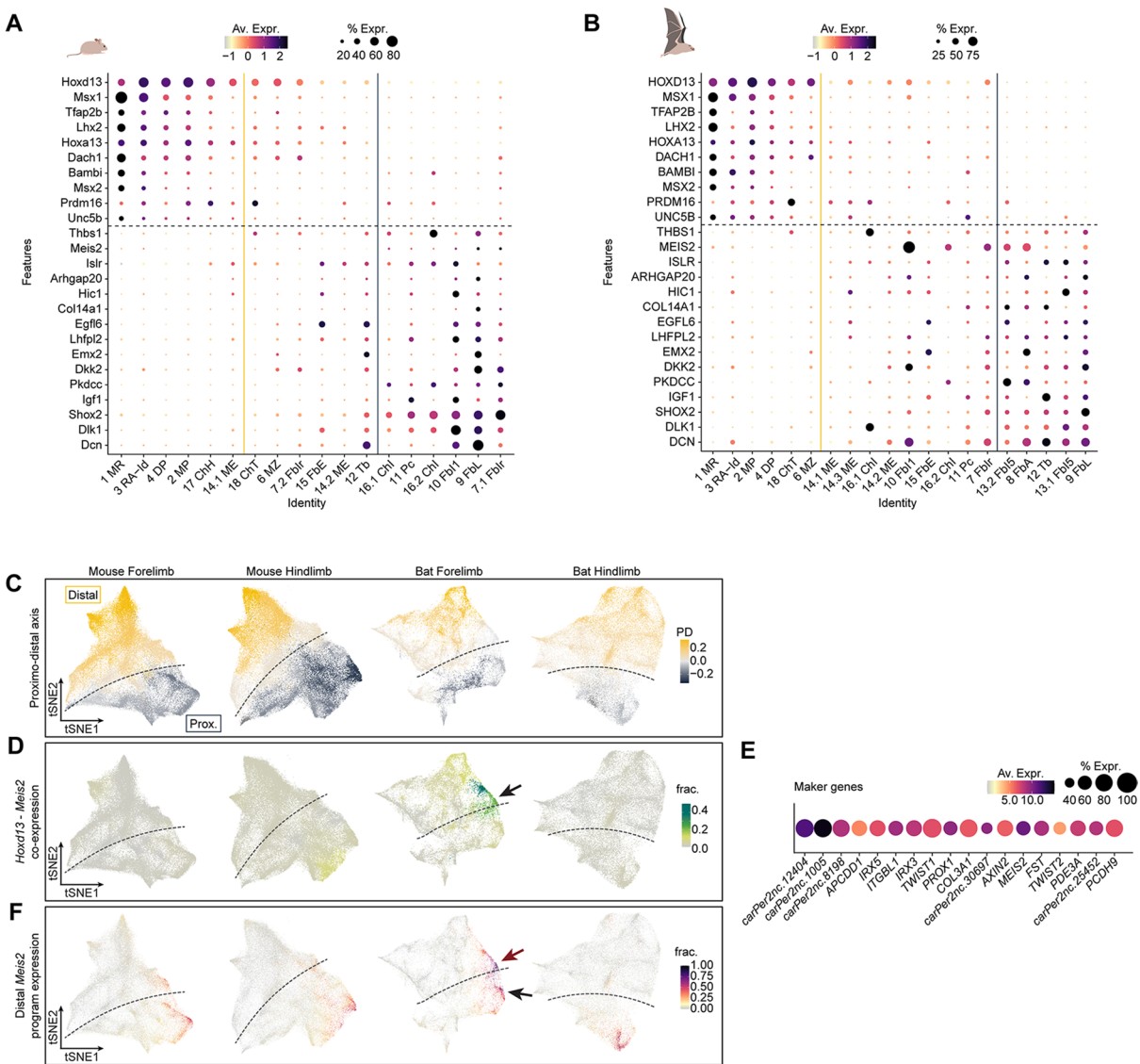

**Extended Data Fig. 5 | Proximo – Distal digital dissection.** Related to Fig. 2.
**a** Expression in mouse FLs of the genes found to be markers of FL distal
mesenchymal cells, and the top 15 markers of FL proximal mesenchymal cells.
Genes are ordered from top to bottom from most distal to most proximal.
Clusters in the same order from left to right. **b** Expression in bat FLs of the genes
found to be markers of FL distal mesenchymal cells, and the top 15 markers of
FL proximal mesenchymal cells. **c** Assignment of a proximal (dark blue) or distal
(yellow) identity to each cell of mouse and bat fore- and hindlimbs based on
*Hoxd13 + Msx1* and *Shox2* expression per cell. Shown are the differences in the
proportion of proximally or distally assigned cells per cluster. PD = Difference
of the expression score of the distal genes, minus the expression score of
the proximal gene. **d** Co-expression of distal autopodial marker *Hoxd13* and
chiropatagium marker *Meis2* in mouse and bat fore- and hindlimbs. Shown is the
fraction of cells co-expressing both genes per cluster. Cluster 7 FbIr is highlighted
with an arrow. **e** Marker genes of bat cluster 7 FbIr based on differential gene
expression between cluster 7 and the rest of the LPM-derived bat FL cells. **f** Bat
cluster 7 FbIr gene set expression in mouse and bat fore- and hindlimbs. Shown is
the fraction of co-expression score of the whole marker gene set shown in E.

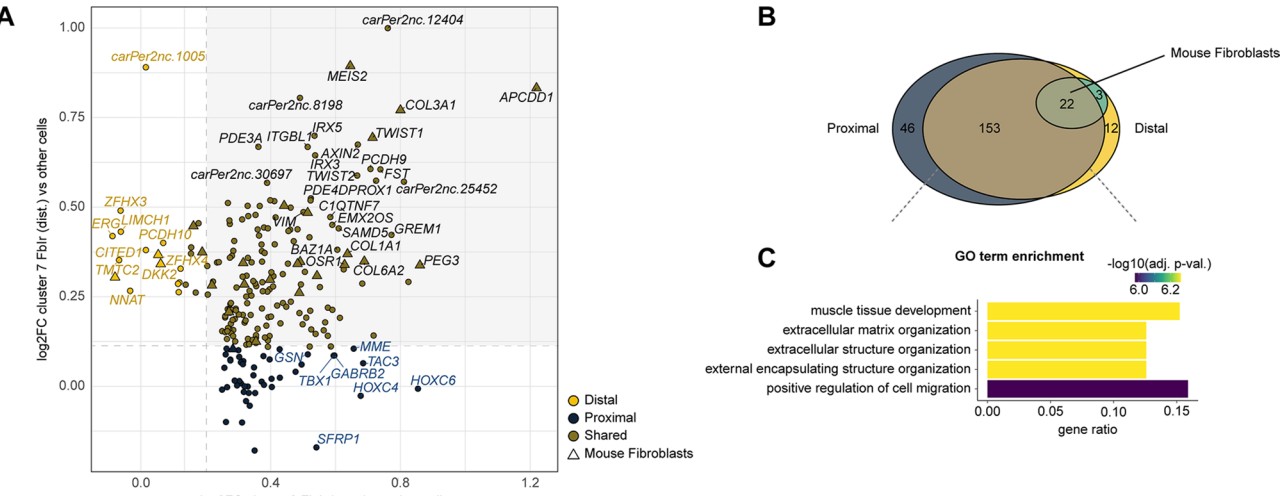

**Extended Data Fig. 6 | Second proximal gene program in the distal bat forelimb.** Related to Fig. 3a. **a** Correlation between the differential expression of genes from distal (cluster 7) and proximal (cluster 8) *MEIS2*-positive clusters in the bat forelimb identified in Extended Data Fig. 5. Shown is the logfold change (log2FC) of differential gene expression analysis of the respective cluster versus non-fibroblast LPM-derived cells (Supplementary Data 10). Genes shared with mouse fibroblasts are depicted as triangles. **b** Venn diagram showing the overlap (brown) between the genes enriched in the proximal (dark blue) and distal (yellow) cell subset as well as the fraction of genes shared with mouse fibroblasts (green). **c** GO term enrichment analysis of the 175 shared genes. Shown are the top 5 enriched GO terms. Over-representation analysis implemented in ClusterProfiler (see methods).

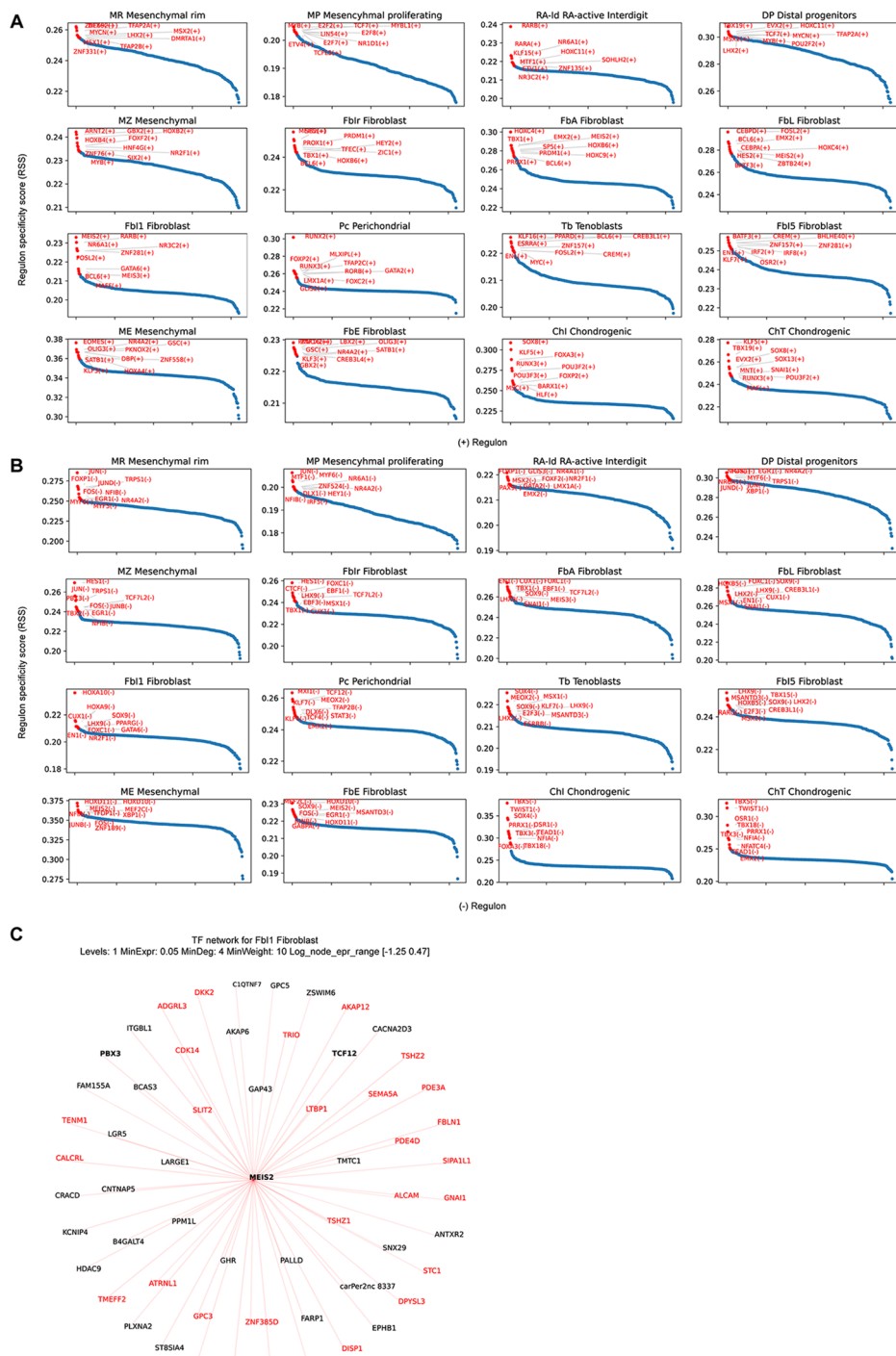

**Extended Data Fig. 7 | Gene regulatory network analyses.** Related to Fig. 3e. **a** and **b** Positive (A) and negative (B) regulons from the gene regulatory network analyses with SCENIC. **c** Transcription factor network showing downstream regulators for *MEIS2* in bat forelimb cluster 10 FbI1.

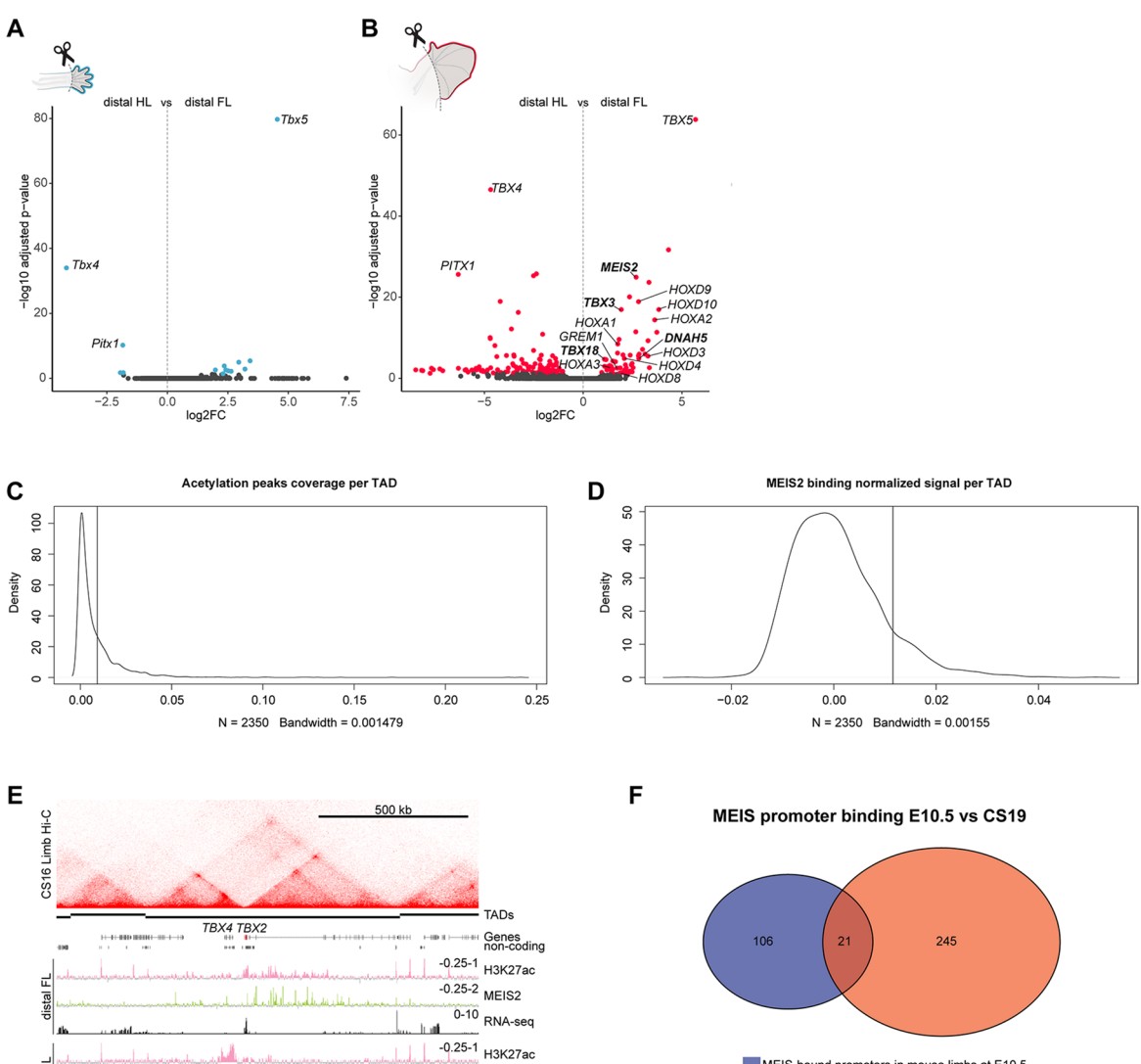

**Extended Data Fig. 8 | Epigenetic and transcriptomic bulk analyses.** Related to Fig. 3. **a** and **b** Volcano plot of the differentially expressed genes from bulk RNA-seq between the dissected distal FL and distal HL in mice (A) and in bats (B). **c** Distribution of the TADs in the Bat genome, and the fraction of each of them covered by H3K27ac ChIP peaks specific to the distal FL. The vertical line shows the cutoff used to determine the enriched TADs. **d** Distribution of the TADs in the Bat genome, and the normalized (input subtracted) signal from MEIS binding found within each of them. The vertical line shows the cutoff used to determine

the enriched TADs. **e** Bat *TBX4/TBX2* locus with Hi-C from CS16 FLs on top, TAD calling below. The Input subtracted H3K27ac ChIP-seq track is depicted in pink and input subtracted MEIS2 ChIP-seq track is shown in green. RNA-seq tracks are shown in black. ChIP-seq and RNA-seq were performed on distally dissected fore- and hindlimbs at CS18/19. Note the specific expression and acetylation of *TBX4* in the HL and the specific MEIS2 binding sites throughout the *TBX2* domain unique to the distal FL. **f** Venn diagram showing the overlap (21) of MEIS-bound promoters in mouse 10.5 limbs and bat CS19 distal limbs.

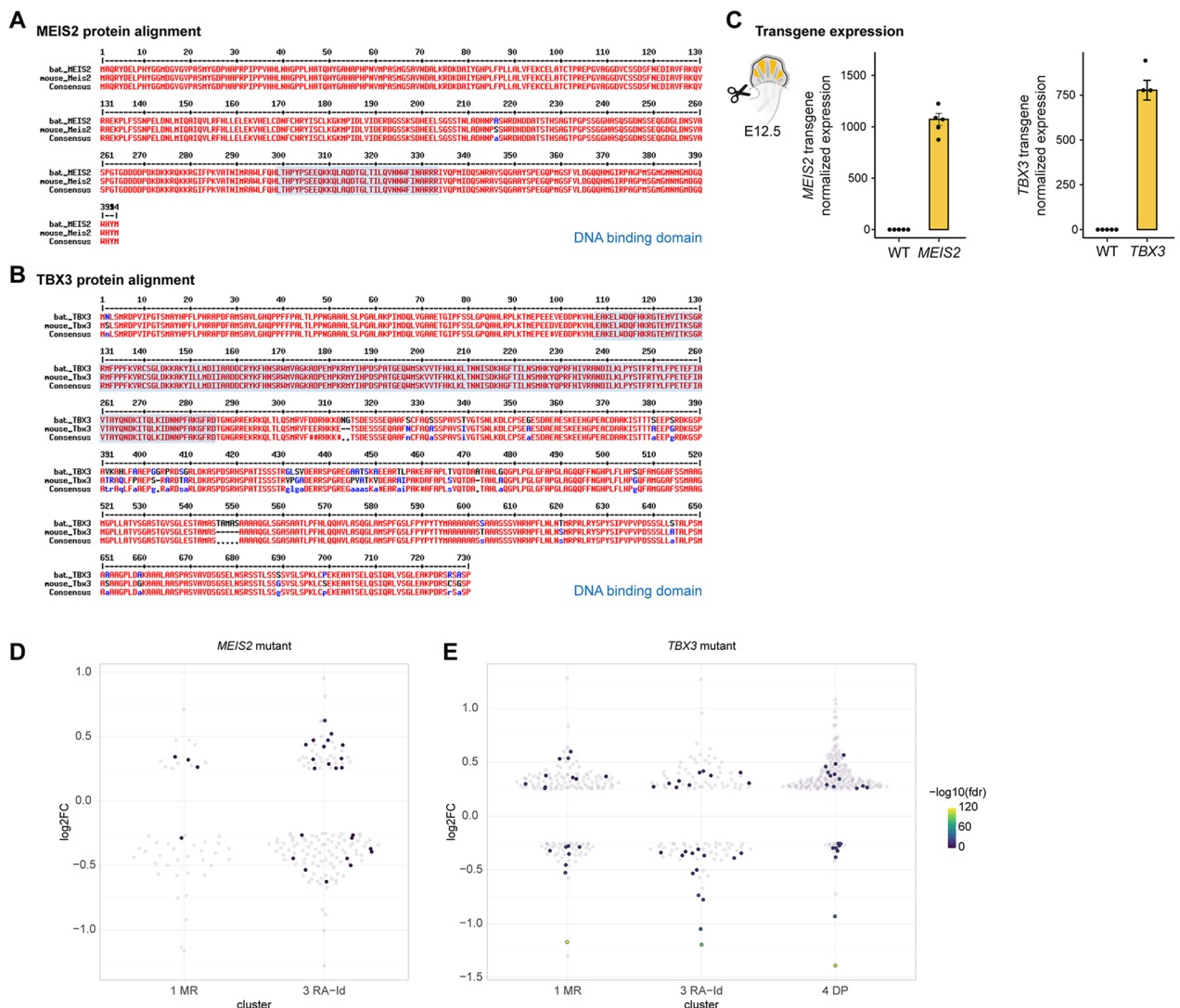

**Extended Data Fig. 9 | MEIS2 and TBX3 protein alignment *and* transgene expression of mutant limbs.** Related to Fig. 4. **a** and **b** Alignment of bat and mouse MEIS2 and TBX3 protein sequences. The DNA-binding domains are highlighted in blue. **c** RNA-seq of distally dissected wildtype and mutant limbs at E12.5 showing the normalized expression of MEIS2 and TBX3 transgenes. n = 5. *Error bars = standard deviation*. **d** and **e** Plots showing all the significantly differentially expressed genes (adjusted p-value < 0.01 & |log2FC| > 0.25) in the affected limb clusters of mouse mutant limbs at E12.5 (1 MR and 3 RA-Id in *MEIS2* mutant; 1 MR, 3 RA-Id and 4 DP in *TBX3* mutant). Differential expression tested using a likelihood-ratio test on a zero-inflated regression implemented in MAST (see methods). Genes from the chiropatagium gene program are highlighted.

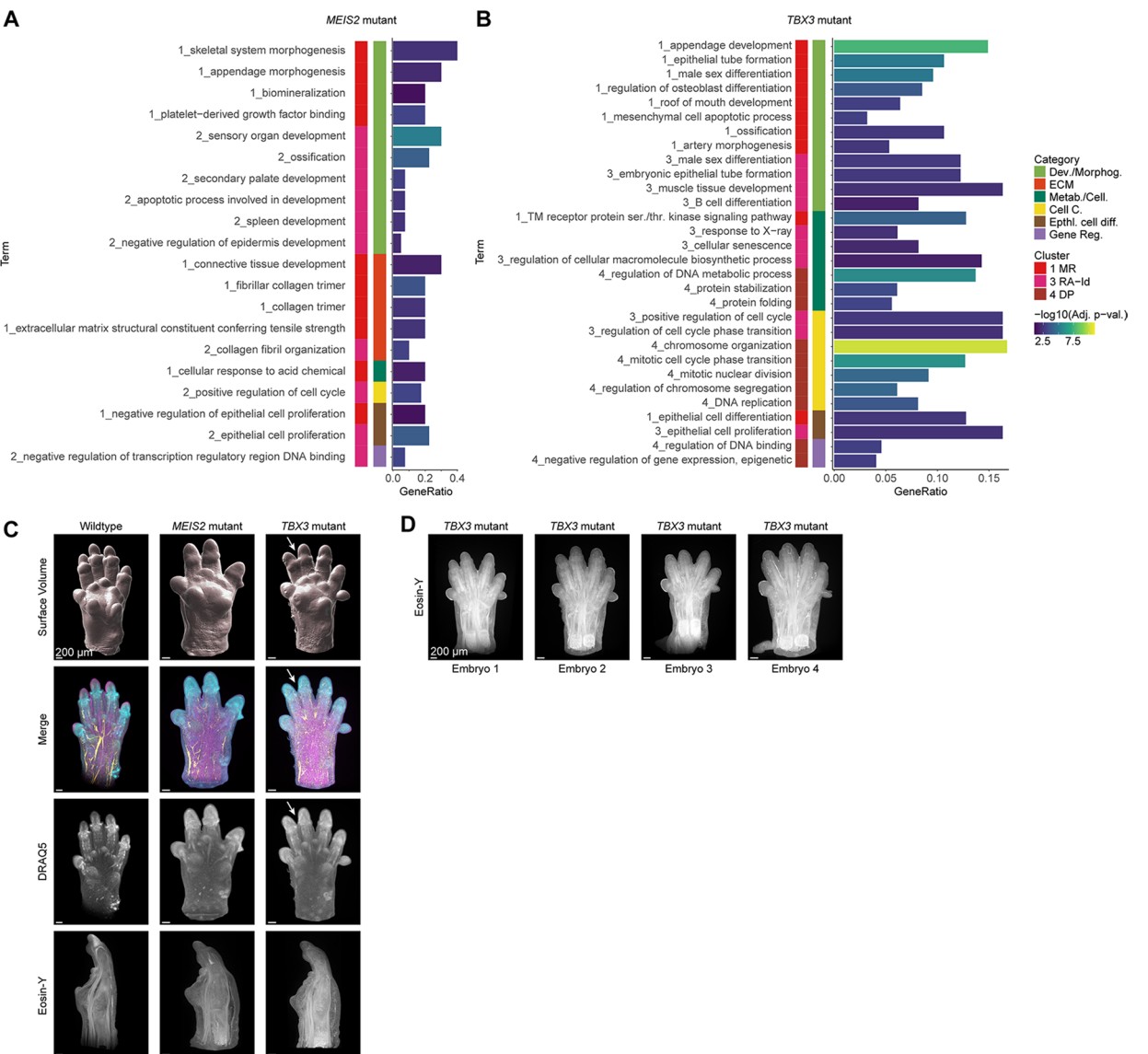

**Extended Data Fig. 10 | GO enrichment and 3D imaging of mutant limbs.**
Related to Fig. 4. **a** and **b** Barplots showing the proportion of GO-Terms categories
found upregulated in mutant mice. GO Term categories reflect biological
functions. Shown are the top 10 GO Terms for every affected cell cluster.
Over-representation analysis implemented in ClusterProfiler (see methods).

**c** 3D imaging of mouse wildtype and mutant whole limbs at E15.5. Shown is a
surface representation, Eosin-Y staining and a merged imaged with an arrow
highlighting syndactyly of digit II and III in the *TBX3* mutant. Magenta = eosin-Y,
Cyan = nuclei, Yellow = autofluorescence. *n* = 4. **d** Eosin-Y staining cross-section of
all *TBX3* mutant mice generated, all showing syndactyly. *n* = 4. Scale = 200 μm.

| | Corresponding author(s): | Darío Lupiáñez, Stefan Mundlos, & Francisca M. Real |
|---|---|---|
| | Last updated by author(s): | May 7, 2025 |

# Reporting Summary

## Statistics

For all statistical analyses, confirm that the following items are present in the figure legend, table legend, main text, or Methods section.

| n/a | Confirmed | |
|---|---|---|
| ☐ | ☒ | The exact sample size (*n*) for each experimental group/condition, given as a discrete number and unit of measurement |
| ☐ | ☒ | A statement on whether measurements were taken from distinct samples or whether the same sample was measured repeatedly |
| ☐ | ☒ | The statistical test(s) used AND whether they are one- or two-sided *Only common tests should be described solely by name; describe more complex techniques in the Methods section.* |
| ☐ | ☒ | A description of all covariates tested |
| ☐ | ☒ | A description of any assumptions or corrections, such as tests of normality and adjustment for multiple comparisons |
| ☐ | ☒ | A full description of the statistical parameters including central tendency (e.g. means) or other basic estimates (e.g. regression coefficient) AND variation (e.g. standard deviation) or associated estimates of uncertainty (e.g. confidence intervals) |
| ☐ | ☒ | For null hypothesis testing, the test statistic (e.g. *F*, *t*, *r*) with confidence intervals, effect sizes, degrees of freedom and *P* value noted *Give P values as exact values whenever suitable.* |
| ☐ | ☐ | For Bayesian analysis, information on the choice of priors and Markov chain Monte Carlo settings |
| ☐ | ☐ | For hierarchical and complex designs, identification of the appropriate level for tests and full reporting of outcomes |
| ☐ | ☐ | Estimates of effect sizes (e.g. Cohen's *d*, Pearson's *r*), indicating how they were calculated |

*Our web collection on statistics for biologists contains articles on many of the points above.*

## Software and code

Policy information about availability of computer code

| Data collection | *Provide a description of all commercial, open source and custom code used to collect the data in this study, specifying the version used OR state that no software was used.* |
|---|---|
| Data analysis | See methods |

For manuscripts utilizing custom algorithms or software that are central to the research but not yet described in published literature, software must be made available to editors and reviewers. We strongly encourage code deposition in a community repository (e.g. GitHub). See the Nature Portfolio guidelines for submitting code & software for further information.

## Data

Policy information about availability of data

All manuscripts must include a data availability statement. This statement should provide the following information, where applicable:
- Accession codes, unique identifiers, or web links for publicly available datasets
- A description of any restrictions on data availability
- For clinical datasets or third party data, please ensure that the statement adheres to our policy

Raw and processed functional data produced in this work have been deposited in the Gene Expression Omnibus (GEO) repositories: GSE275848, GSE275851, GSE275853, GSE275854, GSE275855. The genome assembly has been kindly provided by Prof. Michael Hiller and can be found here: https://genome.senckenberg.de/download/CarolliaData/.

# Research involving human participants, their data, or biological material

Policy information about studies with [human participants or human data](). See also policy information about [sex, gender (identity/presentation), and sexual orientation]() and [race, ethnicity and racism]().

| | |
|---|---|
| Reporting on sex and gender | *Use the terms sex (biological attribute) and gender (shaped by social and cultural circumstances) carefully in order to avoid confusing both terms. Indicate if findings apply to only one sex or gender; describe whether sex and gender were considered in study design; whether sex and/or gender was determined based on self-reporting or assigned and methods used. Provide in the source data disaggregated sex and gender data, where this information has been collected, and if consent has been obtained for sharing of individual-level data; provide overall numbers in this Reporting Summary. Please state if this information has not been collected. Report sex- and gender-based analyses where performed, justify reasons for lack of sex- and gender-based analysis.* |
| Reporting on race, ethnicity, or other socially relevant groupings | *Please specify the socially constructed or socially relevant categorization variable(s) used in your manuscript and explain why they were used. Please note that such variables should not be used as proxies for other socially constructed/relevant variables (for example, race or ethnicity should not be used as a proxy for socioeconomic status). Provide clear definitions of the relevant terms used, how they were provided (by the participants/respondents, the researchers, or third parties), and the method(s) used to classify people into the different categories (e.g. self-report, census or administrative data, social media data, etc.) Please provide details about how you controlled for confounding variables in your analyses.* |
| Population characteristics | *Describe the covariate-relevant population characteristics of the human research participants (e.g. age, genotypic information, past and current diagnosis and treatment categories). If you filled out the behavioural & social sciences study design questions and have nothing to add here, write "See above."* |
| Recruitment | *Describe how participants were recruited. Outline any potential self-selection bias or other biases that may be present and how these are likely to impact results.* |
| Ethics oversight | *Identify the organization(s) that approved the study protocol.* |

Note that full information on the approval of the study protocol must also be provided in the manuscript.

# Field-specific reporting

Please select the one below that is the best fit for your research. If you are not sure, read the appropriate sections before making your selection.

☒ Life sciences ☐ Behavioural & social sciences ☐ Ecological, evolutionary & environmental sciences

For a reference copy of the document with all sections, see [nature.com/documents/nr-reporting-summary-flat.pdf]()

# Life sciences study design

All studies must disclose on these points even when the disclosure is negative.

| | |
|---|---|
| Sample size | Single-cell Mouse limbs = 2 per limb, Bat limbs = 2 per limb, Chiropatagium = 2, Mutant lims = 2 forelimbs per mutant. ChIP-seq: 2 pools of several limbs from different individuals. RNA-seq: Mutants = 5 pools of limbs from different individuals, Bat dFL = 2 limbs, Bat dHL = 4 limbs, Mouse dFL = 4 limbs, Mouse dHL = 2 limbs. |
| Data exclusions | Single cells excluded as reported in methods. |
| Replication | *Describe the measures taken to verify the reproducibility of the experimental findings. If all attempts at replication were successful, confirm this OR if there are any findings that were not replicated or cannot be reproduced, note this and describe why.* |
| Randomization | *Describe how samples/organisms/participants were allocated into experimental groups. If allocation was not random, describe how covariates were controlled OR if this is not relevant to your study, explain why.* |
| Blinding | *Describe whether the investigators were blinded to group allocation during data collection and/or analysis. If blinding was not possible, describe why OR explain why blinding was not relevant to your study.* |

# Behavioural & social sciences study design

All studies must disclose on these points even when the disclosure is negative.

| | |
|---|---|
| Study description | *Briefly describe the study type including whether data are quantitative, qualitative, or mixed-methods (e.g. qualitative cross-sectional, quantitative experimental, mixed-methods case study).* |
| Research sample | *State the research sample (e.g. Harvard university undergraduates, villagers in rural India) and provide relevant demographic information (e.g. age, sex) and indicate whether the sample is representative. Provide a rationale for the study sample chosen. For* |

| | |
|---|---|
| | *studies involving existing datasets, please describe the dataset and source.* |
| Sampling strategy | *Describe the sampling procedure (e.g. random, snowball, stratified, convenience). Describe the statistical methods that were used to predetermine sample size OR if no sample-size calculation was performed, describe how sample sizes were chosen and provide a rationale for why these sample sizes are sufficient. For qualitative data, please indicate whether data saturation was considered, and what criteria were used to decide that no further sampling was needed.* |
| Data collection | *Provide details about the data collection procedure, including the instruments or devices used to record the data (e.g. pen and paper, computer, eye tracker, video or audio equipment) whether anyone was present besides the participant(s) and the researcher, and whether the researcher was blind to experimental condition and/or the study hypothesis during data collection.* |
| Timing | *Indicate the start and stop dates of data collection. If there is a gap between collection periods, state the dates for each sample cohort.* |
| Data exclusions | *If no data were excluded from the analyses, state so OR if data were excluded, provide the exact number of exclusions and the rationale behind them, indicating whether exclusion criteria were pre-established.* |
| Non-participation | *State how many participants dropped out/declined participation and the reason(s) given OR provide response rate OR state that no participants dropped out/declined participation.* |
| Randomization | *If participants were not allocated into experimental groups, state so OR describe how participants were allocated to groups, and if allocation was not random, describe how covariates were controlled.* |

# Ecological, evolutionary & environmental sciences study design

All studies must disclose on these points even when the disclosure is negative.

| | |
|---|---|
| Study description | *Briefly describe the study. For quantitative data include treatment factors and interactions, design structure (e.g. factorial, nested, hierarchical), nature and number of experimental units and replicates.* |
| Research sample | *Describe the research sample (e.g. a group of tagged Passer domesticus, all Stenocereus thurberi within Organ Pipe Cactus National Monument), and provide a rationale for the sample choice. When relevant, describe the organism taxa, source, sex, age range and any manipulations. State what population the sample is meant to represent when applicable. For studies involving existing datasets, describe the data and its source.* |
| Sampling strategy | *Note the sampling procedure. Describe the statistical methods that were used to predetermine sample size OR if no sample-size calculation was performed, describe how sample sizes were chosen and provide a rationale for why these sample sizes are sufficient.* |
| Data collection | *Describe the data collection procedure, including who recorded the data and how.* |
| Timing and spatial scale | *Indicate the start and stop dates of data collection, noting the frequency and periodicity of sampling and providing a rationale for these choices. If there is a gap between collection periods, state the dates for each sample cohort. Specify the spatial scale from which the data are taken* |
| Data exclusions | *If no data were excluded from the analyses, state so OR if data were excluded, describe the exclusions and the rationale behind them, indicating whether exclusion criteria were pre-established.* |
| Reproducibility | *Describe the measures taken to verify the reproducibility of experimental findings. For each experiment, note whether any attempts to repeat the experiment failed OR state that all attempts to repeat the experiment were successful.* |
| Randomization | *Describe how samples/organisms/participants were allocated into groups. If allocation was not random, describe how covariates were controlled. If this is not relevant to your study, explain why.* |
| Blinding | *Describe the extent of blinding used during data acquisition and analysis. If blinding was not possible, describe why OR explain why blinding was not relevant to your study.* |

Did the study involve field work? ☐ Yes ☒ No

# Reporting for specific materials, systems and methods

We require information from authors about some types of materials, experimental systems and methods used in many studies. Here, indicate whether each material, system or method listed is relevant to your study. If you are not sure if a list item applies to your research, read the appropriate section before selecting a response.

## Materials & experimental systems

| n/a | Involved in the study |
|---|---|
| ☐ | ☒ Antibodies |
| ☐ | ☒ Eukaryotic cell lines |
| ☐ | ☐ Palaeontology and archaeology |
| ☐ | ☒ Animals and other organisms |
| ☐ | ☐ Clinical data |
| ☐ | ☐ Dual use research of concern |
| ☐ | ☐ Plants |

## Methods

| n/a | Involved in the study |
|---|---|
| ☐ | ☒ ChIP-seq |
| ☐ | ☐ Flow cytometry |
| ☐ | ☐ MRI-based neuroimaging |

## Antibodies

| Antibodies used | H3K27ac, anti-MEISa, anti-MEIS2 |
|---|---|
| Validation | Previously published and commercial antibodies |

## Eukaryotic cell lines

Policy information about cell lines and Sex and Gender in Research

| Cell line source(s) | Primary C. Perspicillata embryonic fibroblast and Mouse embryonic stem cells (G4) |
|---|---|
| Authentication | Bat cells are not authenticated, Mouse cells are commercially available. |
| Mycoplasma contamination | Bat cells are not tested, Mouse ESC tested negative. |
| Commonly misidentified lines (See ICLAC register) | *Name any commonly misidentified cell lines used in the study and provide a rationale for their use.* |

## Palaeontology and Archaeology

| Specimen provenance | *Provide provenance information for specimens and describe permits that were obtained for the work (including the name of the issuing authority, the date of issue, and any identifying information). Permits should encompass collection and, where applicable, export.* |
|---|---|
| Specimen deposition | *Indicate where the specimens have been deposited to permit free access by other researchers.* |
| Dating methods | *If new dates are provided, describe how they were obtained (e.g. collection, storage, sample pretreatment and measurement), where they were obtained (i.e. lab name), the calibration program and the protocol for quality assurance OR state that no new dates are provided.* |

☐ Tick this box to confirm that the raw and calibrated dates are available in the paper or in Supplementary Information.

| Ethics oversight | *Identify the organization(s) that approved or provided guidance on the study protocol, OR state that no ethical approval or guidance was required and explain why not.* |
|---|---|

Note that full information on the approval of the study protocol must also be provided in the manuscript.

## Animals and other research organisms

Policy information about studies involving animals; ARRIVE guidelines recommended for reporting animal research, and Sex and Gender in Research

| Laboratory animals | Mus musculus, Carollia Perspicillata |
|---|---|
| Wild animals | *Provide details on animals observed in or captured in the field; report species and age where possible. Describe how animals were caught and transported and what happened to captive animals after the study (if killed, explain why and describe method; if released, say where and when) OR state that the study did not involve wild animals.* |
| Reporting on sex | No |
| Field-collected samples | *For laboratory work with field-collected samples, describe all relevant parameters such as housing, maintenance, temperature, photoperiod and end-of-experiment protocol OR state that the study did not involve samples collected from the field.* |

| Ethics oversight | LAGeSo, RP Darmstadt |

Note that full information on the approval of the study protocol must also be provided in the manuscript.

# Clinical data

Policy information about clinical studies

All manuscripts should comply with the ICMJE guidelines for publication of clinical research and a completed CONSORT checklist must be included with all submissions.

| Clinical trial registration | *Provide the trial registration number from ClinicalTrials.gov or an equivalent agency.* |
| Study protocol | *Note where the full trial protocol can be accessed OR if not available, explain why.* |
| Data collection | *Describe the settings and locales of data collection, noting the time periods of recruitment and data collection.* |
| Outcomes | *Describe how you pre-defined primary and secondary outcome measures and how you assessed these measures.* |

# Dual use research of concern

Policy information about dual use research of concern

## Hazards

Could the accidental, deliberate or reckless misuse of agents or technologies generated in the work, or the application of information presented in the manuscript, pose a threat to:

No | Yes

☒ ☐ Public health

☒ ☐ National security

☒ ☐ Crops and/or livestock

☒ ☐ Ecosystems

☒ ☐ Any other significant area

## Experiments of concern

Does the work involve any of these experiments of concern:

No | Yes

☒ ☐ Demonstrate how to render a vaccine ineffective

☒ ☐ Confer resistance to therapeutically useful antibiotics or antiviral agents

☒ ☐ Enhance the virulence of a pathogen or render a nonpathogen virulent

☒ ☐ Increase transmissibility of a pathogen

☒ ☐ Alter the host range of a pathogen

☒ ☐ Enable evasion of diagnostic/detection modalities

☒ ☐ Enable the weaponization of a biological agent or toxin

☒ ☐ Any other potentially harmful combination of experiments and agents

# Plants

| Seed stocks | *Report on the source of all seed stocks or other plant material used. If applicable, state the seed stock centre and catalogue number. If plant specimens were collected from the field, describe the collection location, date and sampling procedures.* |
| Novel plant genotypes | *Describe the methods by which all novel plant genotypes were produced. This includes those generated by transgenic approaches, gene editing, chemical/radiation-based mutagenesis and hybridization. For transgenic lines, describe the transformation method, the number of independent lines analyzed and the generation upon which experiments were performed. For gene-edited lines, describe the editor used, the endogenous sequence targeted for editing, the targeting guide RNA sequence (if applicable) and how the editor was applied.* |
| Authentication | *Describe any authentication procedures for each seed stock used or novel genotype generated. Describe any experiments used to assess the effect of a mutation and, where applicable, how potential secondary effects (e.g. second site T-DNA insertions, mosiacism, off-target gene editing) were examined.* |

# ChIP-seq

## Data deposition

☒ Confirm that both raw and final processed data have been deposited in a public database such as GEO.

☒ Confirm that you have deposited or provided access to graph files (e.g. BED files) for the called peaks.

| | |
|---|---|
| Data access links<br>*May remain private before publication.* | https://www.ncbi.nlm.nih.gov/geo/query/acc.cgi?acc=GSE275851 |
| Files in database submission | Meis2_D-FL_CS19_carPer2_WT_Rep2_L30879_S23_R1_001.fastq.gz<br>Meis2_D-FL_CS19_carPer2_WT_Rep2_L30879_S23_R2_001.fastq.gz<br>Meis2_D-FL_CS19_carPer2_WT_Rep3_L31146_S5_R1_001.fastq.gz<br>Meis2_D-FL_CS19_carPer2_WT_Rep3_L31146_S5_R2_001.fastq.gz<br>Meis2_D-HL_CS19_carPer2_WT_Rep2_L30881_S25_R1_001.fastq.gz<br>Meis2_D-HL_CS19_carPer2_WT_Rep2_L30881_S25_R2_001.fastq.gz<br>Meis2_D-HL_CS19_carPer2_WT_Rep3_L31148_S7_R1_001.fastq.gz<br>Meis2_D-HL_CS19_carPer2_WT_Rep3_L31148_S7_R2_001.fastq.gz<br>Meis2_Input_D-FL_CS19_carPer2_WT_Rep2_L30875_S19_R1_001.fastq.gz<br>Meis2_Input_D-FL_CS19_carPer2_WT_Rep2_L30875_S19_R2_001.fastq.gz<br>Meis2_Input_D-FL_CS19_carPer2_WT_Rep3_L31150_S9_R1_001.fastq.gz<br>Meis2_Input_D-FL_CS19_carPer2_WT_Rep3_L31150_S9_R2_001.fastq.gz<br>Meis2_Input_D-HL_CS19_carPer2_WT_Rep2_L30877_S21_R1_001.fastq.gz<br>Meis2_Input_D-HL_CS19_carPer2_WT_Rep2_L30877_S21_R2_001.fastq.gz<br>Meis2_Input_D-HL_CS19_carPer2_WT_Rep3_L31152_S11_R1_001.fastq.gz<br>Meis2_Input_D-HL_CS19_carPer2_WT_Rep3_L31152_S11_R2_001.fastq.gz<br>H3K27ac_D-FL_CS19_carPer2_WT_Rep1_L30030_S61_R1_001.fastq.gz<br>H3K27ac_D-FL_CS19_carPer2_WT_Rep1_L30030_S61_R2_001.fastq.gz<br>H3K27ac_D-FL_CS19_carPer2_WT_Rep2_L30038_S69_R1_001.fastq.gz<br>H3K27ac_D-FL_CS19_carPer2_WT_Rep2_L30038_S69_R2_001.fastq.gz<br>H3K27ac_D-HL_CS19_carPer2_WT_Rep1_L30032_S63_R1_001.fastq.gz<br>H3K27ac_D-HL_CS19_carPer2_WT_Rep1_L30032_S63_R2_001.fastq.gz<br>H3K27ac_D-HL_CS19_carPer2_WT_Rep2_L30040_S71_R1_001.fastq.gz<br>H3K27ac_D-HL_CS19_carPer2_WT_Rep2_L30040_S71_R2_001.fastq.gz<br>Input_D-FL_CS19_carPer2_WT_Rep1_L30288_S21_R1_001.fastq.gz<br>Input_D-FL_CS19_carPer2_WT_Rep1_L30288_S21_R2_001.fastq.gz<br>Input_D-FL_CS19_carPer2_WT_Rep2_L30292_S25_R1_001.fastq.gz<br>Input_D-FL_CS19_carPer2_WT_Rep2_L30292_S25_R2_001.fastq.gz<br>Input_D-HL_CS19_carPer2_WT_Rep1_L30290_S23_R1_001.fastq.gz<br>Input_D-HL_CS19_carPer2_WT_Rep1_L30290_S23_R2_001.fastq.gz<br>Input_D-HL_CS19_carPer2_WT_Rep2_L30294_S27_R1_001.fastq.gz<br>Input_D-HL_CS19_carPer2_WT_Rep2_L30294_S27_R2_001.fastq.gz<br>H3K27ac_D-FL_CS19_carPer2_WT_merged_-IP.bw<br>H3K27ac_D-HL_CS19_carPer2_WT_merged_-IP.bw<br>Meis2_D-FL_CS19_carPer2_WT_merged-IP.bw<br>Meis2_D-HL_CS19_carPer2_WT_-IP_merged.bw<br>Meis2_carPer_CS19_dist_forelimb.narrowPeak<br>Meis2_carPer_CS19_dist_hindlimb.narrowPeak<br>diff_dFL_vs_dHL_common.bed<br>diff_dFL_vs_dHL_dFL.bed<br>diff_dFL_vs_dHL_dHL.bed |
| Genome browser session<br>(e.g. UCSC) | *Provide a link to an anonymized genome browser session for "Initial submission" and "Revised version" documents only, to enable peer review. Write "no longer applicable" for "Final submission" documents.* |

## Methodology

| | |
|---|---|
| Replicates | 2 |
| Sequencing depth | 50 Million pair end reads |
| Antibodies | H3K27ac, anti-MEISa, anti-MEIS2 |
| Peak calling parameters | Described in the methods |
| Data quality | *Describe the methods used to ensure data quality in full detail, including how many peaks are at FDR 5% and above 5-fold enrichment.* |
| Software | MACS2 |

# Flow Cytometry

## Plots

Confirm that:

☐ The axis labels state the marker and fluorochrome used (e.g. CD4-FITC).

☐ The axis scales are clearly visible. Include numbers along axes only for bottom left plot of group (a 'group' is an analysis of identical markers).

☐ All plots are contour plots with outliers or pseudocolor plots.

☐ A numerical value for number of cells or percentage (with statistics) is provided.

## Methodology

| | |
|---|---|
| Sample preparation | *Describe the sample preparation, detailing the biological source of the cells and any tissue processing steps used.* |
| Instrument | *Identify the instrument used for data collection, specifying make and model number.* |
| Software | *Describe the software used to collect and analyze the flow cytometry data. For custom code that has been deposited into a community repository, provide accession details.* |
| Cell population abundance | *Describe the abundance of the relevant cell populations within post-sort fractions, providing details on the purity of the samples and how it was determined.* |
| Gating strategy | *Describe the gating strategy used for all relevant experiments, specifying the preliminary FSC/SSC gates of the starting cell population, indicating where boundaries between "positive" and "negative" staining cell populations are defined.* |

☐ Tick this box to confirm that a figure exemplifying the gating strategy is provided in the Supplementary Information.

# Magnetic resonance imaging

## Experimental design

| | |
|---|---|
| Design type | *Indicate task or resting state; event-related or block design.* |
| Design specifications | *Specify the number of blocks, trials or experimental units per session and/or subject, and specify the length of each trial or block (if trials are blocked) and interval between trials.* |
| Behavioral performance measures | *State number and/or type of variables recorded (e.g. correct button press, response time) and what statistics were used to establish that the subjects were performing the task as expected (e.g. mean, range, and/or standard deviation across subjects).* |

## Acquisition

| | |
|---|---|
| Imaging type(s) | *Specify: functional, structural, diffusion, perfusion.* |
| Field strength | *Specify in Tesla* |
| Sequence & imaging parameters | *Specify the pulse sequence type (gradient echo, spin echo, etc.), imaging type (EPI, spiral, etc.), field of view, matrix size, slice thickness, orientation and TE/TR/flip angle.* |
| Area of acquisition | *State whether a whole brain scan was used OR define the area of acquisition, describing how the region was determined.* |

Diffusion MRI     ☐ Used     ☐ Not used

## Preprocessing

| | |
|---|---|
| Preprocessing software | *Provide detail on software version and revision number and on specific parameters (model/functions, brain extraction, segmentation, smoothing kernel size, etc.).* |
| Normalization | *If data were normalized/standardized, describe the approach(es): specify linear or non-linear and define image types used for transformation OR indicate that data were not normalized and explain rationale for lack of normalization.* |
| Normalization template | *Describe the template used for normalization/transformation, specifying subject space or group standardized space (e.g. original Talairach, MNI305, ICBM152) OR indicate that the data were not normalized.* |
| Noise and artifact removal | *Describe your procedure(s) for artifact and structured noise removal, specifying motion parameters, tissue signals and physiological signals (heart rate, respiration).* |

| Volume censoring | *Define your software and/or method and criteria for volume censoring, and state the extent of such censoring.* |

## Statistical modeling & inference

| Model type and settings | *Specify type (mass univariate, multivariate, RSA, predictive, etc.) and describe essential details of the model at the first and second levels (e.g. fixed, random or mixed effects; drift or auto-correlation).* |

| Effect(s) tested | *Define precise effect in terms of the task or stimulus conditions instead of psychological concepts and indicate whether ANOVA or factorial designs were used.* |

Specify type of analysis: ☐ Whole brain ☐ ROI-based ☐ Both

| Statistic type for inference | *Specify voxel-wise or cluster-wise and report all relevant parameters for cluster-wise methods.* |

(See Eklund et al. 2016)

| Correction | *Describe the type of correction and how it is obtained for multiple comparisons (e.g. FWE, FDR, permutation or Monte Carlo).* |

## Models & analysis

| n/a | Involved in the study |
| --- | --- |
| ☐ | ☐ Functional and/or effective connectivity |
| ☐ | ☐ Graph analysis |
| ☐ | ☐ Multivariate modeling or predictive analysis |

| Functional and/or effective connectivity | *Report the measures of dependence used and the model details (e.g. Pearson correlation, partial correlation, mutual information).* |

| Graph analysis | *Report the dependent variable and connectivity measure, specifying weighted graph or binarized graph, subject- or group-level, and the global and/or node summaries used (e.g. clustering coefficient, efficiency, etc.).* |

| Multivariate modeling and predictive analysis | *Specify independent variables, features extraction and dimension reduction, model, training and evaluation metrics.* |

