## [Peer Review File · Nature Ecology & Evolution]

Comparative single-cell analyses reveal evolutionary repurposing of a conserved gene program in bat wing development.

Corresponding Author: Dr Francisca M. Real

Version 0:

Decision Letter:

12th February 2025

Dear Francisca,

Your manuscript entitled "Comparative single-cell analyses reveal evolutionary repurposing of a conserved gene program in bat wing development." has now been seen by three reviewers, whose comments are attached. The reviewers have raised a number of concerns which will need to be addressed before we can offer publication in Nature Ecology & Evolution. We will therefore need to see your responses to the criticisms raised and to some editorial concerns, along with a revised manuscript, before we can reach a final decision regarding publication.

We therefore invite you to revise your manuscript taking into account all reviewer and editor comments. Please highlight all changes in the manuscript text file in Microsoft Word format.

* If you have not done so already please begin to revise your manuscript so that it conforms to our Article format instructions at <http://www.nature.com/natecolevol/info/final-submission>. Refer also to any guidelines provided in this letter.

* Extended Data Figures - please ensure that any supplementary figures and tables that are crucial to the manuscript's conclusions are converted into Extended Data figures and tables to increase visibility of these data. Extended Data figures and tables are online-only (present in the online PDF and full-text HTML versions of the paper), peer-reviewed display items that provide essential background to the article but are not included in the main article due to space constraints. A maximum of ten Extended Data display items (figures and tables) is permitted.

Link Redacted

We hope to receive your revised manuscript within four to eight weeks. If you cannot send it within this time, please let us

know. We will be happy to consider your revision so long as nothing similar has been accepted for publication at Nature Ecology & Evolution or published elsewhere.

Nature Ecology & Evolution is committed to improving transparency in authorship. As part of our efforts in this direction, we are now requesting that all authors identified as 'corresponding author' on published papers create and link their Open Researcher and Contributor Identifier (ORCID) with their account on the Manuscript Tracking System (MTS), prior to acceptance. ORCID helps the scientific community achieve unambiguous attribution of all scholarly contributions. You can create and link your ORCID from the home page of the MTS by clicking on 'Modify my Springer Nature account'. For more information please visit www.springernature.com/orcid.

[redacted]

Reviewer expertise:

Reviewer #1: bioinformatics, single cell comparative analysis

Reviewer #2: vertebrate evo-devo

Reviewer #3: vertebrate limb development

Reviewers' comments:

Reviewer #1 (Remarks to the Author):

Mechanisms of chiropatagium development in bats are unknown and the authors presented a thorough investigation to provide cellular and molecular explanations. Utilizing single-cell RNA-seq and cross-species comparisons, the author found that chiropatagium cells have a unique trajectory characterized by MEIS2 and TBX3 expression, which is also shown to be a repurposed proximal gene program. Further omics and experimental validations were performed to support this finding. Overall, this study is high in novelty and would be of interest to researchers in various areas of Evo-devo. I appreciate the author's collaborative effort in combing through diverse approaches to pursue a hypothesis extensively. My main concerns lie in the correctness of the theories and methods referred to; the clarity of some analyses; the structure and logic of the manuscript; and the accuracy of some statements.

The correctness of theories and methods referred to:

1. In line 108, please note that the SCTransform algorithm is not an integration tool, but a normalization method (see DOI: <https://doi.org/10.1186/s13059-019-1874-1>). Further in line 891 in methods, IntegrateData is also not an integration tool but a function to apply an integration method of choice. Please check which integration method was used in this function during the analysis and correct it accordingly.
2. The statement in line 119 can be misleading. I found out in the figure legends that Fig. 1E shows genes used to annotate the integrated cluster, rather than the top markers calculated based on integrated cross-species data. It would be more correct to say that e.g. "Expression of marker genes used for cluster annotation is shown in Fig. 1E, from which we confirmed their conserved specific expression patterns between species".
3. The term "deep-homology" is used to describe the conservation of gene expression patterns in non-homologous structures. However, mouse forelimbs and bat wings both derive from the forelimbs of a shared tetrapod ancestor, and are therefore homologous tissues. The reference to deep homology as an explanation for cell type conservation, in this case, seems invalid. In contrast, the idea of TF repurposing is highly valid and could be expanded further. Therefore, could the authors please reformulate the relevant discussions (lines 397-401).

Clarity of some analysis:

4. The orthology mapping between bat genes and mouse genes is an essential step for the integration analysis and its quality can greatly impact the output. However, the details about how this mapping has been done in the main text or methods seem missing. I saw that the authors annotated the bat genome with the human genome, but how is the orthology mapping to the mouse done? Please provide the bat-mouse orthology mapping table as supplementary data. In methods, please indicate how bat-mouse orthologs were called for cross-species integration. Were only one-to-one orthologs used, or non-one-to-one orthologs were also included in some way?
5. It is not clear why both human reference genomes hg19 and hg38 are used during *C. perspicillata* genome annotation (see line 818 and line 826). Could the authors please verify?
6. The sentence in line 134 could be further developed. It is highly relevant where the *Grem1* gene is actually expressed, since it is a known gene specific to the bat interdigital tissues. Adding an additional figure to show the expression pattern of *Grem1* gene in each cluster for both species will be helpful.
7. Based on the correlation heatmap shown in Fig. S3 C, D, I would say that many clusters have a group-level

correspondence instead of a one-to-one correspondence, as indicated by blocks spanning a few clusters on the heatmap. This could be biological, as studies have shown that cell types might exhibit strong group-level conservation, especially at such high granularity. However, this could also be because using only the top 10 marker genes reduced the sensitivity. Could the authors please try to use more marker genes to perform this correlation? Based on the results, please modify the statements about cross-species cluster conservation accordingly.

Structure and logic of the manuscript:

8. The introduction part (under "Main Text") seems to lack a clearly defined thesis statement for this paper. This is to propose the main research question and briefly outline the main work done. While I see that alike statements are positioned at the beginning of discussions, it will be much clearer to put them at the end of the introduction. Instead, the discussions can focus on the implications of these new findings instead of the necessity of the research.

9. Furthermore, to improve the clarity and logic of the introduction, the second paragraph could be split into two paragraphs at line 87. The former part is to explain the developmental process of the limb, while the latter addresses the hypothesis and brings out the key research question.

Accuracy and necessary explanations:

10. In line 97, single-cell approaches can be applied to "any" organism is an overstatement and "many" would be more appropriate.

11. A few suggestions on the use of acronyms:

1) Clusters in scRNA-seq data can be better named and consistently referred to. For instance, the cluster of interest can be named "3 RA-Id" rather than "RA-Id cluster 3" or "RA-Id cluster", which can cause confusion such as in line 131 vs line 179 vs line 193.

2) The log fold change value of gene expression has a canonical acronym as "log2FC", instead of "lfc". These two acronyms are also used in a mixture throughout the manuscript and please harmonize them.

3) Please always include an explanation of acronyms used in Figures at the end of the figure legends. Many might seem self-explanatory but it is good practice to always clearly state this. For instance, what does "PD" mean in Fig. 2P?

12. In line 186, slingshot could use a citation. In line 257 SCENIC could use a citation.

13. In line 267, could the authors please add "in bat FL" after the "most upregulated genes" so it's easier for the reader to follow the fact that several proximal markers are elevated in bat FL.

14. Could the authors please mark genes from the chiroptagium gene program to Figure 4 C to make it easier to understand? From my understanding, not all genes in the x-axis are associated with this program and it's possible to note which genes are.

15. What do the values in the pie chart of GO enrichment stand for in Fig. 4D? Using such a Pie chart to show GO can be confusing as it's fairly unconventional. There are many standardized representations of GO enrichment results such as dot heatmaps or bar heatmaps (such as in Figure 3 D), and I suggest remaking the plot to these forms which could improve clarity.

16. P-values in Fig. 4 I-K are not explained and please add details of the statistical tests to the legend.

17. Gene names are not in italic font in the methods, please format them like in the main text.

Reviewer #2 (Remarks to the Author):

The manuscript by Schindler et al., "Comparative single-cell analyses reveal evolutionary repurposing of a conserved gene program in bat wing development", explores the development of the evolutionary innovation that allows for powered flight in bats, the chiroptagium. The authors argue that, because there is a transcriptionally similar population of apoptotic cells in all the mouse and bat limbs they sampled, that the chiroptagium is not the result of an exceptional reduction in apoptosis in the bat forelimb. They present a compelling case for a bat-forelimb-specific population of distal fibroblasts in the chiroptagium. Using mouse transgenics, they show that MEIS2 and TBX3 can partially induce a chiroptagium-like genetic signature. Thus, the authors argue for a mechanism by which a set of genes from the proximal limb are redeployed in service of producing an evolutionary novelty in the distal limb.

Overall, I think the authors have done a nice job with this paper: it is clearly written and the figures are well constructed. The methods appear to be thorough. The authors have marshaled a large amount of data here, and they use it to tell a compelling story that will substantively contribute to the literature on deep homology and evolutionary novelty. Most of my concerns are minor.

Comments:

Missing Ns: It isn't clear from the descriptions in the results, figure captions, and methods, how many embryos contributed cells to the pools in the scRNA-seq experiments, or to the bulk transcriptomic and epigenomic experiments. Fig. S1A-C shows the scRNA-seq samples, but I couldn't easily find the others. Similarly, on line 339 the authors state that 'all TBX3 mutants displayed fusion of at least two digits.' How many mutants were produced?

Fig. 1G and S2A: What threshold do the dashed lines represent? Should be stated in the legend.

Fig. 1H, S2B, 4H, S9D: The figure panels or legends should indicate what the colors in the merged micrographs represent.

Fig. S3E–H: The x-axis label text is stretched and difficult to read.

Fig. S3B, D: Curious what's going on with the hindlimb RA-id cluster in the bat. From these two panels, it looks like there is no clear RA-id cluster in the bat hindlimb; expression of the marker genes from the mouse RA-id cluster appear to cluster with that of a few other clusters (Fig. S3D), including those of two MR clusters. This is confusing because, in Fig. S2A, a comparison is shown between genes expressed in the bat and mouse hindlimb RA-id clusters. Is the bat HL RA-id I understand that the hindlimb is not the focus of the paper, but it seems like an important comparison to the forelimb, and has been set up as such in Fig S2.

Typo: In Fig. 2S, 'Maker' should be 'Marker'

In my PDF, the bottom of the RNAseq track in Fig. 3J is partially cut off.

Line : Could the choice to proceed with functional test of TBX3 be clarified? The authors say that 'The striking pattern of chromatin activity profiles (H3K27ac and MEIS2 binding) being constrained within regulatory domains is exemplified for the TBX3 domain (Fig. 3J, and TBX2 in Fig. S8).' Indeed, the tracks in Fig. 3J and S8E show an enrichment of MEIS2 binding and H3K27Ac within these TADs, but the reader is left wondering why the authors didn't test TBX2, or any of the other TFs with strong coincident acetylation and MEIS2 binding.

Fig. 4E: What do the dashed lines represent? Should be explained in the legend.

Fig. 4G, S9D: The images of the Meis2 mutant show a reduced digit number, while those of the Tbx3 mutant indicate a possible extra digit, suggesting non-specific patterning effects of the transgene expression. Should we worry that the phenotypic effects in these mutants, especially the syndactyly, is more of a patterning issue and less of a chiropatagium-like phenotype?

Line 1115: ATAC-seq is described in the methods but I didn't see ATAC-seq in the results; did I miss it?

Reviewer #3 (Remarks to the Author):

The molecular mechanisms underlying chiropatagium development in bat forewings remain poorly understood. Schindler, Feregrino et al. investigated this question using comparative single-cell RNA sequencing of bat wings and mouse limbs. Contrary to the prevailing hypothesis that interdigital tissue persistence is due to anti-apoptotic mechanisms, the authors observed that apoptosis in bat forewings proceeds similarly to mouse limbs and bat hindlimbs. They identified a unique population of fibroblasts in bat forelimbs, characterized by expression of Meis2 and Tbx3 transcription factors, which are associated with proximal limb identity. Further chromatin and regulatory network analysis suggest that these factors play a key role in driving fibroblast differentiation in bat forewings. To functionally test this hypothesis, the authors generated transgenic mice expressing Meis2 and Tbx3 in distal interdigital tissue. The resulting phenotype exhibited moderate digit fusion (syndactyly) and increased ECM deposition, suggesting that Meis2/Tbx3 activation alters distal limb cell identity and contributes to interdigital tissue retention. Based on these findings the authors conclude that bat chiropatagium formation is driven by the spatial repurposing of a proximally active gene program. While the findings and differences described within this manuscript are not entirely novel and rather represent an incremental advance utilizing single-cell analysis, the manuscript itself is well-written, and the single-cell data analysis is both convincing and a valuable resource for researchers in the field. However, some conclusions appear overinterpreted or lack sufficient explanation. A more nuanced framing with alternative interpretations would strengthen the study. Here, I provide the authors constructive feedback, comments, and suggestions to incorporate into their revised manuscript.

The authors state in lines 121-123: "Using this inter-species single-cell atlas, we first sought to address the prevailing hypothesis that chiropatagium development is driven by inhibition or reduction of apoptotic cell death in the interdigital tissue."

They conclude in lines 166-167: "Since cell death occurs similarly in both bat and mouse interdigital clusters, it cannot account for the persistence of interdigital tissue."

Throughout...the manuscript implies that apoptosis suppression does not play a significant role in chiropatagium development. However, the provided evidence does not definitively exclude apoptosis suppression as a potentially important mechanism (among others) for interdigital tissue retention.

1. The forelimb and forewing, at E13.5 and CS17, both exhibit slightly regressed and curved interdigital tissue at the distal periphery (e.g., Mason et al., 2015). LysoTracker/Caspase images reveal that the apoptotic domains at the distal end in bats are comparable to those in mice. However, LysoTracker and Caspase staining in mice show an additional apoptotic domain in the proximal interdigit (see e.g., Kaltcheva et al., 2016). Therefore, an apoptosis-suppressing mechanism (e.g., in Meis2+ fibroblasts) is likely active in the bat proximal forewing and persists through subsequent developmental stages. Thus, the conclusion reached in lines 166-167 is not adequately supported or the authors should provide more convincing evidence.

2. The comparison of the RA-ID cluster vs. LMPs of bat and mouse shows that Aldh1a2- and Rdh10-positive cells have a similar transcriptional profile of pro-/anti-apoptotic genes. However, in Fig. S1J, the number of Aldh1a2/Rdh10-positive cells appears relatively lower than in mice. Also, in Fig. 2K the Aldh1a2 is seemingly smaller than in panel F. Could the author

comment on this?

3. *Meis2* is expressed in interdigit tissue at E13.5 and CS17 (e.g., Mason et al., 2015) and spatial expression seems comparable at these stages. Also, in *Rdh10* mutant limb buds, where RA signaling is absent, interdigital tissue is retained, and these cells are *Meis2*-positive (Mason et al., 2015). Could the authors exclude the possibility that apoptosis inhibition is a prerequisite for the persistence and expansion of *Meis2*+ cells in the interdigit regions of both bats and *Rdh10* mutant mice?

4. The study primarily focuses on the RA-ID cluster. However, apoptosis suppression could also occur through repression of RA and BMP signaling pathway components. The authors show that *Meis2*+ fibroblasts develop independently from the *Aldh1a2* trajectory and share a common developmental origin. While this confirms that the chiropatagium does not directly originate from the RA-ID cluster, it does not exclude the possibility that active suppression of RA/BMP signaling at the point of origin is required for *Meis2*+ fibroblast expansion—potentially separating cells sensitive to pro-apoptotic signals (*Aldh1a2*+) from the *Meis2*+ population.

5. If active suppression of the RA and BMP signaling pathways is occurring, it might not be reflected within the RA-ID cluster but rather in other cell populations or even in *Meis2*+ fibroblasts. Can the authors confirm that *Aldh1a2* and *Rdh10* are absent in other clusters? Low-level expression of these genes would suggest active suppression or downregulation of the RA pathway. For example, in the *Meis2* mutant mouse model (Fig. 4C), *Aldh1a2* expression is downregulated in *Meis2*+ cells, which could argue for active suppression of RA signaling in these cells.

6. The authors emphasize that *Grem1* is not expressed in the apoptotic RA-ID cluster. However, as an extracellular and diffusible protein, *Grem1* can exert a significant paracrine effect without being expressed in these cells. Indeed, *Meis2/Tbx3* fibroblasts express *Grem1*, suggesting that these cells may actively counteract pro-apoptotic BMP signaling, thereby protecting themselves or surrounding cells (e.g., other *Meis2*- fibroblasts in the chiropatagium) from BMP-induced apoptosis. Given that *Grem1* downregulation coincides with the onset of interdigital apoptosis in mice, its prolonged expression in bats could provide an additional layer of apoptosis suppression. Since *Bmpr1a* is essential for interdigital apoptosis, it would be relevant to assess how this receptor is expressed in bat and mouse interdigital regions.

8. Given that canonical WNT signaling is essential for interdigit tissue specification and maintenance in mice (Malkmus et al., 2024) and that components of this pathway are expressed in bat forewings (Eckalbar et al., 2016), could the authors compare mouse forelimb and bat forewing datasets? Specifically, identifying components of the canonical WNT signaling pathway expressed in the interdigit at E11.5/CS15 and E13.5/CS17 may be relevant. The persistence of WNT signaling pathway component expression in bat interdigits at later stages could provide an alternative mechanistic explanation for interdigit tissue retention in bat forewings. Additionally, given that *TBX3* functions as a tissue-specific cofactor of canonical WNT signaling in E10.5 forelimbs (see Zimmerli et al., 2020), incorporating WNT and *TBX3* link could strengthen the argument for repurposing the proximal limb program.

9. It remains unclear whether *MEIS2* predominantly binds to enhancers or promoters; a peak distribution plot would clarify this.

10. Can the authors provide further characterization of bat *MEIS2*-bound regulatory elements? Specifically, what proportion of these elements are conserved (bat vs. mouse)? Are any of these enhancers active in the proximal limbs of bats and mice?

11. Do *MEIS2* peaks overlap with bat accelerated regions (BARs) identified in previous studies (Booker et al., 2016; Eckalbar et al., 2016)? That would provide an additional support to the role of *Meis2* in bat chiropatagium formation.

12. The manuscript describes interdigital tissue retention, but apart from webbing between some digits (Fig. 4G/H), it is difficult to determine what exactly is considered interdigital retention?

13. *Tbx3* transgene expression is not shown—does it behave in the same manner as the *Meis2* transgene?

14. For Figure 4F/H, the sample size (n-values) is not well documented. How many limbs were analyzed, and how many exhibited the interdigit phenotype? It seems the 3D imaging panels in Fig4 and Fig S9 are the same limb buds with different angles. Instead authors should show different representative examples.

15. The use of the *BMP2* distal enhancer for functional studies is surprising, given that these interdigit cells are likely to undergo apoptosis in mice. The "interdigit retention phenotype" presented in this report is not convincing. The authors should consider alternative explanations; for instance, the moderate phenotype observed could be due to the elimination of *Meis2*+ cells by interdigital apoptosis.

Minor Corrections

- The cluster comparison appears to be based on pooled developmental stages (Fig. 1 & Fig. S1), which could dilute key differences that may only become apparent at later, apoptosis-relevant stages. Perhaps a direct late-stage comparison between mouse and bat forelimbs would provide a more precise assessment.
- In Figure 3F, the label reads "gene distance in bp"—it would be helpful to clarify the exact meaning of this term.
- Line 341: Should reference Fig. 4H.
- Line 343: Should reference Fig. 4I-K.

Booker, B. M., Friedrich, T., Mason, M. K., VanderMeer, J. E., Zhao, J., Eckalbar, W. L., Logan, M., Illing, N., Pollard, K. S. and Ahituv, N. (2016). Bat Accelerated Regions Identify a Bat Forelimb Specific Enhancer in the HoxD Locus. *Plos Genet* 12, e1005738.

Eckalbar, W. L., Schlebusch, S. A., Mason, M. K., Gill, Z., Parker, A. V., Booker, B. M., Nishizaki, S., Muswamba-Nday, C., Terhune, E., Nevonen, K., et al. (2016). Transcriptomic and epigenomic characterization of the developing bat wing. *Nat Genet* 48, 528–536.

Kaltcheva, M. M., Anderson, M. J., Harfe, B. D. and Lewandoski, M. (2016). BMPs are direct triggers of interdigital programmed cell death. *Dev. Biol.* 411, 266–276.

Mason, M. K., Hockman, D., Curry, L., Cunningham, T. J., Duester, G., Logan, M., Jacobs, D. S. and Illing, N. (2015). Retinoic acid-independent expression of Meis2 during autopod patterning in the developing bat and mouse limb. *EvoDevo* 6, 6.

Malkmus, J., Morabito, A., Lopez-Delisle, L., Esteban, L.A., Mayran, A., Zuniga, A., Sharpe, J., Zeller, R., Sheth, R. *bioRxiv* 2024. doi:<https://doi.org/10.1101/2024.12.25.629665> WNT signaling coordinately controls mouse limb bud outgrowth and establishment of the digit-interdigit pattern

Zimmerli, D., Borrelli, C., Jauregi-Miguel, A., Söderholm, S., Brüttsch, S., Doumpas, N., Reichmuth, J., Murphy-Seiler, F., Aguet, M., Basler, K., Moor, A.E., Cantù, C. TBX3 acts as tissue-specific component of the Wnt/ β -catenin transcriptional complex. *Elife*. 2020 Aug 18;9:e58123.

*****END*****

Version 1:

Decision Letter:

28th March 2025

Dear Francisca,

Your revised manuscript entitled "Comparative single-cell analyses reveal evolutionary repurposing of a conserved gene program in bat wing development." has now been seen by the same reviewers, whose comments are attached. The reviewers are mostly satisfied with your revisions but Reviewer#3 has some issues with the apoptosis assay which will need to be addressed before we can offer publication in *Nature Ecology & Evolution*. We will therefore need to see your responses to the criticisms raised and to some editorial concerns, along with a revised manuscript, before we can reach a final decision regarding publication.

We therefore invite you to revise your manuscript taking into account all reviewer and editor comments. Please highlight all changes in the manuscript text file in Microsoft Word format.

* If you have not done so already please begin to revise your manuscript so that it conforms to our Article format instructions at <http://www.nature.com/natecolevol/info/final-submission>. Refer also to any guidelines provided in this letter.

* Extended Data Figures - please ensure that any supplementary figures and tables that are crucial to the manuscript's conclusions are converted into Extended Data figures and tables to increase visibility of these data. Extended Data figures and tables are online-only (present in the online PDF and full-text HTML versions of the paper), peer-reviewed display items that provide essential background to the article but are not included in the main article due to space constraints. A maximum of ten Extended Data display items (figures and tables) is permitted.

Link Redacted

Nature Ecology & Evolution is committed to improving transparency in authorship. As part of our efforts in this direction, we are now requesting that all authors identified as 'corresponding author' on published papers create and link their Open Researcher and Contributor Identifier (ORCID) with their account on the Manuscript Tracking System (MTS), prior to acceptance. ORCID helps the scientific community achieve unambiguous attribution of all scholarly contributions. You can create and link your ORCID from the home page of the MTS by clicking on 'Modify my Springer Nature account'. For more information please visit www.springernature.com/orcid.

[redacted]

Reviewers' comments:

Reviewer #1 (Remarks to the Author):

I appreciate the authors for effectively responding to the comments and the manuscript has improved significantly.

One small comment is that the submitted manuscript PDF file seems to have markups that are still in suggesting mode. Please process the final changes so it's clear which ones were incorporated.

In discussion lines 545-548, I understand the author's idea, but it could be argued that finding similar clusters in the dissected chiroptagium in bat is because of the label transfer. Therefore referring to the dissected chiroptagium data might not be the best entry point to make the statement.

From my understanding, the authors can reformulate this part to explain that: you did find highly similar cell populations in mouse and bat FL, which suggests that the overall expression patterns between cell types in these structures are conserved. However, by various analyses and assays, you discovered that the morphological differences came from a distinct regulatory program that introduced a bat-specific developmental trajectory. Then bring up TF repurposing and other evolutionary examples. As the authors mentioned, it is discovered elsewhere that convergence can lead to similar cellular expression patterns in scRNA-seq but then they have different regulatory programs that give them distinct phenotypes.

Reviewer #2 (Remarks to the Author):

The authors have satisfactorily addressed my concerns in their revision.

Reviewer #3 (Remarks to the Author):

I appreciate the authors' thorough responses and the additional data provided, which have significantly improved the manuscript. I believe the study now merits publication in Nature Ecology & Evolution.

However, I have remaining concerns regarding the apoptosis assay and its interpretation, which the authors may wish to address. While the shift to an intraspecies comparison is an improvement, I remain unconvinced by certain observations:

- **Caspase-3 Staining:** The bat forelimb's Caspase-3 staining appears different between digits I/II and the remaining digits, especially in the overview image. The brighter, more uniform staining in digits I/II along the proximodistal axis hints at potential differences in apoptotic activity.

- **Forelimb-Hindlimb Comparison:** This comparison may be flawed due to potential heterochrony between the limbs, as observed in mice. Existing bat forelimb and hindlimb WISH data suggest a developmental delay in the hindlimb, potentially undermining its suitability as a control.

- **Imaging Methodology:** The imaging technique lacks clarity. While the figures originate from whole-mount samples, it's unclear whether they are maximum intensity projections or selected z-stacks. If they are selected Z stacks- it would benefit from additional data points or 3D reconstructions to support the conclusions.

- **Biological Relevance of 3 RA-ID Cluster Differences:** The biological relevance of the differences within the 3 RA-ID cluster remains unclear. While the expression levels of apoptotic genes may appear similar, the number and spatial distribution of RA-ID cells may be of greater significance. The analysis, which relies on relative expression comparisons (3 RA-ID vs.

mesenchymal cells), does not clearly explain—in either the text or figures—how these comparisons inform our understanding of potential differences in the abundance or location of RA-ID cells between species.

Despite these reservations, the revisions and the incorporation of the idea that differentiation may be a prerequisite for apoptosis evasion is appropriate and broaden the interpretation of the findings.

*****END*****

Version 2:

Decision Letter:

4th April 2025

Dear Francisca,

Thank you for submitting your revised manuscript "Comparative single-cell analyses reveal evolutionary repurposing of a conserved gene program in bat wing development." (NATECOLEVOL-24123513B). We have checked your revisions and are happy in principle to publish your paper in Nature Ecology & Evolution, pending minor revisions to comply with our editorial and formatting guidelines.

If you have not done so already, please ensure that you also email us completed copies of the Reporting summary and Editorial policy checklists:

Reporting summary: https://www.nature.com/documents/nr-reporting-summary.pdf

Editorial policy checklist: https://www.nature.com/documents/nr-editorial-policy-checklist.pdf

[redacted]

*We have replied to the reviewers' invaluable concerns and comments in **PURPLE**. All the changes made to the manuscript, derived from these comments are **highlighted in yellow** in the revised manuscript document.*

Reviewers' comments:

Reviewer #1 (Remarks to the Author):

Mechanisms of chiroptagium development in bats are unknown and the authors presented a thorough investigation to provide cellular and molecular explanations. Utilizing single-cell RNA-seq and cross-species comparisons, the author found that chiroptagium cells have a unique trajectory characterized by MEIS2 and TBX3 expression, which is also shown to be a repurposed proximal gene program. Further omics and experimental validations were performed to support this finding. Overall, this study is high in novelty and would be of interest to researchers in various areas of Evo-devo. I appreciate the author's collaborative effort in combing through diverse approaches to pursue a hypothesis extensively. My main concerns lie in the correctness of the theories and methods referred to; the clarity of some analyses; the structure and logic of the manuscript; and the accuracy of some statements.

We would like to thank the reviewer for their constructive comments and the effort put into reviewing our manuscript to improve its precision and readability.

The correctness of theories and methods referred to:

1. In line 108, please note that the SCTransform algorithm is not an integration tool, but a normalization method (see DOI: <https://doi.org/10.1186/s13059-019-1874-1>). Further in line 891 in methods, IntegrateData is also not an integration tool but a function to apply an integration method of choice. Please check which integration method was used in this function during the analysis and correct it accordingly.

We thank the reviewer for spotting these mistakes, which have been corrected accordingly:

Corrected line 108: Using the Seurat v3 single-cell integration tool, we generated an inter-species single-cell transcriptomics limb atlas (Fig. 1C).

Corrected line 891 (Methods): These anchors were used with the Seurat v4.3.0 function IntegrateData and the normalization method "SCT".

2. The statement in line 119 can be misleading. I found out in the figure legends that Fig. 1E shows genes used to annotate the integrated cluster, rather than the top markers calculated based on integrated cross-species data. It would be more correct to say that e.g. "Expression of marker genes used for cluster annotation is shown in Fig. 1E, from which we confirmed their conserved specific expression patterns between species".

This is indeed misleading in the text. In Figure 1E we show the most well-known markers for each cell cluster, chosen by us. To clarify this further, we have corrected line 119 accordingly and added a new plot to the supplementary figures, showing the main unsupervised markers that define each cell cluster in the integration. These show a similar conservation.

Corrected line 119: The expression of the marker genes used for cluster annotation (Fig. 1E), and marker genes differentially expressed in each cluster (Fig. S1A), were also conserved across species.

New panel in Fig S1

Dot-plot showing the top 3 differentially expressed marker genes per cluster. The color intensity indicates the expression level (blue: mouse; red: bat); the dot size represents the percentage of cells expressing respective genes.

3. The term “deep-homology” is used to describe the conservation of gene expression patterns in non-homologous structures. However, mouse forelimbs and bat wings both derive from the forelimbs of a shared tetrapod ancestor, and are therefore homologous tissues. The reference to deep homology as an explanation for cell type conservation, in this case, seems invalid. In contrast, the idea of TF repurposing is highly valid and could be expanded further. Therefore, could the authors please reformulate the relevant discussions (lines 397-401).

The reviewer is correct. Deep homology does not relate to the process we describe. We have now removed it from the discussion.

Corrected line 397: It is well-documented that during convergent evolution, the same set of genes is often reused.

Clarity of some analysis:

4. The orthology mapping between bat genes and mouse genes is an essential step for the

integration analysis and its quality can greatly impact the output. However, the details about how this mapping has been done in the main text or methods seem missing. I saw that the authors annotated the bat genome with the human genome, but how is the orthology mapping to the mouse done? Please provide the bat-mouse orthology mapping table as supplementary data. In methods, please indicate how bat-mouse orthologs were called for cross-species integration. Were only one-to-one orthologs used, or non-one-to-one orthologs were also included in some way?

Following this suggestion, we provide a new supplementary orthology table between mouse and bat annotations. Additionally, we have expanded the methods section to explain this.

Insertion in line 850 (Methods): For the comparative analysis of mouse genes, we used genome version mm39 (GCF_000001635.27) with annotation release 109. Only gene entries of type gene, exon, CDS, pseudogene, transcript, primary_transcript, and RNA types (excluding guide_RNA) were processed further. Finally, gene models overlapping exons of known genes, or predicted transcripts where an alternative curated RefSeq-entry (ID starting with NM_ or NR_) existed were removed. Additionally, 3 fusion transcripts were removed. Ortholog relationship was determined by a one-to-one comparison of the Carollia and Mouse genomes via LAST (same parameter settings as for hg19). A mouse gene was defined as an ortholog of the Carollia gene with maximum of shared exon boundaries. In case of ambiguity the gene with highest overlap was assigned. As a consequence, only one-to-one ortholog assignments were generated (extended data S11).

New additional file: Orthology table between Mouse and Bat annotations.

5. It is not clear why both human reference genomes hg19 and hg38 are used during *C. perspicillata* genome annotation (see line 818 and line 826). Could the authors please verify?

There is no methodological reason for this discrepancy; it reflects the fact that initial annotation and subsequent refinement were carried out by different collaborators.

The initial genome annotation was performed using hg38, which is the most complete version. The refinement, performed with hg19, aimed to resolve specific conflicts, such as UTR overlaps in the Hox clusters. However, given that the gene space remains highly similar between these genome versions, we are confident that this does not introduce any significant bias in the refinement process.

6. The sentence in line 134 could be further developed. It is highly relevant where the *Grem1* gene is actually expressed, since it is a known gene specific to the bat interdigital tissues. Adding an additional figure to show the expression pattern of *Grem1* gene in each cluster for both species will be helpful.

The expression of *Grem1* is discussed further down in the manuscript. Indeed, in Fig2 F/J and K/O, the expression pattern of *Grem1* is shown in the different distal cell populations and differentiation trajectories in Mouse and Bat. Furthermore, *Grem1* is shown in Fig. 1G where it now has been highlighted. To better explain this, we made the following change.

Insertion in line 192: Moreover, this *MEIS2*⁺ trajectory also showed high expression of *GREM1*. Both of these have been shown to be specifically expressed in the interdigital tissue

of bat wings, as well as other interdigital markers like *Aldh1a2* (Weatherbee et al. 2006; Mason et al. 2015). Thus, confirming that this cell population shares this space with the cluster 3 RA-Id in bats.

7. Based on the correlation heatmap shown in Fig. S3 C, D, I would say that many clusters have a group-level correspondence instead of a one-to-one correspondence, as indicated by blocks spanning a few clusters on the heatmap. This could be biological, as studies have shown that cell types might exhibit strong group-level conservation, especially at such high granularity. However, this could also be because using only the top 10 marker genes reduced the sensitivity. Could the authors please try to use more marker genes to perform this correlation? Based on the results, please modify the statements about cross-species cluster conservation accordingly.

This is a valid point. As per reviewer's suggestions, we performed the correlation, using all the marker genes, instead of the top 10 per cluster. This new analysis resulted in an almost identical set of correlations (see plots below):

Same heatmap as in Fig. S3 C, but using all the marker genes, instead of the top 10 per cluster.

Distribution of the correlations used to produce heatmap in figure Fig. S3 C and the correlations presented in the heatmap above.

This result is somewhat expected, given the nature of the LPM-derived mesenchymal cells. Limb single-cell analyses (refs. 20, 21, 22, 24, and others) show a highly homogeneous group of cells, with diffuse boundaries between cell populations. Limb fibroblast populations have been even referred to as a “continuum of promiscuous fibroblast identities” (Hirsinger et al. 2024), reflecting the plasticity and lack of clear discrete mesenchymal identities. While group-level conservation is notorious, for many clusters we can still find a stronger one-to-one correspondence within their main group/lineage. In the initial version of our manuscript, we already avoided referring to clear one-to-one cluster correspondence. Based on this analysis, we believe that no specific changes should be implemented in the manuscript regarding this aspect.

Structure and logic of the manuscript:

8. The introduction part (under “Main Text”) seems to lack a clearly defined thesis statement for this paper. This is to propose the main research question and briefly outline the main work done. While I see that alike statements are positioned at the beginning of discussions, it will be much clearer to put them at the end of the introduction. Instead, the discussions can focus on the implications of these new findings instead of the necessity of the research.

We thank the reviewer for this suggestion which we agree will improve the readability of the manuscript. Accordingly, the following paragraph has been added at the end of the main text.

Insertion in line 101: To investigate the molecular origins of wing formation, we performed single-cell transcriptomics (scRNA-seq) at multiple time points during bat and equivalent mouse embryonic limb development. Our data reveal conserved cell clusters and gene expression patterns across species, including within the apoptosis-related cell population. Additionally, we characterized the origin of the chiroptagium, which is composed of fibroblastic cells that follow a differentiation trajectory independent of retinoic-acid-active interdigital cells and repurpose a gene program typically restricted to the proximal limb. By ectopically expressing two upstream transcription factors of this program, *MEIS2* and *TBX3*, in the distal limb of transgenic mice, we recapitulated key molecular and morphological features observed in developing bat wings. Altogether, our findings demonstrate that an existing proximal cell state and its gene regulatory program are repurposed in the distal bat forelimb to generate a novel tissue in a different spatial location.

We have also added new paragraphs to the discussion to further clarify our results and their implications for the evolution of limb development, and addressing the specific suggestions of Reviewer 3. Modifications are highlighted with track changes.

9. Furthermore, to improve the clarity and logic of the introduction, the second paragraph could be split into two paragraphs at line 87. The former part is to explain the developmental process of the limb, while the latter addresses the hypothesis and brings out the key research question.

Thank you for this suggestion. The text has been modified accordingly.

Correction at line 87: New paragraph.

Accuracy and necessary explanations:

10. In line 97, single-cell approaches can be applied to “any” organism is an overstatement and “many” would be more appropriate.

The text was changed accordingly.

Corrected line 96: Recently, single-cell approaches have provided new tools to investigate cell identity and function at unprecedented resolution in many organisms, holding great potential to unravel the basis of evolutionary innovation (18).

11. A few suggestions on the use of acronyms:

1) Clusters in scRNA-seq data can be better named and consistently referred to. For instance, the cluster of interest can be named “3 RA-Id” rather than “RA-Id cluster 3” or “RA-Id cluster”, which can cause confusion such as in line 131 vs line 179 vs line 193.

We agree with the reviewer. For consistency we have chosen to only use the term “3 RA-Id” and have modified the text accordingly.

2) The log fold change value of gene expression has a canonical acronym as “log2FC”, instead of “lfc”. These two acronyms are also used in a mixture throughout the manuscript and please harmonize them.

We thank the reviewer for spotting this issue. The text has been modified accordingly.

3) Please always include an explanation of acronyms used in Figures at the end of the figure legends. Many might seem self-explanatory but it is good practice to always clearly state this. For instance, what does “PD” mean in Fig. 2P?

We once again thank the reviewer and have modified the text accordingly.

Insertion in line 163 (Fig. 1 legend): ID = Interdigital region.

Insertion in line 232 (Fig. 2 legend): PD = Difference of the expression score of distal genes (Hoxd13), minus the expression score of the proximal gene (SOx2).

Correction in line 298 (Fig. 3 legend): SCENIC Transcription Factors network analysis for genes enriched in cluster 10.

Correction in line 299 (Fig. 3 legend): Tornado plot showing H3K27ac peaks specific to the distal forelimb (dFL) as well as common peaks of distal forelimb and distal hindlimb (dHL).

Insertion in line 674 (Fig. S1 legend): nCount = Number of UMI counts per cell, nFeature = Number of detected expressed genes, percent.rp = percentage of UMIs originating from ribosomal genes.

Insertion in line 691 (Fig. S2 legend): ID = Interdigital region.

Insertion in line 727 (Fig. S5 legend): PD = Difference of the expression score of the distal genes, minus the expression score of the proximal gene.

12. In line 186, slingshot could use a citation. In line 257 SCENIC could use a citation.

As a matter of style, we chose to keep methodological citations within the methods section, where all tools we used are listed and referenced.

13. In line 267, could the authors please add "in bat FL" after the "most upregulated genes" so it's easier for the reader to follow the fact that several proximal markers are elevated in bat FL.

Thanks for this comment, we modified the text accordingly.

Corrected line 267: Among the most upregulated genes in bat FL we found the TFs *MEIS2*, *HOXD9*, *HOXD10*, *HOXA2* and *TBX3*, genes known to be early proximal markers and patterning factors (38, 39) (Fig. S8).

14. Could the authors please mark genes from the chiropatagium gene program to Figure 4 C to make it easier to understand? From my understanding, not all genes in the x-axis are associated with this program and it's possible to note which genes are.

In fact, we only show the genes which are in the chiropatagium gene program. To clarify this, we suggest the following changes:

Correction in line 355 (Fig. 4 legend): Expression heatmaps showing differentially expressed genes from the chiropatagium gene program in affected limb clusters of mouse mutant limbs at E12.5 (1 MR and 3 RA Id in *MEIS2* mutant; 1 MR, 3 RA Id and 4 DP in *TBX3* mutant). Non-significant differences, and differences below 0.25 log₂FC were all set to 0. The number of genes from the chiropatagium gene program, total number of differentially expressed genes, and p-value from fibroblast gene program over-representation are shown on the right.

Correction in Fig 4C: We have added the total number of genes that show a differential expression.

Moreover, we have added new panels to the supplementary figure 9 to represent all the differentially expressed genes between the mutant and wt, highlighting those that belong to the chiropatagium gene program.

New panels in Fig S9:

Plots showing all the significantly differentially expressed genes (adjusted p-value < 0.01 & $|\log_2\text{FC}| > 0.25$) in the affected limb clusters of mouse mutant limbs at E12.5 (1 MR and 3 RA Id in MEIS2 mutant; 1 MR, 3 RA Id and 4 DP in TBX3 mutant). Genes from the chiropatagium gene program are highlighted.

15. What do the values in the pie chart of GO enrichment stand for in Fig. 4D? Using such a Pie chart to show GO can be confusing as it's fairly unconventional. There are many standardized representations of GO enrichment results such as dot heatmaps or bar heatmaps (such as in Figure 3 D), and I suggest remaking the plot to these forms which could improve clarity.

We agree with the reviewer that there may be a more effective way to represent this data. Therefore, we propose to change the pie charts with the following panels and supplementary figures.

Substitution of panel D in Fig. 4:

Correction in line 358 (Fig. 4 legend): Proportion of GO-Term categories (biological functions) upregulated in mutant mice. From the top 10 GO Terms of the affected cell clusters. Individual GO Terms in Fig. S9.

New panels in Fig S9:

Barplots showing the proportion of GO-Terms categories found upregulated in mutant mice. GO Term categories reflect biological functions. Shown are the top 10 GO Terms for every affected cell cluster.

16. P-values in Fig. 4 I-K are not explained and please add details of the statistical tests to the legend.

We apologize for this mistake, which has been corrected as follows:

Insertion in line 366 (Fig. 4 legend): Numbers on the brackets are p-values of the differences of the mean calculated using a Dunnett test following a one-way ANOVA.

17. Gene names are not in italic font in the methods, please format them like in the main text.

We have reviewed the text and corrected it accordingly.

Reviewer #2 (Remarks to the Author):

The manuscript by Schindler et al., *Comparative single-cell analyses reveal evolutionary repurposing of a conserved gene program in bat wing development*, explores the development of the evolutionary innovation that allows for powered flight in bats, the chiropatagium. The authors argue that, because there is a transcriptionally similar population of apoptotic cells in all the mouse and bat limbs they sampled, that the chiropatagium is not the result of an exceptional reduction in apoptosis in the bat forelimb. They present a compelling case for a bat-forelimb-specific population of distal fibroblasts in the chiropatagium. Using mouse transgenics, they show that *MEIS2* and *TBX3* can partially induce a chiropatagium-like genetic signature. Thus, the authors argue for a mechanism by which a set of genes from the proximal limb are redeployed in service of producing an evolutionary novelty in the distal limb.

Overall, I think the authors have done a nice job with this paper: it is clearly written and the figures are well constructed. The methods appear to be thorough. The authors have marshaled a large amount of data here, and they use it to tell a compelling story that will substantively contribute to the literature on deep homology and evolutionary novelty. Most of my concerns are minor.

We would like to thank the reviewer for the positive feedback of our work. We have modified the text to incorporate the reviewer's suggestions as indicated below.

Comments:

Missing Ns: It isn't clear from the descriptions in the results, figure captions, and methods, how many embryos contributed cells to the pools in the scRNA-seq experiments, or to the bulk transcriptomic and epigenomic experiments. Fig. S1A–C shows the scRNA-seq samples, but I couldn't easily find the others. Similarly, on line 339 the authors state that 'all *TBX3* mutants displayed fusion of at least two digits.' How many mutants were produced?

We apologize for the lack of clarity in the text.

-Regarding the single-cell experiments, we state in the methods (line 877) that the experiments were performed in biological duplicates. Each replicate comes from a single individual, with no pooling. We have modified this line for clarity as follows:

Corrected line 877: Single-cell experiments were performed in biological duplicates, with each replicate derived from a single different individual.

-Regarding the bulk RNA-seq, in the methods we state that the experiments were performed at least in biological duplicates (line 1048). To be more precise we have modified the text as follows:

Corrected line 1048: RNA-seq experiments were performed in biological duplicates for the bat samples. For the mouse samples 3 and 5 biological replicates were used for the *TBX3* transgene and for the *MEIS2* transgene and wildtype, respectively.

-Regarding the ChIPseq experiments, as stated in line 1083 all experiments were performed in biological duplicates.

-Regarding the number of mutant limbs analyzed, as stated in the methods, lines 1255 and 1257, 4 independent samples were imaged. To clarify the statement in the main text, we have made the following corrections:

Corrected line 339: all *TBX3* mutants displayed fusion of at least two digits (Fig. 4H, n=4).

Corrected Figure legend of Fig. 4. F and G 3D imaging of mouse wildtype and mutant limbs at E15.5 (n=4).

We see full penetrance of the syndactyly phenotype in the *Tbx3* mutants.

Furthermore, we added the following panel in figure S9, to show images of all *TBX* mutants.

New panel in Fig S9:

Fig. 1G and S2A: What threshold do the dashed lines represent? Should be stated in the legend.

We thank the reviewer for drawing our attention to this detail. We have made the following changes.

Insertion in line 159 (Fig. 1 legend): Dashed lines represent a difference of 0.25 and -0.25 of the log₂FCs.

Insertion in line 688 (Fig. S2 legend): Dashed lines represent a difference of 0.25 and -0.25 of the log₂FCs.

Fig. 1H, S2B, 4H, S9D: The figure panels or legends should indicate what the colors in the merged micrographs represent.

We thank the reviewer for pointing out this mistake. We made the following changes to make this clear.

Corrected line 162 (Fig. 1 legend): Merged images show DAPI (white) and LysoTracker (red) or Cleaved Caspase-3 (yellow) signal.

Insertion in line 364 (Fig. 4 legend): Magenta = eosin-Y, Cyan = nuclei, Yellow = autofluorescence.

Corrected line 691 (Fig. S2 legend): Merged images show DAPI (white) and LysoTracker (red) or Cleaved Caspase-3 (yellow) signal.

Insertion in line 772 (Fig. S9 legend): Magenta = eosin-Y, Cyan = nuclei, Yellow = autofluorescence.

Fig. S3E–H: The x-axis label text is stretched and difficult to read.

We apologize for this mistake, which is an artifact of PDF conversion. We will ensure, together with the editorial team, that this is corrected in an eventual final version.

Fig. S3B, D: Curious what's going on with the hindlimb RA-id cluster in the bat. From these two panels, it looks like there is no clear RA-id cluster in the bat hindlimb; expression of the marker genes from the mouse RA-id cluster appear to cluster with that of a few other clusters (Fig. S3D), including those of two MR clusters. This is confusing because, in Fig. S2A, a comparison is shown between genes expressed in the bat and mouse hindlimb RA-id clusters. Is the bat HL RA-id I understand that the hindlimb is not the focus of the paper, but it seems like an important comparison to the forelimb, and has been set up as such in Fig S2.

Indeed, when analysing the individual tissue samples, we were not able to recover an individual RA-Id cluster for the bat HL. We used the same clustering settings (relevant here, a resolution of 0.7) for all our analyses to maintain uniformity. A higher resolution would increase the number of clusters overall, and add further confusion by having every cluster doubled or tripled.

Nonetheless the RA-Id cells exist in this sample. Most of the *Aldh1a2*⁺ cells are found between cluster 1.2 MR and 17 ChH (see Fig S3B, S3D in the column corresponding to Mm 3 RA-Id, and the inserts below). However, as most of the cells in these clusters correspond to either cluster 1, or 17 of our interspecies integration, they were labeled correspondingly.

However, Fig. S2A corresponds to Fig. 1, and the first section of our results. Fig. S2 is referenced in the paragraph line 121, which reads “Using **this inter-species single-cell atlas**, we first sought to address the prevailing hypothesis that chiroptagium development is driven by inhibition or reduction of apoptotic cell death in the interdigital tissue (13)”. We refer here to the integrated dataset, which has been clustered on its own. Here we have a clear cluster 3 RA-Id composed of FL and HL cells from Mm and Cp (see below). We used this integrated dataset to make sure we are comparing the most corresponding sets of cells.

Same plot as in Fig. 1D, but only showing Cp HL cells. In Mexican Pink the 3 RA-Id cluster, in gray the rest of the cells. This constitutes the comparison shown in the x-axis of Fig. S2A. See Fig. 1F to compare with *ALDH1A2* expression pattern.

Same plot as in Fig. 1D, but only showing Mm HL cells. In Mexican Pink the 3 RA-Id cluster, in gray the rest of the cells. This constitutes the comparison shown in the y-axis of Fig. S2A. See Fig. 1F to compare with *Aldh1a2* expression pattern.

To improve clarity on this aspect, we performed the following modifications in the text.

Correction in line 684 (Fig. S2 legend): Correlation of pro- (yellow) and anti-apoptotic (red) genes in the interspecies-integrated 3 RA-Id cell population of mouse and bat.

Typo: In Fig. 2S, 'Maker' should be 'Marker'

We corrected this typo.

Correction in Fig. 2S: Marker genes.

In my PDF, the bottom of the RNAseq track in Fig. 3J is partially cut off.

This is an artifact of PDF conversion. We would make sure, together with the editorial team, that this is not the case for an eventual final version.

Line : Could the choice to proceed with functional test of TBX3 be clarified? The authors say that 'The striking pattern of chromatin activity profiles (H3K27ac and MEIS2 binding) being constrained within regulatory domains is exemplified for the TBX3 domain (Fig. 3J, and TBX2 in Fig. S8).' Indeed, the tracks in Fig. 3J and S8E show an enrichment of MEIS2 binding and H3K27Ac within these TADs, but the reader is left wondering why the authors didn't test TBX2, or any of the other TFs with strong coincident acetylation and MEIS2 binding.

We chose *TBX3* for functional validation for a variety of reasons. First, based on chromatin activity profiles (accessible H3K27ac- marked chromatin), *TBX3* is among the highest-ranked transcription factors (ordered from left to right) of the top 20 MEIS-bound chiropatagium gene program TADs as shown in Fig. 3I. Second, *TBX3* is a known important player of limb development and has been proposed in the literature as a potential candidate for bat wing development based on its expression pattern. *TBX3* is of particular interest as it has been reported earlier as a possible candidate in bat wing development (Dai et al. 2014) and its prominent role in limb development (Davenport et al. 2003). To better describe our rationale, we implemented the following changes in the text:

Correction in line 279: The top 20 genes displaying the highest overall MEIS2 binding signal in their regulatory domains included genes from the fibroblast gene program, like ECM components and TFs such as *TBX3* and *TBX18* (Fig. 3I, ranked from left to right according to the acetylation coverage).

We agree with the reviewer that other transcription factors, including *Tbx2* or *Tbx18*, could be of great interest for further analysis. However, given the extensive lab work and animal use required for these types of experiments, we prioritized two prominent transcription factors from the program, *Meis2* and *Tbx3*.

Fig. 4E: What do the dashed lines represent? Should be explained in the legend.

We have made the following changes to better explain the dashed lines.

Insertion in Fig. 4E: "Mean"

Insertion in line 361 (Fig. 4 legend): Dashed line = mean.

Fig. 4G, S9D: The images of the *Meis2* mutant show a reduced digit number, while those of the *Tbx3* mutant indicate a possible extra digit, suggesting non-specific patterning effects of the transgene expression. Should we worry that the phenotypic effects in these mutants, especially the syndactyly, is more of a patterning issue and less of a chiropatagium-like phenotype?

We thank the reviewer for this comment. The “possible extra digit” observed in *Tbx3* mutants is a common polymorphism of the mouse line used in these experiments (B6/129) and is frequently observed when generating transgenic mice. This is not an actual extra digit, as it lacks any chondrogenic condensation; it is rather just a piece of tissue.

In contrast, the loss of a digit in *Meis2* mutants is a true phenotype associated with the genetic mutation that we introduced. This phenotype is consistent, reproducible, and has been observed in all 4 individuals of our aggregation experiment.

It is important to note that the enhancer used in our transgene experiment is also expressed early in the distal part of the limb, whereas in bats, *MEIS2* is expressed proximally and is activated only later distally. Since it is impossible to fully recapitulate the bat's endogenous *MEIS2* / *TBX3* expression, additional effects cannot be completely ruled out and should be considered. Previous experiments have shown similar ectopic expression of *Meis2* can lead to a range of phenotypes, including digit loss (Capdevila et al. 1999). Therefore, we believe, as the reviewer suggests, the digit loss to be an unrelated patterning phenotype.

The syndactyly phenotype, however, is due to the retention of interdigital tissue, or, a lack of interdigital tissue removal. This is evident seen in the scans of Fig. 4H. In the wt, the interdigital skin surrounds the digit separating them from each other, whereas it spans the digits in the mutants. Thus, what we observe here is clearly related to digit separation and is not a patterning effect.

Line 1115: ATAC-seq is described in the methods but I didn't see ATAC-seq in the results; did I miss it?

We used the ATAC-seq data to narrow down the epigenomic regions shown in figure 3 and supplementary figure 8. We have now made the following corrections to clarify this.

Corrected line 268: Differential enrichment analyses for active epigenomic regions (marked by accessible chromatin regions detected using ATAC-seq and H3K27ac ChIP-seq) revealed a significant number of regions specific to the distal bat FL, enriched in TF binding sites for RFX, ATF, GATA, ATG and, most significantly, MEIS (Fig. 3F, G and Fig. S8).

Corrected line 277: By intersecting accessible H3K27ac- and MEIS2-binding enriched domains with genes from the distal/proximal fibroblast gene program, we narrowed down the list of candidate genes potentially regulated by MEIS2 to 71 (Fig. 3H).

Reviewer #3 (Remarks to the Author):

The molecular mechanisms underlying chiropatagium development in bat forewings remain poorly understood. Schindler, Feregrino et al. investigated this question using comparative single-cell RNA sequencing of bat wings and mouse limbs. Contrary to the prevailing hypothesis that interdigital tissue persistence is due to anti-apoptotic mechanisms, the authors observed that apoptosis in bat forewings proceeds similarly to mouse limbs and bat hindlimbs. They identified a unique population of fibroblasts in bat forelimbs, characterized by expression of *Meis2* and *Tbx3* transcription factors, which are associated with proximal limb identity. Further chromatin and regulatory network analysis suggest that these factors play a key role in driving fibroblast differentiation in bat forewings. To functionally test this hypothesis, the authors generated transgenic mice expressing *Meis2* and *Tbx3* in distal interdigital tissue. The resulting phenotype exhibited moderate digit fusion (syndactyly) and increased ECM deposition, suggesting that *Meis2/Tbx3* activation alters distal limb cell identity and contributes to interdigital tissue retention. Based on these findings the authors conclude that bat chiropatagium formation is driven by the spatial repurposing of a proximally active gene program. While the findings and differences described within this manuscript are not entirely novel and rather represent an incremental advance utilizing single-cell analysis, the manuscript itself is well-written, and the single-cell data analysis is both convincing and a valuable resource for researchers in the field. However, some conclusions appear overinterpreted or lack sufficient explanation. A more nuanced framing with alternative interpretations would strengthen the study. Here, I provide the authors constructive feedback, comments, and suggestions to incorporate into their revised manuscript.

The authors state in lines 121-123: "Using this inter-species single-cell atlas, we first sought to address the prevailing hypothesis that chiropatagium development is driven by inhibition or reduction of apoptotic cell death in the interdigital tissue." They conclude in lines 166-167: "Since cell death occurs similarly in both bat and mouse interdigital clusters, it cannot account for the persistence of interdigital tissue."

Throughout...the manuscript implies that apoptosis suppression does not play a significant role in chiropatagium development. However, the provided evidence does not definitively exclude apoptosis suppression as a potentially important mechanism (among others) for interdigital tissue retention.

We would like to thank the reviewer for the constructive feedback to improve the quality of our work. In light of their comments, we have made several revisions to the text, particularly in the discussion to better clarify and contextualize the implications of our study. One concern raised by the reviewer is that some of our conclusions might appear overinterpreted or lacking support. We have carefully considered different possible explanations provided by the reviewer and others.

1. The forelimb and forewing, at E13.5 and CS17, both exhibit slightly regressed and curved interdigital tissue at the distal periphery (e.g., Mason et al., 2015). LysoTracker/Caspase images reveal that the apoptotic domains at the distal end in bats are comparable to those in mice. However, LysoTracker and Caspase staining in mice show an additional apoptotic domain in the proximal interdigit (see e.g., Kaltcheva et al., 2016). Therefore, an apoptosis-

suppressing mechanism (e.g., in Meis2+ fibroblasts) is likely active in the bat proximal forewing and persists through subsequent developmental stages. Thus, the conclusion reached in lines 166-167 is not adequately supported or the authors should provide more convincing evidence.

We appreciate the reviewer's perspective and would like to present the evidence supporting our conclusion:

- We agree with the fact that the LysoTracker/Caspase domains in the mouse appear to extend more proximally into the interdigital tissue. However, a comparison between species is difficult in these qualitative assays. We therefore compare primarily within a species. Between species comparisons can only be done on a very general and qualitative level. In our experimental setting we use the different areas of the bat limb as internal controls: in the bat hindlimbs all digits separate completely, whereas in the forelimb only the first digit separates from the second. Digits 2-5, in contrast, are not separated, eventually forming the chiroptagium. Thus, we can compare the signal not only between fore- and hindlimb, but also between digits of the forelimb. In these comparisons, the LysoTracker and Caspase staining patterns are highly similar between digits I-II of the forelimb and the rest of the digits, as well as between the forelimb and hindlimb. This implies that similar mechanisms are at work. However, given the lack of cellular and 3D resolution it is impossible to tell if subtle differences in the distribution and intensity of apoptosis are present between the different limbs/digits.

- Our quantitative approach (Fig. 1G and S2A) analyzing the only cell cluster marked by expression of apoptotic factors (3 RA-Id), revealed no significant expression differences in known cell-death-related pathways within this cluster. And, analysis of the interdigital tissue at a later stage revealed that cells from the retinoic acid cluster were no longer present, strongly suggesting they had undergone apoptosis, in line with the staining results.

Taken together, these findings lead us to the conclusion that cluster 3 RA-Id likely undergoes apoptosis, which we can observe in staining assays, and cannot account for the wing tissue. We have modified line 166 to be more specific and avoid confusion.

Corrected line 135: To further investigate the presence, intensity and distribution of apoptosis, we stained bat limbs with LysoTracker, a marker of lysosomal activity which correlates with cell death (28).

Insertion and correction in line 136: The differential digit separation in bat limbs was used as an internal control: in bat hindlimbs all digits separate completely, whereas in the forelimb only the first digit separates from the second. Digits II-V, in contrast, do not separate in the wing, forming the chiroptagium. We found pronounced staining in all interdigital zones of bat FLs, with no discernable differences to interdigit I-II. Likewise, staining in the HL interdigit tissue was similar in intensity and distribution (Fig. 1H and Fig S2).

Corrected line 138: In addition, we confirmed that cell death in bat wings occurs via an apoptotic process activated by the caspase cascade, as indicated by the positive staining for cleaved Caspase-3 protein in a similar distribution as the described for LysoTracker staining (Fig. 1H and Fig. S2).

New paragraph in line 141

Corrected line 142: Furthermore, cell death, as shown by the assays used here, occurs similarly in all interdigital tissues in the bats regardless of whether the digits get separated or not. Although it is difficult to compare between species, our results show that interdigital apoptosis is a feature of both bats and mice.

Corrected line 166: Since cell death occurs similarly in both bat and mouse cluster 3 RA-Id, and spatially in both bat FL and HL, its inhibition is unlikely to account for the persistence of interdigital tissue.

Corrected line 179: Notably, the cluster 3 RA-Id was minimally represented in the chiropatagium (~ 1%, Fig. 2D), which is consistent with the results of the apoptosis staining (Fig. 1H).

2. The comparison of the RA-ID cluster vs. LMPs of bat and mouse shows that Aldh1a2- and Rdh10-positive cells have a similar transcriptional profile of pro-/anti-apoptotic genes. However, in Fig. S1J, the number of Aldh1a2/Rdh10-positive cells appears relatively lower than in mice. Also, in Fig. 2K the Aldh1a2 is seemingly smaller than in panel F. Could the author comment on this?

While the differences appear interesting, they are minimal, and simply reflect the uncertainty of an interspecies organogenesis experiment. These are not numbers of cells, but relative proportions. Given that bat limbs are larger, break symmetry early, and are posteriorly larger, it's likely that other cell types (e.g. fibroblasts) are more numerous. This would affect proportions overall. Moreover, we believe that different-shaped limbs would show different relative cell proportions.

The length observed in the diffusion maps (Fig. 2K) is not quantitative. We plot the cells in only 2 diffusion components calculated separately for each limb. The length of the trajectories cannot be compared based on these plots.

3. Meis2 is expressed in interdigit tissue at E13.5 and CS17 (e.g., Mason et al., 2015) and spatial expression seems comparable at these stages. Also, in Rdh10 mutant limb buds, where RA signaling is absent, interdigital tissue is retained, and these cells are Meis2-positive (Mason et al., 2015). Could the authors exclude the possibility that apoptosis inhibition is a prerequisite for the persistence and expansion of Meis2+ cells in the interdigit regions of both bats and Rdh10 mutant mice?

Indeed, we cannot rule out this possibility without genetically manipulating bat embryos. However, we find evidence for the presence of apoptosis (Cluster 3 RA-Id, its absence in later stages, and cell-death staining). We cannot rule out that expansion of Meis2+ cells is a byproduct of apoptosis inhibition in Rdh10⁻ mice, or that it would happen in bats if apoptosis were inhibited. But we do observe apoptosis and persistence of MEIS2+ cells at the same time. Therefore, we don't believe it to be a "prerequisite", at least in bats.

Additionally, we would like to highlight that interdigital remodeling is likely more complex and less understood than a classical view attributing it solely to apoptosis. As elegantly

demonstrated by Kashgari et al. (Dev Cell 2020), digit separation involves additional mechanisms, such as epidermal cell migration. Their study showed that, in mutant mice with soft syndactyly, interdigital cell death still occurs, yet a defect in periderm migration prevents complete digit separation. This serves as a compelling example of how interdigital tissue retention can occur even when apoptosis is taking place and illustrates that this process is likely far more intricate than initially thought. In fact, we still do not fully understand how it works even within the mouse.

4. The study primarily focuses on the RA-ID cluster. However, apoptosis suppression could also occur through repression of RA and BMP signaling pathway components. The authors show that Meis2+ fibroblasts develop independently from the Aldh1a2 trajectory and share a common developmental origin. While this confirms that the chiropatagium does not directly originate from the RA-ID cluster, it does not exclude the possibility that active suppression of RA/BMP signaling at the point of origin is required for Meis2+ fibroblast expansion—potentially separating cells sensitive to pro-apoptotic signals (Aldh1a2+) from the Meis2+ population.

The part of our study about apoptosis focuses indeed on the RA-ID cluster. We identified it as the only one cluster in which the known apoptosis components are expressed. Additionally, we show a second cell population sharing the interdigit space in bat wings: MEIS2+ cells which compose most of the chiropatagium at later stages. We characterize the expression profiles of these two cell populations as different between them, and conserved across species. It is well known that all cells in the limb (not only these two) share naive undifferentiated progenitors and derive their transcriptomic identities along differentiation.

We agree with the reviewer that it cannot be excluded that active RA/BMP suppression might be present in the chiropatagium progenitor cells thereby protecting them from proapoptotic signaling. However, this would assume that the progenitor cells are already destined to undergo apoptosis. According to our understanding, this is not supported by published data. In fact, experiments in the chick have shown that cells in the interdigital area are not primed to die, nor they have an early apoptotic commitment (Hurle & Gañan 1987; Gañan et al. 1996; Ros et al. 1997; Merino, Macías & Gañan 1999). Before following an apoptotic fate, these cells have the potential to differentiate to the point that they can develop fully formed digits if signaled with chondrogenic factors (Gañan et al. 1996). Based on these data it has been suggested that maintained undifferentiation and the lack of differentiation signals lead cells into an apoptotic fate (Montero et al. 2020). We therefore suspect that the presence of MEIS2+ fibroblasts might be explained by an early differentiation of this group of cells, previous to the “deadline” of apoptosis fate. This would be in line with the observations of Mason et al. 2015, and experiments in duck hindlimbs (Macias, Gañan & Hurle 1992; Montero, Gañan & Macías 2001; Verheyden & Sun 2017).

However, the precise mechanism triggering Meis2 expression remains unidentified, and while this is an important question, it falls outside the scope of the present manuscript. We would like to suggest the following addition to our discussion:

Correction and insertion in line 378: Cells expressing RA/BMP pro-apoptotic factors in bats are equivalent to the cluster 3 RA-ID observed in mice, where interdigital regression takes place. In contrast, distal bat fibroblast cells express the BMP antagonist GREM1 (Fig. 20)

previously shown to be expressed in the interdigits of the wing, but not the hindlimb (Weatherbee et al. 2006). Even though these cells are in the same interdigital space as the cluster 3 RA-Id cells, they originate from a distinct developmental trajectory eventually constituting the major component of the chiropatagium. While we don't explore the developmental origin of this cell population, their presence and persistence in an otherwise disappearing tissue might be explained by their already differentiated state. Experimental manipulations of developing chicken hindlimbs show that previous an apoptotic fate, the interdigital mesenchyme is naive with full differentiation potential (Hurle & Gañan 1987; Gañan et al. 1996; Ros et al. 1997; Merino, Macías & Gañan 1999), suggesting that apoptosis arises due to the lack of differentiation or further survival signaling (Montero et al. 2020). It is, however, possible that suppression of RA/BMP signaling by factors such as *GREM1*, serves as an additional factor protecting *MEIS2*+ fibroblasts from apoptosis. Furthermore, as shown by our transgenic experiments, ectopic of *MEIS2* results in a downregulation of *Aldh1a2* indicating that *Meis2* itself may have an antiapoptotic effect. **[New paragraph]**

5. If active suppression of the RA and BMP signaling pathways is occurring, it might not be reflected within the RA-ID cluster but rather in other cell populations or even in *Meis2*+ fibroblasts. Can the authors confirm that *Aldh1a2* and *Rdh10* are absent in other clusters?

We agree that this might be possible. *Aldh1a2* and *Rdh10* are predominantly expressed in the RA-Id cluster but low levels of expression can also be found in other clusters.

Expression levels of *ALDH1A2* and *RDH10* is only prominent in cluster 3 RA-Id. In cluster 10 Fbl1 expression levels are similar to that of the Chondrocytes (18 ChT), for example.

Violin / scatter plots showing the normalized expression of *ALDH1A2* and *RDH10* per cluster and cell. Only *bona fide* autopodial (*HOXD13*+, *HOXA13*+, *MSX1*+) bat FL cells are shown.

5b. Low-level expression of these genes would suggest active suppression or downregulation of the RA pathway. For example, in the *Meis2* mutant mouse model (Fig. 4C), *Aldh1a2* expression is downregulated in *Meis2*+ cells, which could argue for active suppression of RA signaling in these cells.

In our *MEIS2* mutant, we ended up driving expression in mainly 2 clusters. Namely naive distal cells, and the RA-Id cells. Here, indeed, we observe a reduction in the expression of *Aldh1a2*,

which could hint to an active downregulation of the RA pathway. To point this out, we suggest to add the following sentence in the result section:

Insertion in line 328: Interestingly, we see a downregulation of *Aldh1a2* and thus apoptotic signaling in *Meis2* overexpressing cells.

Interestingly, we see a downregulation of *Aldh1a2* in the cells of cluster 3 RA-Id, where *Meis2* is ectopically expressed.

In addition, we added to the discussion (see above 4.): Furthermore, as shown by our transgenic experiments, overexpression of *Meis2* results in a downregulation of *Aldh1a2* indicating that *Meis* itself may have an antiapoptotic effect.

6. The authors emphasize that *Grem1* is not expressed in the apoptotic RA-ID cluster. However, as an extracellular and diffusible protein, *Grem1* can exert a significant paracrine effect without being expressed in these cells. Indeed, *Meis2/Tbx3* fibroblasts express *Grem1*, suggesting that these cells may actively counteract pro-apoptotic BMP signaling, thereby protecting themselves or surrounding cells (e.g., other *Meis2*- fibroblasts in the chiropatagium) from BMP-induced apoptosis. Given that *Grem1* downregulation coincides with the onset of interdigital apoptosis in mice, its prolonged expression in bats could provide an additional layer of apoptosis suppression. Since *Bmpr1a* is essential for interdigital apoptosis, it would be relevant to assess how this receptor is expressed in bat and mouse interdigital regions.

MEIS2+ fibroblasts could indeed be protecting themselves and surrounding cells from BMP-induced apoptosis. However, a paracrine effect is, if occurring, limited since apoptosis is present in the developing bat wing. Moreover, we observe a similar pattern of dying cells as in the bat HL, where *GREM1* is not expressed in the interdigital tissue (Weatherbee et al. 2006).

BMPR1A is lowly and ubiquitously sparsely expressed in our data. An *in situ* experiment on developing bat wings is thus unlikely to reveal any new insights.

tSNE plots showing the normalized expression of *Bmpr1a* in mouse and *BMPR1A* in bat. Only *bona fide* autopodial (*HOXD13+*, *HOXA13+*, *MSX1+*) FL cells are shown.

8. Given that canonical WNT signaling is essential for interdigit tissue specification and maintenance in mice (Malkmus et al., 2024) and that components of this pathway are expressed in bat forewings (Eckalbar et al., 2016), could the authors compare mouse forelimb and bat forewing datasets? Specifically, identifying components of the canonical WNT signaling pathway expressed in the interdigit at E11.5/CS15 and E13.5/CS17 may be relevant. The persistence of WNT signaling pathway component expression in bat interdigits at later stages could provide an alternative mechanistic explanation for interdigit tissue retention in bat forewings. Additionally, given that TBX3 functions as a tissue-specific cofactor of canonical WNT signaling in E10.5 forelimbs (see Zimmerli et al., 2020), incorporating WNT and TBX3 link could strengthen the argument for repurposing the proximal limb program.

This is indeed an interesting point, and could help explain the presence of a cell program in the distal limb. However, direct quantitative comparisons across species are extremely challenging. Moreover, we found only one cell population we know for sure to be specifically located in the interdigital space of both species: 3 RA-Id. We show that a fibroblast population is present in the bats' interdigital space only: 10 Fb11. Therefore, we can only compare "interdigital" cells 3 RA-Id between species. These cells are however independent from interdigital bat fibroblasts, and don't give rise to them. A change in the transcriptome of 3 RA-Id, specially at later stages, would be unlikely to impact the development of the chiroptagium fibroblasts.

For the interest of the reviewer, we show an inter-species analysis of 3 RA-Id cells, and the following core elements of canonical WNT signaling:

Wnt1, Wnt2, Wnt3a, Wnt3, Wnt8a, Fzd9, Fzd6, Fzd3, Fzd7, Fzd5, Fzd1, Fzd8, Fzd4, Fzd2, Fzd10, Lrp5, Lrp6, Dvl1, Dvl2, Dvl3, Tcf4, Tcf712, Tcf24, Tcf711, Tcf23, Tcf25, Tcf15, Tcf15, Tcf20, Tcf3, Tcf12, Tcf21, Tcf19, Tcf7, Lef1, Ror1, Ror2, Rspo2, Rspo1, Rspo4, and Rspo3.

We compare expression between cells of the same limb, and then test if there are species-specific differences (Same approach as in Fig. 1G). Here we compare the 3 RA-Id cells, against the rest of the LPM cells per species. In the early stages we didn't find any species-specific differences, on genes that would be specifically expressed in this cluster.

Scatter plot showing the log2FC comparing cells in the 3 RA-Id cells, against the rest of the LPM-derived cells in the FL early stages (E11.5/CS15). We have compared components of the canonical WNT pathway, and added genes for context: markers of this population, genes expressed in bat interdigital space (from WISH in the literature), 20 random marker genes from other cell populations.

In the late stages, we only found the expression of 2 genes to show interesting expression patterns. *Fzd4* with slightly higher relative expression in mouse, and specific expression of *Rspo3* in the bat.

Scatter plot showing the log2FC comparing cells in the 3 RA-Id cells, against the rest of the LPM-derived cells in the FL late stages (E13.5/CS17). We have compared components of the canonical WNT pathway, and added genes for context: markers of this population, genes expressed in bat interdigital space (from WISH in the literature), 20 random marker genes from other cell populations.

For interest, we performed the same comparison in the only other cell populations we are sure to be specifically located in the interdigits: the bat chiroptagium fibroblasts. Since these are only predominantly present in the wing, we compared the same Cp FL populations in the early and late stage. We found again, only *RSPO3* to be slightly more expressed in the later stage.

Scatter plot showing the log2FC comparing cells in the 10 FbI1 and 7 FbI2 cells, against the rest of the LPM-derived cells in the bat FL late at both stages. We have compared components of the canonical WNT pathway, and added genes for context: markers of this population, genes expressed in bat interdigital space (from WISH in the literature), 20 random marker genes from other cell populations.

Similarly, the potential link between *Tbx3* and *Wnt* is noteworthy, as these factors have been reported to form activation complexes. However, our data do not show enrichment of any specific Wnt component. We also examined whether *Tbx3* overexpression in the mutant mice leads to upregulation of Wnt targets, as has been suggested in colorectal cancer, but we did not find Wnt pathway enrichment in the mutants compared to controls.

Since Wnt signaling appears to play a more general role rather than being specific to chiropatagium specification, we do not believe this would strengthen our argument for gene repurposing. For this reason, we have decided not to include this data in the final version of the manuscript.

9. It remains unclear whether MEIS2 predominantly binds to enhancers or promoters; a peak distribution plot would clarify this.

We identified 9808 MEIS-bound peaks from Cp dFL. Only 303 intersect with promoters. Out of these 244 intersect with H3K27ac peaks, and are considered as active. 3968 peaks are potential enhancers (intersect with H3k27ac peaks but not promoters).

We believe that a distribution plot is not helpful. We suggest the following:

Insertion and correction in line 273: We found 4212 MEIS-binding peaks in active accessible bat genomic regions (ATAC + H3K27ac peaks), of which only 244 correspond to gene promoters. As other TFs, MEIS seems to bind to several enhancer regions across large genomic distances (40), we summed up all MEIS-bound regions per regulatory domain, defined by genome-wide chromatin interaction maps (Hi-C) from developing bat limbs.

10. Can the authors provide further characterization of bat MEIS2-bound regulatory elements? Specifically, what proportion of these elements are conserved (bat vs. mouse)? Are any of these enhancers active in the proximal limbs of bats and mice?

Out of the 9808 MEIS peaks found in distal-FL bat, 4259 are within highly conserved genomic regions (alignable between bat and mouse). Out of the 9808 MEIS peaks in late distal-FL bat, 3755 are in early proximal-FL H3k27ac regions. From the 4259 MEIS2 peaks that are within highly conserved genomic regions 1142 are within mouse distal-FL active H3K27ac regions, and 1293 are within mouse distal-HL active regions.

We suggest the following change to our results section, to include this information:

Insertion in line 273: Only 27% (1142) of the MEIS-binding peaks found in conserved mouse / bat genomic regions (4259) also display signatures of enhancer activity (K3K27Ac enrichment) in the mouse distal forelimb. Based on these data we conclude that distal MEIS2 activity appears to associate with, and thus regulate, a set of genes/enhancers that is different from those in the mouse.

11. Do MEIS2 peaks overlap with bat accelerated regions (BARs) identified in previous studies (Booker et al., 2016; Eckalbar et al., 2016)? That would provide an additional support to the role of Meis2 in bat chiropatagium formation.

We used a liftover to transfer the coordinates of bat accelerated regions (Eckalbar et al., 2016) from *Miniopterus natalensis* to *Carollia perspicillata* with the Zoonomia alignment. We were able to recover 2729 out of the 2796 BARs. In bat distal-FL MEIS peaks, 38/9807 ~0.39% overlap with BARs.

Thus, there is no overrepresentation of BARs in the dFL MEIS peaks. This result is somewhat expected, as the accelerated region analysis examines only a small proportion of the genome—specifically, highly conserved genomic regions. The extent to which accelerated regions contribute to adaptation remains uncertain. It is likely that the genetic basis of bat-specific adaptations primarily lies outside these highly conserved elements.

12. The manuscript describes interdigital tissue retention, but apart from webbing between some digits (Fig. 4G/H), it is difficult to determine what exactly is considered interdigital retention?

Our mutant mice exhibited fusion of two digits due to the persistence of interdigital tissue. This phenotype is commonly referred to as cutaneous syndactyly. However, because these phenotypes are induced by genes that we have identified as candidates for wing development, we have chosen to use in some parts of the manuscript the rather descriptive term “interdigital retention” (which was previously used in Weatherbee et al. 2006) to maintain consistency with the field and facilitate clarity for the reader.

Ultimately, this is a matter of terminology and writing style, as both terms describe the same phenomenon. Given that we have explained the phenotype and provided supporting images, we do not believe it is necessary to change the terminology in the manuscript.

Weatherbee et al 2006. Interdigital webbing retention in bat wings illustrates genetic changes underlying amniote limb diversification.

13. *Tbx3* transgene expression is not shown—does it behave in the same manner as the *Meis2* transgene?

We show the expression of the *MEIS2* transgene only, as we use the same enhancer and backbone for both experiments.

Our single-cell data can provide an answer to this question. We observe in Fig. 4C the clusters in which the respective genes from the mutants showed significantly higher expression ($p\text{-val} < 0.01$ & $\log_2\text{FC} > 0.2$) than in WT cells. These clusters correspond between the mutants, and are the cells we expect based on the enhancer activity assays. An exception is Cluster 4 DP, where *Meis2* in the *MEIS2* mutant had an overexpression close to $\log_2\text{FC}$ of 0.1 (albeit, still within significance).

We have performed a new integration, where we combine *bona fide* autopodial (*Hoxd13+* & *Hoxa13+* & *Msx1+*) cells from both mutants, and the WT limbs, to have the exact same cluster distribution. We performed differential expression of *Meis2*, *Tbx3*, and ca. 6,000 other variable genes, comparing each mutant to the WT cells. We found the same result, both *Meis2* and *Tbx3* are highly expressed in their respective mutants in the same clusters (1 MR, 3 RA-Id, and 4 DP).

It must be noted that in this comparison, for the revision, we only test 6,000 genes, while in the manuscript we test more than 18,000 genes. This impacts the false discovery rate correction of p-value. The differences in blue, with relatively high p-values in this comparison, would fall above a $\text{fdr} > 0.01$ when correcting for 18,000 comparisons.

Differential expression of Meis2 and Tbx3 in their respective mutants, against WT cells. The color scale starts at an fdr of 0.01, everything below is gray.

14. For Figure 4F/H, the sample size (n-values) is not well documented. How many limbs were analyzed, and how many exhibited the interdigit phenotype? It seems the 3D imaging panels in Fig4 and Fig S9 are the same limb buds with different angles. Instead authors should show different representative examples.

The number of mutant limbs analyzed was stated in the methods, lines 1255 and 1257. 4 independent samples were imaged. To make it clear in the main text, we have made the following corrections:

Corrected line 339: all TBX3 mutants displayed fusion of at least two digits (Fig. 4H, $n=4$).

Corrected Figure legend of Fig. 4. F and G 3D imaging of mouse wildtype and mutant limbs at E15.5 ($n=4$).

15. The use of the BMP2 distal enhancer for functional studies is surprising, given that these interdigit cells are likely to undergo apoptosis in mice. The "interdigit retention phenotype" presented in this report is not convincing. The authors should consider alternative explanations; for instance, the moderate phenotype observed could be due to the elimination of Meis2+ cells by interdigital apoptosis.

We selected this enhancer because it is the best and closest option available to specifically overexpress a gene in the distal part of the limb. It is essential to acknowledge the challenges posed by interspecies experiments. In mice, interdigital tissue naturally regresses, meaning that any enhancer or gene expressed in this region would encounter the same limitation. We specifically chose this Bmp2 enhancer because it is well-characterized with strong activity in the distal interdigital tissue before its programmed removal. Indeed, our findings suggest that

by overexpressing these factors some of the tissue between the digits is retained. It is still unclear which cells constitute this tissue.

The high-resolution scans clearly show that the epithelium completely surrounds the digits in the WT, whereas it bridges the digits in the mutants. Thus, there is retention of interdigital tissue in the mutant. These results also illustrate the complexity of digit separation. Other mechanisms beyond apoptosis, such as epidermal cell migration and ECM remodeling, are likely involved. Given that we express only one gene at a time from an entire gene program, we do not consider the resulting phenotype to be merely moderate. Rather, it provides valuable insights into the deregulated genes, their functional roles, and their potential implications in wing development.

We have added a sentence in the discussion to emphasize and acknowledge the limitations on transgenic interspecies studies.

Corrected line 424: [...] partially resembles that observed in bats and leads to tissue retention. The recapitulation of only certain aspects of the wing phenotype is expected, as we are manipulating only one gene at a time from an entire gene program. Moreover, interspecies approaches have inherent limitations, as the forced ectopic expression of these genes occurs in a different molecular and cellular context. It is likely that the expression pattern of *MEIS2* observed in bats requires regulatory changes rendering *MEIS2*...

Minor Corrections

- The cluster comparison appears to be based on pooled developmental stages (Fig. 1 & Fig. S1), which could dilute key differences that may only become apparent at later, apoptosis-relevant stages. Perhaps a direct late-stage comparison between mouse and bat forelimbs would provide a more precise assessment.

We pool all of our cells for this comparison. While it is true that temporal-specific differences might be diluted, any difference strong enough will be detected. Moreover, given the nature of the developing limb, we expect to find cells at different stages of differentiation, regardless of the embryonic stage. This is especially important to our study, since we cannot guarantee that the discrete embryo staging is 100% corresponding between Mm and Cp.

We present here the same plots as in Fig. 1, but only using the late stage cells.

We see that the variability presented in Fig. 1G is very similar to the late-only analysis.

- In Figure 3F, the label reads “gene distance in bp”— it would be helpful to clarify the exact meaning of this term.

Corrected in the figure.

- Line 341: Should reference Fig. 4H.

Corrected line 341: Transversal sections of these limbs confirmed the retention of the tissue between digits II and III, resembling cutaneous syndactyly in both mutants (Fig. 4H).

- Line 343: Should reference Fig. 4I-K.

Corrected line 341: Quantification analyses of these images revealed a significant increase in the overall autopod volume, cell number and connective tissue content in both mutants (Fig. 4I-K).

-Booker, B. M., Friedrich, T., Mason, M. K., VanderMeer, J. E., Zhao, J., Eckalbar, W. L., Logan, M., Illing, N., Pollard, K. S. and Ahituv, N. (2016). Bat Accelerated Regions Identify a Bat Forelimb Specific Enhancer in the HoxD Locus. *Plos Genet* 12, e1005738.

-Eckalbar, W. L., Schlebusch, S. A., Mason, M. K., Gill, Z., Parker, A. V., Booker, B. M., Nishizaki, S., Muswamba-Nday, C., Terhune, E., Nevenon, K., et al. (2016). Transcriptomic and epigenomic characterization of the developing bat wing. *Nat Genet* 48, 528–536.

-Kaltcheva, M. M., Anderson, M. J., Harfe, B. D. and Lewandoski, M. (2016). BMPs are direct triggers of interdigital programmed cell death. *Dev. Biol.* 411, 266–276.

-Mason, M. K., Hockman, D., Curry, L., Cunningham, T. J., Duester, G., Logan, M., Jacobs, D. S. and Illing, N. (2015). Retinoic acid-independent expression of Meis2 during autopod patterning in the developing bat and mouse limb. *EvoDevo* 6, 6.

-Malkmus, J., Morabito, A., Lopez-Delisle, L., Esteban, L.A., Mayran, A., Zuniga, A., Sharpe, J., Zeller, R., Sheth, R. bioRxiv 2024. doi:<https://doi.org/10.1101/2024.12.25.629665> WNT signaling coordinately controls mouse limb bud outgrowth and establishment of the digit-interdigit pattern

-Zimmerli, D., Borrelli, C., Jauregi-Miguel, A., Söderholm, S., Brüttsch, S., Doumpas, N., Reichmuth, J., Murphy-Seiler, F., Aguet, M., Basler, K., Moor, A.E., Cantù, C. TBX3 acts as tissue-specific component of the Wnt/β-catenin transcriptional complex. *Elife*. 2020 Aug 18;9:e58123.

We have replied to the reviewers' invaluable concerns and comments in **PURPLE**. All the changes made to the manuscript, derived from these comments are **highlighted in yellow** in the revised manuscript document.

Reviewers' comments:

Reviewer #1 (Remarks to the Author):

I appreciate the authors for effectively responding to the comments and the manuscript has improved significantly. One small comment is that the submitted manuscript PDF file seems to have markups that are still in suggesting mode. Please process the final changes so it's clear which ones were incorporated.

We thank the reviewer for their positive evaluation of our work and apologize for any confusion regarding the markups in the revised version. They were only formatting changes to meet the journal's article guidelines. In this new version, we have incorporated only the changes in response to the reviewers, highlighted in yellow.

In discussion lines 545-548, I understand the author's idea, but it could be argued that finding similar clusters in the dissected chiroptagium in bat is because of the label transfer. Therefore referring to the dissected chiroptagium data might not be the best entry point to make the statement.

From my understanding, the authors can reformulate this part to explain that: you did find highly similar cell populations in mouse and bat FL, which suggests that the overall expression patterns between cell types in these structures are conserved. However, by various analyses and assays, you discovered that the morphological differences came from a distinct regulatory program that introduced a bat-specific developmental trajectory. Then bring up TF repurposing and other evolutionary examples. As the authors mentioned, it is discovered elsewhere that convergence can lead to similar cellular expression patterns in scRNA-seq but then they have different regulatory programs that give them distinct phenotypes.

We truly appreciate the constructive recommendations to improve the study. Following their advice, we have revised the discussion as follows:

Line 543

A major challenge in comparative single-cell analyses lies in data integration, which risks overcorrection and the consequent masking of biological variation⁵¹. This was also of concern during our integration of bat and mouse data, where the interdigital distal fibroblasts forming the bat chiroptagium clustered together with other fibroblasts from both species. However, our independent analyses of the bat-limbs cells revealed conserved composition. Moreover, various analyses, including micro-dissected chiroptagium scRNA-seq, trajectory analyses, and epigenomic profiling, revealed that such clustering was not artifactual. Instead, it reflected the activation of similar transcriptional programs through a distinct regulatory repertoire, ultimately driving a unique bat forewing-specific cell differentiation trajectory. ~~The remarkable similarity we found between cells of such strikingly different limbs raises the question of how this difference is achieved.~~ It is well-documented that during evolution, the same set of genes is often reused⁵². For example, the formation of lateral patagia enabling gliding has independently appeared multiple times in marsupials through convergent evolution, where the upstream factor *Emx2* is activated by distinct regulatory elements in different glider species⁵³.

Reviewer #2 (Remarks to the Author):

The authors have satisfactorily addressed my concerns in their revision.

Reviewer #3 (Remarks to the Author):

I appreciate the authors' thorough responses and the additional data provided, which have significantly improved the manuscript. I believe the study now merits publication in *Nature Ecology & Evolution*.

However, I have remaining concerns regarding the apoptosis assay and its interpretation, which the authors may wish to address. While the shift to an intraspecies comparison is an improvement, I remain unconvinced by certain observations:

We thank the reviewer for their positive comments and appreciation of the work done in the revised manuscript. Below, we address the specific points raised.

- **Caspase-3 Staining:** The bat forelimb's Caspase-3 staining appears different between digits I/II and the remaining digits, especially in the overview image. The brighter, more uniform staining in digits I/II along the proximodistal axis hints at potential differences in apoptotic activity.
- **Forelimb-Hindlimb Comparison:** This comparison may be flawed due to potential heterochrony between the limbs, as observed in mice. Existing bat forelimb and hindlimb WISH data suggest a developmental delay in the hindlimb, potentially undermining its suitability as a control.

We would like to stress that Caspase-3 staining is a qualitative, not a quantitative, method. Therefore, we simply state that the staining appears “*similar*”. We acknowledge that the staining may seem more prominent between digits I-II in the FL compared to digits IV-V. However, we also observe a slightly stronger signal between digits I-II in the HL than between digits IV-V. This is why we refrained from making claims about relative differences in apoptosis between digits, especially since these subtle variations were not observed with the LysoTracker staining.

Our key finding is the presence of apoptosis, confirmed through two independent qualitative assays (LysoTracker and Caspase-3). The quantification comes from the scRNA-seq data. Whether apoptosis is slightly reduced in the FL or whether subtle differences are not captured due to heterochrony between FL and HL is difficult to assess. Our aim was to determine whether apoptosis occurs in the developing bat wing, which we demonstrated using several assays from different perspectives. A detailed study of apoptosis would require a time-series covering the entire developmental process, which is not feasible given the limited availability of bat embryos. Additionally, our later single-cell time points no longer detect these cells, further supporting their elimination through apoptosis.

That said, we acknowledge the reviewers' concerns and have made small adjustments to the text to improve clarity and be more precise in describing the results.

Line 152

We found **pronounced positive** staining in all interdigital zones of bat FLs, with **no discernable minor** differences to interdigit I-II. Likewise, staining in the HL interdigit tissue was similar in intensity and distribution (Fig. 1H and Fig. S2).

Line 159

In summary, our analysis revealed that the cell composition between mouse and bat limbs is highly conserved. Furthermore, cell death, as shown by the qualitative assays used here, occurs similarly is present in all interdigital tissues in the bats regardless of whether the digits get separated or not. However, it appears more intense between digits I-II of the FLs and HLs than in the other digits. Although it is difficult to compare between species, our results show that interdigital apoptosis is a feature of both bats and mice.

- **Imaging Methodology:** The imaging technique lacks clarity. While the figures originate from whole-mount samples, it's unclear whether they are maximum intensity projections or selected z-stacks. If they are selected Z stacks- it would benefit from additional data points or 3D reconstructions to support the conclusions.

The images are the result of the maximum intensity projections, as it stated in the method section, line 1240. For clarity we have modified the method section as follows:

Whole-mount limbs were then imaged with a Zeiss LSM880 confocal laser-scanning microscope in fast-Airyscan mode. At least 20 Z-stacks were imaged, covering the entire limbs. Z-stacks were then merged as Maximum intensity projection with the ZEN software and Airyscan processing was performed. Scale bars were added with Fiji.

- **Biological Relevance of 3 RA-ID Cluster Differences:** The biological relevance of the differences within the 3 RA-ID cluster remains unclear. While the expression levels of apoptotic genes may appear similar, the number and spatial distribution of RA-ID cells may be of greater significance. The analysis, which relies on relative expression comparisons (3 RA-ID vs. mesenchymal cells), does not clearly explain—in either the text or figures—how these comparisons inform our understanding of potential differences in the abundance or location of RA-ID cells between species.

We appreciate the reviewer's feedback. That information was already in the manuscript, but we are happy to clarify it again.

Regarding the abundance of cluster 3 RA-ID cells, Extended Data Fig. 2 shows that the proportion in bats is very similar to that in mice. This figure provides the relative cell proportions for each cluster, though it only states the broader percentage for the main lineages. As requested, the specific percentages are 5% for mouse FL, 4.3% for bat FL, and 3.2% and 3.8% for mouse and bat hindlimbs respectively, confirming the similarity we highlight in the manuscript. These percentages are now being added to the caption of Extended Data Fig. 2. However, we must note that cell population proportions on two very differently shaped limbs are not informative. Even when two limbs were to regress interdigits, e.g. mouse and chicken hindlimbs, the shapes and sizes are so different, that expecting a 100% conservation of RA-active cell proportion is not sustained.

Regarding spatial distribution, dissociative single-cell sequencing methods like the ones used by us, are defined by the loss of spatial context. We, however, know that these cells are located in the same space based on the expression of genes which have been spatially identified using WISH, like *Aldh1a2* and *Meis2*. These have been previously reported to be conserved (Mason et al., 2015).

The biological relevance is that this terminally differentiated cluster is conserved across species, and absent in later stages of bat wing development, suggesting a strong evolutionary constraint on this pathway. This, leading to an alternative mechanism where fibroblasts proliferate in this space repurposing an existing gene program to drive wing formation, as evidenced by our data.

Despite these reservations, the revisions and the incorporation of the idea that differentiation may be a prerequisite for apoptosis evasion is appropriate and broaden the interpretation of the findings.